# 🧑‍💻 Talk2Event: Grounded Understanding of Dynamic Scenes from Event Cameras

**Lingdong Kong**[1,2,*]  **Dongyue Lu**[1,*]  **Ao Liang**[1,*]  **Rong Li**[3]  **Yuhao Dong**[4]
**Tianshuai Hu**[5]  **Lai Xing Ng**[6,†]  **Wei Tsang Ooi**[1,†]  **Benoit R. Cottereau**[7,8,†]

[1]NUS  [2]CNRS@CREATE  [3]HKUST(GZ)  [4]NTU  [5]HKUST  [6]I[2]R, A*STAR
[7]IPAL, CNRS IRL 2955, Singapore  [8]CerCo, CNRS UMR 5549, Université Toulouse III

[*]Equal Contributions  [†]Corresponding Authors

🌐 **Project Page:** `talk2event.github.io`  😺 **Code & Dataset:** `talk2event/toolkit`

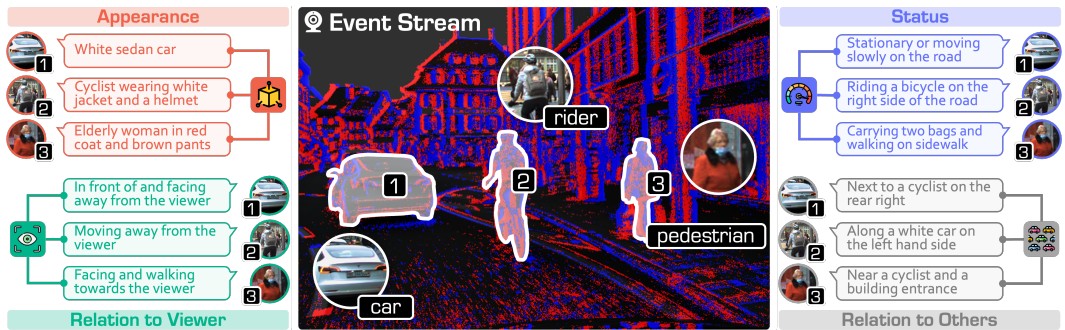

Figure 1: **Grounded scene understanding from event streams.** This work presents 🧑‍💻 Talk2Event, a novel task for localizing objects from event cameras using natural language, where each unique object in the scene is defined by **four key attributes**: ①`Appearance`, ②`Status`, ③`Relation-to-Viewer`, and ④`Relation-to-Others`. We find that modeling these attributes enables precise, interpretable, and temporally-aware grounding across diverse dynamic environments in the real world.

## Abstract

Event cameras offer microsecond-level latency and robustness to motion blur, making them ideal for understanding dynamic environments. Yet, connecting these asynchronous streams to human language remains an open challenge. We introduce **Talk2Event**, the first large-scale benchmark for *language-driven object grounding* in event-based perception. Built from real-world driving data, we provide over 30,000 validated referring expressions, each enriched with four grounding attributes – *appearance*, *status*, *relation to viewer*, and *relation to other objects* – bridging spatial, temporal, and relational reasoning. To fully exploit these cues, we propose **EventRefer**, an *attribute-aware grounding framework* that dynamically fuses multi-attribute representations through a *Mixture of Event-Attribute Experts (MoEE)*. Our method adapts to different modalities and scene dynamics, achieving consistent gains over state-of-the-art baselines in event-only, frame-only, and event-frame fusion settings. We hope our dataset and approach will establish a foundation for advancing multimodal, temporally-aware, and language-driven perception in real-world robotics and autonomy.

## 1 Introduction

Event cameras [11, 23, 80] have emerged as a promising alternative to traditional frame-based sensors, offering unique advantages such as microsecond-level latency [107, 6], low power consumption

39th Conference on Neural Information Processing Systems (NeurIPS 2025).

[75, 79, 22], and robustness to motion blur and low-light conditions [77, 38, 40, 20]. These properties make event cameras highly suitable for high-speed and dynamic scenarios, as demonstrated in various perception tasks including detection [30, 27, 63], segmentation [82, 31, 32, 45], and visual odometry [10, 70, 35]. However, a key capability remains unexplored in the event domain: **visual grounding** – the ability to localize objects in the scene based on free-form language descriptions.

Visual grounding [93, 54] is a cornerstone of multimodal perception, enabling applications such as human-AI interaction, language-guided navigation, and open-vocabulary object localization [91, 40]. While extensive efforts have been made in frame-based [99, 97, 96] and 3D grounding [100, 102, 13, 1, 106, 51] across images [95], videos [55], and remote sensing data [81, 104, 47, 110], these benchmarks are built upon dense sensors that struggle under motion blur, lighting changes, or fast-moving objects. Despite their advantages, event cameras have not been studied in this context, leaving open questions about how to bridge asynchronous sensing with free-form, natural language.

To fill this gap, we introduce Talk2Event, the first benchmark for *language-driven object grounding* in event-based perception. The dataset provides 5,567 scenes, 13,458 annotated objects, and 30,690 high-quality referring expressions. To move beyond coarse descriptions, we introduce **four grounding attributes** – ①Appearance, ②Status, ③Relation-to-Viewer, and ④Relation-to-Others – that explicitly capture spatiotemporal and relational cues critical for grounding in dynamic environments. As shown in Fig. 1, our dataset features multi-caption supervision and fine-grained attribute annotations, setting a new standard for multimodal, temporally-aware event-based grounding.

Complementing the dataset, we propose EventRefer, an **attribute-aware grounding** framework that models the four grounding attributes via a *Mixture of Event-Attribute Experts (MoEE)*. MoEE dynamically fuses attribute-specific features, allowing the model to adapt to appearance, motion, and relational cues. By treating attributes as co-located pseudo-targets, our design provides dense supervision without increasing decoder complexity. At inference, a lightweight fusion selects the most informative attributes for precise grounding. Supporting event-only, frame-only, and event-frame fusion, EventRefer outperforms strong baselines [37, 58], especially in dynamic scenes.

The key contributions of this work can be summarized as follows:

- Talk2Event, the *first* large-scale event-based visual grounding benchmark, with linguistically rich and attribute-aware annotations spanning 5,567 scenes and 30,690 expressions.

- A multi-attribute annotation protocol that captures appearance, motion, egocentric relations, and inter-object context, enabling interpretable and compositional grounding.

- EventRefer, an attribute-aware grounding framework with a mixture of event-attribute experts, achieving state-of-the-art performance across event-only, frame-only, and fusion settings.

## 2 Related Work

**Dynamic Visual Perception.** Event cameras have advanced dynamic scene understanding under high-speed or low-light conditions, supported by benchmarks in driving and indoor scenarios [12, 3, 28, 112, 73] and synthetic datasets for scalable training [42, 25, 17]. Recent works address robustness to noise and illumination changes [14, 115, 89], extending applications to action recognition [74, 109, 5] and autonomous driving [7, 116, 19, 36, 69]. Popular tasks include object detection [26, 117, 111, 48, 27, 30, 65, 98], semantic segmentation [2, 24, 85, 84, 82, 44, 39, 94, 4], optical flow [29, 113, 114], and SLAM or odometry [76, 35]. However, these focus on geometric or low-level semantics, leaving open-vocabulary grounding unexplored. Talk2Event fills this gap as the first benchmark linking event data and natural language for multimodal, temporally grounded understanding.

**Visual Grounding.** Object localization from RGB images has been widely studied using region-ranking [86, 88, 97] and transformer-based methods [41, 37]. These models typically learn from short phrases on static datasets [78, 49, 15], with extensions to video grounding [53] and RGB-D scenes [13, 1, 106, 100]. Despite advances, existing datasets rely on dense frames or depth, lacking temporally sparse, high-speed sensing like events. Our work introduces the first benchmark and method for grounding in asynchronous event data, where our proposed EventRefer further models attribute-aware reasoning to bridge motion, spatial, and relational cues in event streams.

**Multimodal Dynamic Scene Understanding.** Beyond RGB, grounding has been explored in point clouds [102] and remote sensing [81, 104, 47, 110]. 3D methods either rely on proposal-based

Table 1: **Summary of visual grounding benchmarks**. We compare datasets from aspects including: [1]**Sensor** (📷 Frame, 📺 RGB-D, 🖥 LiDAR, 🎥 Event), [2]**Type**, [3]**Statistics** (number of scenes, objects, referring expressions, and average length per caption), and supported [4]**Attributes** for grounding, *i.e.*, ①Appearance ($\delta_a$), ②Status ($\delta_s$), ③Relation-to-Viewer ($\delta_v$), ④Relation-to-Others ($\delta_o$).

| Dataset | Venue | Sensory Data | Scene Type | Statistics | | | | Attributes | | | |
| --- | --- | --- | --- | --- | --- | --- | --- | --- | --- | --- | --- |
| | | | | Scene | Obj. | Expr. | Len. | $\delta_a$ | $\delta_s$ | $\delta_v$ | $\delta_o$ |
| RefCOCO+ [101] | ECCV'16 | 📷 | Static | 19,992 | 49,856 | 141,564 | 3.53 | ✓ | ✗ | ✗ | ✗ |
| RefCOCOg [101] | ECCV'16 | 📷 | Static | 26,711 | 54,822 | 85,474 | 8.43 | ✓ | ✗ | ✗ | ✓ |
| Nr3D [1] | ECCV'20 | 📷📺 | Static | 707 | 5,878 | 41,503 | - | ✓ | ✗ | ✗ | ✓ |
| Sr3D [1] | ECCV'20 | 📷📺 | Static | 1,273 | 8,863 | 83,572 | - | ✓ | ✗ | ✗ | ✓ |
| ScanRefer [13] | ECCV'20 | 📷📺 | Static | 800 | 11,046 | 51,583 | 20.3 | ✓ | ✗ | ✗ | ✓ |
| Text2Pos [43] | CVPR'22 | 🖥 | Static | - | 6,800 | 43,381 | - | ✓ | ✗ | ✗ | ✗ |
| CityRefer [68] | NeurIPS'23 | 🖥 | Static | - | 5,866 | 35,196 | - | ✓ | ✗ | ✗ | ✓ |
| Ref-KITTI [90] | CVPR'23 | 📷 | Static | 6,650 | - | 818 | - | ✓ | ✗ | ✓ | ✗ |
| M3DRefer [105] | AAAI'24 | 📷 | Static | 2,025 | 8,228 | 41,140 | 53.2 | ✓ | ✗ | ✓ | ✗ |
| STRefer [52] | ECCV'24 | 📷🖥 | Static | 662 | 3,581 | 5,458 | - | ✓ | ✗ | ✗ | ✗ |
| LifeRefer [52] | ECCV'24 | 📷🖥 | Static | 3,172 | 11,864 | 25,380 | - | ✓ | ✗ | ✗ | ✗ |
| 🧑‍🤝‍🧑 Talk2Event | Ours | 📷🎥🖥 | **Dynamic** | 5,567 | 13,458 | 30,690 | 34.1 | ✓ | ✓ | ✓ | ✓ |

matching [21, 103, 59, 46] or direct regression [64, 56, 33, 60]. Recent works have started to explore vision-language models for event data [50, 57], but none address grounding with spatial boxes or multi-attribute reasoning. To push beyond frame-based methods, our work expands the frontier of multimodal grounding by introducing the first large-scale benchmark and method for grounding in event-based dynamic scenes, while connecting to broader efforts in multimodal scene understanding.

## 3 🧑‍🤝‍🧑 Talk2Event: Dataset & Benchmark

In this section, we first introduce the formal task definition of event-based visual grounding and its multimodal grounding objectives (Sec. 3.1), and then present the data curation pipeline of Talk2Event, featuring linguistically rich, attribute-aware annotations built on real-world driving data (Sec. 3.2).

### 3.1 Task Formulation: Visual Grounding from Event Streams

**Problem Definition.** We define event-based grounding as the task of localizing an object in dynamic scenes captured by event cameras, based on a free-form language description. Formally, given a voxelized event representation $\mathbf{E}$ and a referring expression $\mathcal{S} = \{w_1, w_2, \ldots, w_C\}$ of $C$ tokens, the goal is to predict a bounding box $\hat{\mathbf{b}} = (x, y, w, h)$ that correctly localizes the referred object.

Event cameras produce asynchronous streams of events $\mathcal{E} = \{e_k\}_{k=1}^N$, where each event $e_k = (x_k, y_k, t_k, p_k)$ encodes the spatial coordinates, timestamp, and polarity $p_k \in \{-1, +1\}$. Following prior work [30, 63], we discretize the stream into a spatiotemporal voxel grid, that is:

$$\mathbf{E}(p, \tau, x, y) = \sum_{e_k \in \mathcal{E}} \delta(p - p_k), \delta(x - x_k, y - y_k), \delta(\tau - \tau_k), \qquad (1)$$

where $\tau_k = \left\lfloor \frac{t_k - t_a}{t_b - t_a} \times T \right\rfloor$ maps each timestamp to one of $T$ temporal bins within the observation window $[t_a, t_b]$. This process produces a dense 4D tensor $\mathbf{E} \in \mathbb{R}^{2 \times T \times H \times W}$ that preserves the spatiotemporal structure and polarity of the events, making it compatible with modern backbones.

**Benchmark Configuration.** In addition to the event voxel grid $\mathbf{E}$, our benchmark optionally provides synchronized frames $\mathbf{F} \in \mathbb{R}^{3 \times H \times W}$ captured at timestamp $t_0$. This design supports three evaluation configurations: grounding with [1]event data only, [2]frame data only, or a [3]combination of both. This setup allows systematic analysis of individual modalities and their fusion in dynamic scenes.

**Grounding Objectives.** To facilitate fine-grained, interpretable, and compositional grounding, we annotate each referring expression with **four attribute categories** that capture complementary aspects of the target object and its surrounding context:

• Appearance: Describes *static visual properties* of the object, such as category, color, size ("large", "small"), and geometric shape. This attribute supports traditional appearance-based localization.

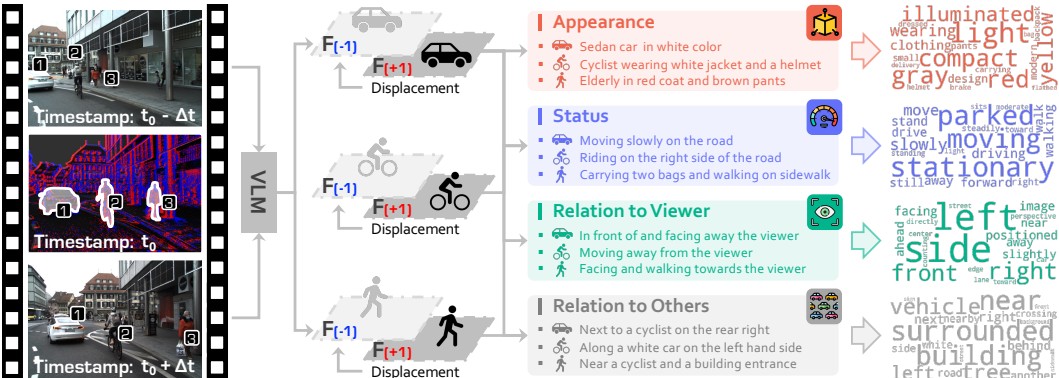

Figure 2: **Pipeline of dataset curation.** We leverage two surrounding frames at $t_0 \pm \Delta t$ to generate context-aware referring expressions at $t_0$, covering appearance, motion, spatial relations, and interactions. Word clouds on the right highlight distinct linguistic patterns across the four grounding attributes.

• `Status`: Refers to *dynamic behaviors or states*, including motion (*e.g.*, "moving", "stopped"), trajectory (*e.g.*, "turning", "approaching"), or action (*e.g.*, "crossing"). This is crucial for grounding objects in dynamic environments captured by high temporal-resolution sensors.

• `Relation-to-Viewer`: Captures *egocentric spatial relationships* between the object and the observer, such as position (*e.g.*, "on the left", "in front"), distance (*e.g.*, "nearby", "far"), or perspective (*e.g.*, "facing towards", "looking in the same direction"). This supports view-conditioned grounding.

• `Relation-to-Others`: Models *relational context with other objects* in the scene, such as spatial arrangements (*e.g.*, "next to a bus", "behind a car") or joint configurations (*e.g.*, "two pedestrians walking together"). This enables context-aware disambiguation in crowded or complex scenes.

We design these four attributes to explicitly expose the diverse spatiotemporal cues that are critical for grounding in event-based streams (along with the optional frames for more appearance and semantic cues). As summarized in Tab. 1, existing grounding benchmarks primarily focus on static scenes and lack such structured attribute-level supervisions, which are crucial for understanding dynamic scenes.

### 3.2 Dataset Curation

We build Talk2Event on top of DSEC [28], a large-scale dataset featuring time-synchronized events and high-resolution frames captured in diverse urban environments. Our goal is to transform this raw sensory data into a comprehensive event-based visual grounding benchmark with linguistically rich and attribute-aware annotations, as depicted in Fig. 1. Below, we detail our curation pipeline.

**Context-Aware Referring Expression Generation.** As illustrated in Fig. 2, we leverage temporal context to generate rich and diverse referring expressions. Given two surrounding frames at $t_0 - \Delta t$ and $t_0 + \Delta t$ ($\Delta t = 200$ ms), we prompt Qwen2-VL [87] to describe the target object at $t_0$. This context exposes object displacement and scene dynamics, encouraging descriptions that capture both appearance and motion, as well as spatial and relational cues. We generate three distinct expressions per object, refined through human validation for correctness and diversity. On average, each object is described by $34.1$ words – making Talk2Event one of the most linguistically rich grounding datasets. Attribute-specific word clouds in Fig. 2 further highlight how our prompting covers the four attributes introduced in Sec. 3.1. Due to space limits, detailed elaborations are placed in the **Appendix**.

**Attribute Annotation and Verification.** Each expression is further decomposed into four compositional attributes – ①`Appearance` ($\delta_a$), ②`Status` ($\delta_s$), ③`Relation-to-Viewer` ($\delta_v$), and ④`Relation-to-Others` ($\delta_o$) – using a semi-automated pipeline that combines fuzzy matching with language model assistance. Human verification ensures the semantic accuracy of these annotations, providing structured and interpretable supervision for multi-attribute grounding.

**Quality Assurance.** We apply rigorous filtering and validation to ensure data quality: (i) *visibility filtering* removes small, occluded, or ambiguous objects; (ii) *redundancy filtering* ensures that the three captions per object are linguistically distinct; (iii) *attribute validation* checks that each caption

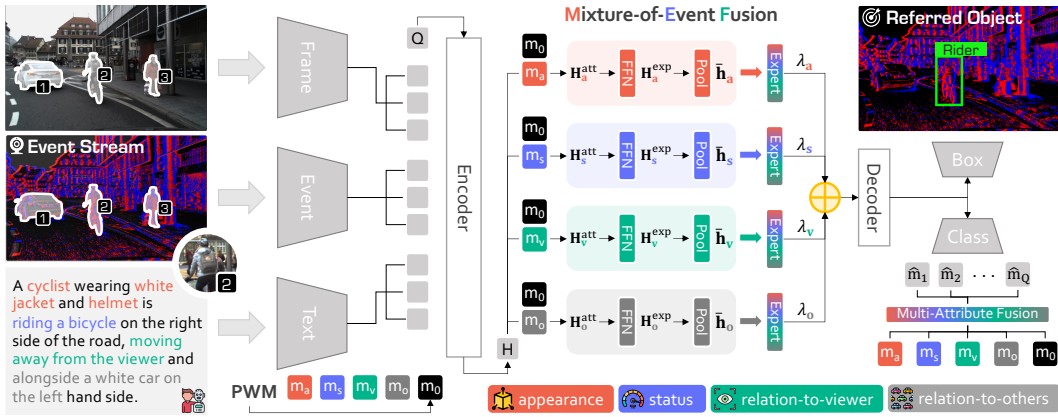

Figure 3: **Overview of architecture.** Given event stream $\mathbf{E}$, frame $\mathbf{F}$ (optional), and the corresponding referring expression $\mathcal{S}$, we aim to ground the target (object #2 in this example) from the scene using multi-attribute fusion. We first match each attribute's cue phrase into a token-level map (Sec. 4.1). The *Mixture of Event-Attribute Experts* masks, refines, and fuses event–text features to produce the fused representation (Sec. 4.2). *Multi-Attribute Fusion* treats the four attributes as co-located pseudo-targets and, at inference, combines their scores to select the final bounding box. (Sec. 4.3)

meaningfully references at least one attribute. This pipeline yields $5{,}567$ curated scenes, $13{,}458$ annotated objects, and $30{,}690$ high-quality referring expressions, establishing Talk2Event as a robust resource for studying multimodal, language-driven grounding in dynamic environments.

Due to space limits, additional annotation details and dataset examples are placed in the **Appendix**.

## 4 EventRefer: Attribute-Aware Grounding Framework

Building upon Talk2Event, we introduce a novel grounding framework that leverages the rich but implicit information in event streams (see Fig. 3). Different from frames, events carry almost no appearance texture; however, their asynchronous nature excels at capturing *motion cues* and *relationships among objects* over time. We aim to inject missing semantic cues, preserve the fine-grained temporal resolution of events, and explicitly model the contribution of each attribute.

### 4.1 Positive Word Matching from Attributes

Grounding requires knowing *where* in an expression the *referred* object or its attributes are mentioned. Manual token-span labels, as provided for RGB corpora [37, 41], are impractical for the four attributes and the expressions in Talk2Event. Instead, we employ a lightweight fuzzy matcher: for each attribute $\delta_i$ in $\{\delta_\mathbf{a}, \delta_\mathbf{s}, \delta_\mathbf{v}, \delta_\mathbf{o}\}$, *e.g.*, a short cue phrase such as "*moving left*" for *status* ($\delta_\mathbf{s}$), the matcher will locate all occurrences (with synonym handling) in the raw referring expression.

The expression is then tokenized, each matched character span is projected onto token indices to form a binary positive map $\mathbf{m}_i \in \{0,1\}^C$, where $[\mathbf{m}_i]_j = 1$ if and only if token $j$ lies inside any span for attribute $\delta_i$, and $C$ is the encoded token length. Finally, we apply softmax to $\mathbf{m}_i$, assigning equal probability to each positive position and $0$ elsewhere, which encourages the model to attend to all tokens expressing attribute $\delta_i$ while remaining robust to paraphrasing.

### 4.2 MoEE: Mixture of Event-Attribute Experts

**Attribute-Aware Masking.** The event features are extracted with a recurrent Transformer backbone following RVT [30], while the referring expressions are embedded with RoBERTa [61]. As shown in Fig. 3, we concatenate these embeddings together and feed them into a DETR [9]-style Transformer encoder, yielding hidden states $\mathbf{H} \in \mathbb{R}^{B \times Q \times C}$, where $B$ is the batch size, $Q$ is the number of queries, and $C$ is the channel dimension. For each attribute $\delta_i$, we construct a binary mask $\mathbf{m}_i^{\text{att}} = \mathbf{m}_i \vee \mathbf{m}_0$, where $\mathbf{m}_0$ represents the union of all tokens that do not belong to any specific attribute, providing general contextual information (referred to as *public context*). This ensures that attribute-specific

reasoning retains surrounding context, improving robustness to incomplete or noisy attribute cues. Applying the mask gives the attribute-specific hidden states $\mathbf{H}_i^{\text{att}} = \mathbf{m}_i^{\text{att}} \odot \mathbf{H}$, which spotlight the positions relevant to attribute $\delta_i$ while retaining neighbouring context. These four parallel features are then passed to the mixture-of-experts fusion module for further information processing.

**Mixture-of-Experts Fusion.** Each attribute-aware feature $\mathbf{H}_i^{\text{att}}$ is first refined by a lightweight FFN, producing expert features $\mathbf{H}_i^{\text{exp}}$. We mean-pool over the query dimension to obtain a compact descriptor $\bar{\mathbf{h}}_i \in \mathbb{R}^{B \times C}$ for each expert. The four descriptors are concatenated and passed through a learnable projection $\mathbf{W} \in \mathbb{R}^{C \times 4}$ to generate gating logits. Following previous work [108], a small Gaussian perturbation encourages exploration, that is:

$$\boldsymbol{\lambda} = \texttt{softmax}(([\bar{\mathbf{h}}_1; \bar{\mathbf{h}}_2; \bar{\mathbf{h}}_3; \bar{\mathbf{h}}_4]\, \mathbf{W}) + \sigma\epsilon), \quad \epsilon \sim \mathcal{N}(0, 1), \tag{2}$$

where $\sigma$ is a learnable scale. The final fused representation is $\mathbf{H}^{\text{fuse}} = \sum_{i=1}^{4} \lambda_i \mathbf{H}_i^{\text{exp}}$. The weights $\lambda_i$ adaptively emphasize whichever attribute cues are most informative for the current sample (*e.g.*, motion cues at night, appearance cues in daylight) and make the model's prediction process more interpretable: large signals of *status* ($\delta_{\text{s}}$) imply reliance on motion, whereas a high weight of *relation-to-viewer* ($\delta_{\text{v}}$) highlights egocentric relations. The injected noise prevents early collapse to a single expert and empirically improves robustness across lighting and speed variations.

## 4.3 Effective Multi-Attribute Fusion for Grounding

During training, every data sample yields one ground-truth box $\mathbf{b}$ and four attribute token maps $\{\mathbf{m}_i\}_{i=1}^{4}$. The decoder, however, produces a *single* token-distribution logit $\hat{\mathbf{m}}_n \in \mathbb{R}^C$ for each query $n$, together with its box $\hat{\mathbf{b}}_n$. To exploit every attribute without inflating the head, we treat the four attributes as co-located *pseudo-objects* during matching and later fuse their scores, giving dense, consistent supervision and precise grounding.

**Training as Multi-Object Grounding.** We treat this problem as in a multi-object setting: The target list is duplicated four times (one per attribute), *i.e.*, $\{\mathbf{b}_i\}_{i=1}^{4}$, and the Hungarian matching is applied between queries and the target bounding box. The cost $\mathcal{C}_{(n,i)}$ for assigning query $n$ to the target $i$ is:

$$\mathcal{C}_{(n,i)} = \beta_{\text{box}}\big(\|\hat{\mathbf{b}}_n - \mathbf{b}_i\|_1 + \text{GIoU}(\hat{\mathbf{b}}_n, \mathbf{b}_i)\big) + \beta_{\text{attr}}\, \mathcal{L}_{\text{attr}}\big(\hat{\mathbf{m}}_n, \mathbf{m}_i\big), \tag{3}$$

which combines a box regression term (L1 loss plus the Generalized Intersection over Union GIoU) with the attribute alignment loss $\mathcal{L}_{\text{attr}}$ (cross-entropy on the soft token maps). The weights $\beta_{\text{box}}$ and $\beta_{\text{attr}}$ balance spatial accuracy against textual grounding. Since pseudo-targets share the same box but different token maps, this encourages the decoder to converge to a single spatial prediction with distinct textual alignments. The total loss sums matching costs over all query–target pairs.

**Inference.** At test time, each query outputs one box and its token logit $\hat{\mathbf{m}}_n$. We build the four attribute maps for the caption as in Sec. 4.1, then score every query between target $i$ by $\text{score}_{(n,i)} = \langle \texttt{softmax}(\hat{\mathbf{m}}_n), \texttt{softmax}(\mathbf{m}_i) \rangle$, *i.e.*, the dot-product between predictions and the probability distribution of positive tokens. The final prediction is the box with the highest score. This late fusion works because the box geometry is shared across attributes; what changes is the confidence level at which the query's language head fires on the respective token sets, allowing the model to rely on the attribute that carries the clearest signal for the current scene.

This multi-attribute fusion design lets us exploit all four attributes without enlarging the decoder, then merge their evidence at test time into a single, reliable score, improving grounding accuracy and interpretability while keeping the framework compact.

## 5 Experiments

### 5.1 Experimental Settings

**Baselines & Competitors.** We benchmark EventRefer against three groups of methods. [1]*Frame-Only*: we retrain traditional visual grounding methods [41, 37] on Talk2Event and report zero-shot results from the large-scale generalist models [67, 58, 16]. [2]*Event-Only*: as no event-based grounding method exists, we adapt leading event perception methods [30, 92, 72, 118, 83] by attaching a DETR Transformer and a grounding head. [3]*Event-Frame Fusion*: we re-implement leading event-frame fusion perception methods [27, 8, 111, 63] under the same DETR Transformer and grounding head.

Table 2: **Comparisons among state-of-the-art methods** on the *val* set of the 👥Talk2Event dataset. The methods are grouped based on input modalities. Symbol † denotes our reproductions with *event-only* grounding outputs. Symbol ‡ denotes our reproductions with *event-frame fusion* grounding outputs. All scores are given in percentage (%). The best scores under each metric are highlighted.

| Method | Venue | mAcc | Ped | Rider | Car | Bus | Truck | Bike | Motor | mIoU |
|---|---|---|---|---|---|---|---|---|---|---|
| **● Modality: Frame Only** | | | | | | | | | | |
| MDETR [41] | ICCV'21 | 39.73 | 15.35 | 6.13 | 53.17 | 26.19 | 25.93 | 5.73 | 10.26 | 80.09 |
| BUTD-DETR [37] | ECCV'22 | 48.91 | 22.66 | 20.44 | 61.94 | 33.93 | 35.93 | 16.56 | 17.95 | 84.30 |
| OWL [67] | ECCV'22 | 40.37 | 9.76 | 15.41 | 53.35 | 38.69 | 22.22 | 10.62 | 15.38 | 69.89 |
| OWL-v2 [66] | NeurIPS'23 | 43.41 | 7.32 | 16.35 | 56.17 | 37.50 | 38.15 | 22.08 | 46.15 | 72.81 |
| YOLO-World [16] | CVPR'24 | 34.08 | 16.79 | 26.67 | 39.72 | 31.90 | 35.96 | 17.86 | 28.21 | 59.76 |
| GroundingDINO [58] | ECCV'24 | 44.50 | 15.62 | 8.62 | 57.70 | 32.52 | 41.20 | 11.76 | 64.10 | 68.67 |
| 👥EventRefer | **Ours** | **55.47** | **27.64** | **51.10** | **65.76** | **47.02** | 32.22 | **28.24** | 10.27 | **85.76** |
| **● Modality: Event Only** | | | | | | | | | | |
| RVT† [30] | CVPR'23 | 26.28 | 14.94 | 3.46 | 35.22 | 7.74 | 5.56 | 2.76 | 23.08 | 75.01 |
| LEOD† [92] | CVPR'24 | 24.84 | 14.33 | 6.45 | 32.33 | 9.52 | 10.74 | 4.02 | 25.64 | 74.37 |
| SAST† [72] | CVPR'24 | 26.71 | 14.84 | 8.02 | 34.73 | 7.14 | 12.96 | 5.31 | 20.51 | 74.94 |
| SSMS† [118] | CVPR'24 | 28.22 | 13.92 | 5.35 | 37.89 | 8.93 | 9.26 | 3.18 | 15.38 | 75.14 |
| EvRT-DETR† [83] | arXiv'24 | 29.34 | 15.45 | 5.50 | 39.24 | 7.74 | 9.26 | 3.82 | 15.38 | 75.66 |
| 👥EventRefer | **Ours** | **31.96** | 12.09 | **25.00** | **40.83** | 15.48 | 16.30 | 4.03 | 15.13 | **76.46** |
| **● Modality: Event-Frame Fusion** | | | | | | | | | | |
| RVT‡ [30] | CVPR'23 | 56.76 | 27.64 | 40.09 | 68.88 | 35.71 | 29.63 | 34.82 | 23.08 | 86.64 |
| RENet‡ [111] | ICRA'23 | 56.50 | **33.43** | 32.55 | 68.30 | 39.29 | 34.44 | 31.42 | 17.95 | 87.02 |
| CAFR‡ [8] | ECCV'24 | 57.76 | 27.54 | **52.52** | 68.24 | 38.69 | 32.59 | 38.43 | 25.64 | 86.13 |
| DAGr‡ [27] | Nature'24 | 58.31 | 31.45 | 32.85 | 70.41 | 38.10 | **41.85** | 35.53 | 30.77 | 86.90 |
| FlexEvent‡ [63] | arXiv'24 | 59.40 | 30.39 | 33.50 | 71.34 | **47.85** | 38.58 | **40.74** | **38.46** | 86.83 |
| 👥EventRefer | **Ours** | **61.82** | 31.15 | 44.23 | **73.85** | 41.07 | 41.70 | 39.53 | 33.33 | **87.32** |

We additionally built a simple fusion baseline "RVT+ResNet+Attention". All baselines receive the full referring expression but are supervised only with class-name positive tokens, following prior practice. This emphasizes the gains of our multi-attribute supervision in our approach.

**Implementation Details.** All models are built in PyTorch [71]. For the event-frame fusion model, the frames are encoded with a ResNet-101 [34] pre-trained on ImageNet [18]; multi-scale features are flattened and concatenated, each token having 256 channels. We train with AdamW [62] using learning rates of $1 \times 10^{-6}$ (frame backbone), $5 \times 10^{-6}$ (textual encoder) and $5 \times 10^{-5}$ (remaining layers). The DETR Transformer weights are initialized from BUTD-DETR [37], and the event backbone weights are from FlexEvent [63]. Due to space limits, see the appendix for more details.

**Evaluation Metrics.** Following practice, we report *Top-1 Acc.*, *i.e.*, the proportion of samples whose highest-scoring box overlaps the ground truth by at least the chosen IoU threshold. We use a stringent threshold IoU@0.95 to stress precise localization, and complement it with mean IoU over all predictions for a holistic boundary measure. Please refer to the appendix for more details.

## 5.2 Comparative Study

**Traditional Visual Grounding.** We first compare the traditional frame-based grounding models [41, 37] and more recent generalist models, *i.e.*, OWL-ViT [67], GroundingDINO [58], and YOLO-World [16]. As shown in Tab. 2 (top), EventRefer achieves 55.47% mAcc and 85.76% mIoU, outperforming all baselines. Notably, we observe substantial improvements on small or dynamic objects such as pedestrians (+5.0%) and riders (+24.4%), demonstrating the ability to leverage attribute-level reasoning beyond simple appearance matching.

**Grounding from Event Streams.** In the event-only setting, we compare with state-of-the-art event-based perception methods [30, 92, 72, 118, 83]. As shown in Tab. 2 (middle), despite their strong detection capabilities, these methods are not explicitly designed for language grounding. EventRefer, by contrast, achieves 31.96% mAcc and 76.46% mIoU, outperforming all event-based baselines. Additionally, we find that event-only performance is generally lower than frame-based models, which is expected since event streams lack rich texture and appearance details. However, the ability to capture motion dynamics and temporal changes brings advantages, especially in low-light or high-speed scenarios where frame-based models tend to struggle.

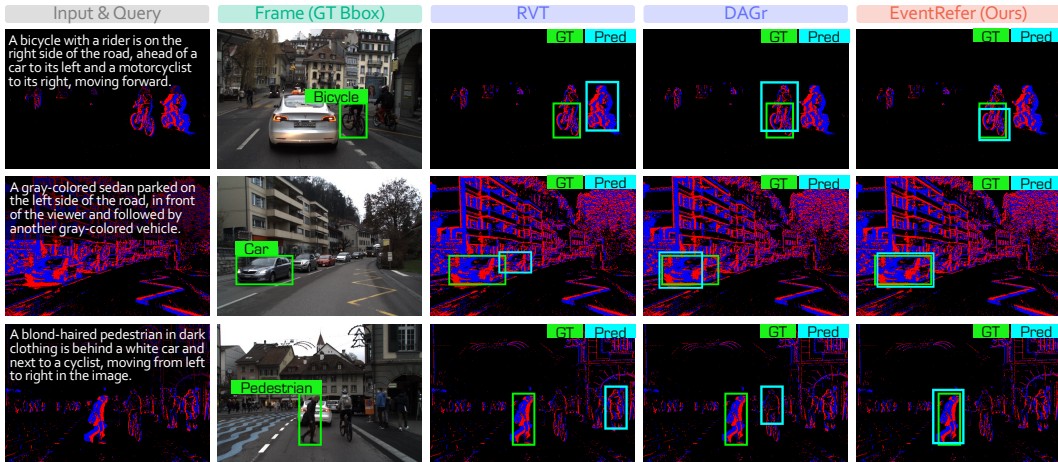

Figure 4: **Qualitative assessment** of grounding approaches on 👥 Talk2Event. The ground truth and predicted boxes are denoted in green and blue colors, respectively. See **Appendix** for more examples.

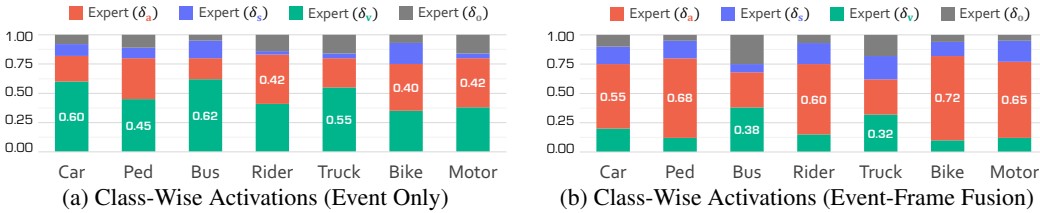

(a) Class-Wise Activations (Event Only)  (b) Class-Wise Activations (Event-Frame Fusion)

Figure 5: **Class-wise attribute expert activations.** We visualize the proportion of each attribute experts in MoEE, under two grounding settings. The top-1 proportion of each class is highlighted.

**Fusion between Event and Frame.** Combining event streams with RGB frames provides complementary benefits, leveraging both high-temporal motion cues and rich appearance information. As shown in Tab. 2 (bottom), EventRefer surpasses all existing fusion baselines such as DAGr [27] and FlexEvent [63]. Notably, we observe consistent improvements across all object categories, particularly rider (+11.7%), bicycle (+4.8%), and truck (+3.1%), indicating that attribute-aware fusion effectively balances appearance, motion cues, and object relationships for robust grounding.

**Qualitative Assessments.** Fig. 4 shows qualitative examples comparing our approach with two strong baselines, RVT [30] and DAGr [27], under the event-frame fusion setting. We observe that previous methods often fail to precisely align the bounding box with the described object due to their limited grounding capability. In contrast, EventRefer produces tighter and more semantically accurate predictions, successfully leveraging attribute-aware reasoning to handle complex descriptions. Due to space limits, please refer to our appendix for additional analyses and visualizations.

### 5.3 Ablation Study

We conduct detailed ablations to analyze the contribution of each design in our framework, using the *event-only* setting throughout. The results are from the validation set of our Talk2Event dataset.

**Component Analysis.** We first evaluate the three key components in EventRefer: positive word matching (PWM), multi-attribute fusion (MAF), and the mixture of event experts (MoEE). As shown in Tab. 3, adding PWM alone improves mAcc from 22.07% to 26.38% by linking token-level supervision with attribute spans. MAF alone achieves 27.01% by enabling independent reasoning over different attributes. Combining both pushes performance to 29.66%, confirming their complementarity. Finally, adding MoEE achieves the best 31.96%, demonstrating the benefit of adaptive expert fusion that dynamically weighs attribute importance across varying scenes.

**Effects of Different Attributes.** Next, we investigate the individual impact of each attribute on grounding performance. As shown in Tab. 4, using only *appearance* ($\delta_a$) achieves a strong baseline of 27.98% mAcc, indicating the importance of visual descriptions such as class shape and object

Table 3: Ablation on **components**: positive word matching (PWM), multi-attribute fusion (MAF), and mixture of event experts (MoEE).

| PWM | MAF | MoEE | mAcc |
|:---:|:---:|:---:|:---:|
| ✗ | ✗ | ✗ | $22.07_{(+0.00)}$ |
| ✓ | ✗ | ✗ | $26.38_{(+4.31)}$ |
| ✗ | ✓ | ✗ | $27.01_{(+4.94)}$ |
| ✓ | ✓ | ✗ | $29.66_{(+7.59)}$ |
| ✓ | ✓ | ✓ | $31.96_{(+9.89)}$ |

Table 4: Ablation on the use of **different attributes** (appearance, status, viewer, others) for event-based visual grounding.

| $\delta_a$ | $\delta_s$ | $\delta_v$ | $\delta_o$ | mAcc (%) |
|:---:|:---:|:---:|:---:|:---:|
| ✓ | ✗ | ✗ | ✗ | $27.98_{(+0.00)}$ |
| ✗ | ✓ | ✗ | ✗ | $28.90_{(+0.92)}$ |
| ✗ | ✗ | ✓ | ✗ | $27.03_{(-0.95)}$ |
| ✗ | ✗ | ✗ | ✓ | $26.97_{(-1.01)}$ |
| ✓ | ✓ | ✓ | ✓ | $31.96_{(+3.98)}$ |

Table 5: Comparisons between MoEE and other strategies for **fusion** of the multi-attribute features.

| Strategy | mAcc |
|:---:|:---:|
| None | $26.38_{(+0.00)}$ |
| Add | $28.39_{(+2.01)}$ |
| Concat | $27.50_{(+1.12)}$ |
| Attention | $29.66_{(+3.28)}$ |
| **MoEE** (Ours) | $31.96_{(+5.58)}$ |

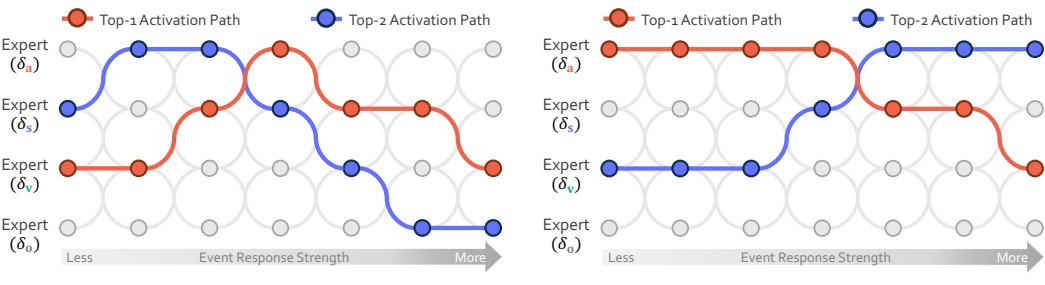

(a) Activation Paths (Event Only)   (b) Activation Paths (Event-Frame Fusion)

Figure 6: **Attribute expert activations.** We analyze the dominant attribute activations (top-1 and top-2) across seven levels of event response strength, from low to high, under two grounding settings.

boundary. Using *status* ($\delta_s$) yields an improvement of $28.90\%$, showing that motion cues provide complementary benefits, especially in dynamic scenes. Interestingly, *viewer-centric* ($\delta_v$) and *object-centric* ($\delta_o$) relations alone yield slightly lower performance, but their combined contribution with appearance and status results in the best performance of $31.96\%$. This highlights the value of modeling all four attributes jointly, as they capture different aspects of the spatiotemporal context.

**Fusion Strategies.** We compare different strategies for fusing the four attribute-aware features. As shown in Tab. 5, simple additive fusion achieves $28.39\%$ mAcc, while concatenation yields a slightly lower $27.50\%$. Attention-based fusion improves performance to $29.66\%$, showing the benefit of learning adaptive weights. However, our proposed MoEE achieves the highest $31.96\%$ mAcc, significantly outperforming all other strategies. This result highlights MoEE's ability to not only fuse attribute features effectively but also to adaptively emphasize the most informative attributes based on scene dynamics, object properties, and modality signals.

**Class-Wise Activations.** We further analyze the average attribute activations for each object class. In the *event-only* setting (Fig. 5a), small dynamic classes such as Rider and Bike rely more on status cues, while larger static objects like Bus and Truck favor appearance and viewer relations. In the *event-frame fusion* setting (Fig. 5b), appearance cues become the most dominant across all classes, yet status and relational cues remain important for highly dynamic or interaction-heavy objects like Pedestrian and Rider. These findings demonstrate that EventRefer not only adapts to input modality but also to object category, promoting interpretable and task-specific grounding behavior.

**Activations vs. Event Response Strength.** To understand how event density affects attribute reliance, we first quantify the *event response strength* by counting the total number of events within a fixed spatial-temporal window. Specifically, we compute the number of activated pixels in the event voxel grid (*e.g.*, $2 \times T \times H \times W$) for each sample. Based on this metric, we group samples into seven levels, from low to high response strength, and visualize the top-1 and top-2 expert activations. In the *event-only* setting (Fig. 6a), viewer-centric *relations* ($\delta_v$) and *appearance* ($\delta_a$) dominate low-response scenes, reflecting reliance on static context when little motion is present. As event density increases, *status* ($\delta_s$) and *relational cues* ($\delta_o$) become more important, capturing the dynamics of moving objects and their interactions. In the *event-frame fusion* setting (Fig. 6b), appearance remains dominant, while status and relational cues gain more influence in highly dynamic scenes. This highlights the adaptive behavior of MoEE in leveraging the most informative attributes based on input conditions.

# 6 Conclusion

We presented Talk2Event, the first large-scale benchmark for language-driven object grounding in dynamic event streams. Built on real-world driving data, we introduce linguistically rich, attribute-aware annotations that capture appearance, motion, and relational context – key factors often overlooked in traditional grounding benchmarks. To tackle this, we proposed EventRefer, an attribute-aware grounding framework that adaptively fuses attribute-specific cues. Extensive experiments demonstrate that our approach outperforms strong baselines. We hope this work will inspire future research at the intersection of event-based perception, visual grounding, and robust multimodal scene understanding.

## Acknowledgments

This work is under the programme DesCartes and is supported by the National Research Foundation, Prime Minister's Office, Singapore, under its Campus for Research Excellence and Technological Enterprise (CREATE) programme.

This work is also supported by the Apple Scholars in AI/ML Ph.D. Fellowship program.

The authors would like to sincerely thank the Program Chairs, Area Chairs, and Reviewers for the time and effort devoted during the review process.

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

# Appendix

## A  The Talk2Event Dataset

In this section, we provide an in-depth description of the Talk2Event dataset, elaborating on its composition, coverage, and key statistics. Our benchmark is constructed atop real-world urban scenes, annotated with object-level language referring expressions enriched with multi-attribute labels. These annotations are designed to support compositional, interpretable, and temporally aware visual grounding from asynchronous event data.

### A.1  Overview

The Talk2Event dataset is built upon the DSEC dataset [28], which offers time-synchronized recordings from event cameras and RGB sensors across diverse driving environments in Switzerland. We repurpose this raw multimodal data into a high-quality grounding benchmark by annotating scenes with bounding boxes and rich language descriptions tied to four grounding attributes.

The dataset is partitioned into a **training split** of $4{,}433$ scenes and a **test split** of $1{,}134$ scenes. These scenes span a wide temporal range, with each sequence providing both high-speed and low-light conditions, ensuring robustness to dynamic and challenging scenarios. In total, we annotate **13,458**

**unique object instances**, with each object grounded via **three distinct referring expressions**, yielding **30,690 validated captions**. All expressions are accompanied by attribute-wise supervision labels, covering appearance, motion, egocentric relation, and relational context.

This design allows models trained on Talk2Event to learn not just where objects are, but also why and how specific language cues map to grounded spatial locations. The benchmark thus bridges fine-grained semantics, visual dynamics, and natural language grounding in a way that was previously unavailable in the event camera literature.

## A.2 Statistics and Analyses

### A.2.1 Dataset Statistics

Table 6 and Table 7 report detailed statistics across the training and test sets.

The training split includes:

- 11,248 total targets, of which 7,675 pass visibility and quality checks;
- 10,321 unique objects, grounded with 23,025 total captions;
- An average caption length of 34.37 words, with a maximum of 80 and a minimum of 12, confirming the diversity and descriptive richness of the language annotations;
- A total labeling effort of 8,612 minutes.

The test set comprises:

- 3,331 total targets, with 2,555 valid targets after filtering;
- 3,137 unique objects and 7,665 total captions;
- An average sentence length of 33.82, with the longest caption reaching 87 words;
- A total labeling effort of 3,370 minutes.

These statistics indicate that Talk2Event contains some of the most linguistically expressive annotations among existing grounding datasets. The significant length and variance in expressions challenge models to handle compositional phrases, motion-related descriptions, and spatial references under real-world constraints.

### A.2.2 Scene and Semantic Distributions

We further analyze how objects are distributed across scenes to understand dataset diversity and grounding complexity. Table 8 and Table 9 enumerate how many objects are grounded per scene in the training and test splits.

- In the **training set**, most scenes include 1–5 objects. Notably, 1,681 scenes contain a single object, while 1,853 scenes feature exactly two objects, supporting both simple and moderately complex grounding cases. A tail of scenes (*e.g.*, 6, 9, or more objects) introduces higher scene density, encouraging relational reasoning and multi-object disambiguation.
- The **test set** exhibits a similar profile, with a balanced mix of sparse and cluttered scenes. For example, 519 test scenes contain two objects, while 365 scenes contain four, and 11 scenes feature more than nine objects. This ensures that evaluation captures both isolated and context-rich object grounding.

To assess semantic diversity, we refer to the class-level distributions (Table 10 and Table 11). Across the seven categories – Car, Pedestrian, Bus, Truck, Bike, Motorcycle, and Rider – the data is relatively well-balanced. Dynamic and small-object classes (*e.g.*, Pedestrian, Bike, Rider) are sufficiently represented, posing additional grounding challenges in scenes with motion blur or complex interactions.

Together, these statistics confirm that Talk2Event provides comprehensive coverage of scene scales, object types, and linguistic attributes. It supports evaluation across key dimensions: from low-level appearance to high-level relations, from sparse to dense contexts, and from static to dynamic scenes – offering a solid foundation for the next generation of event-based grounding methods.

Table 6: Summary of key statistic from the **training set** of the proposed 🧑‍💻Talk2Event dataset.

| # | Sequence | Targets (total) | Targets (valid) | # of Scenes | # of Objects | # of Captions | # of Words Per Captions | | | | Time (minutes) |
|---|---|---|---|---|---|---|---|---|---|---|---|
| | | | | | | | Avg | Med | Max | Min | |
| - | **Summary [training]** | $11,248$ | $7,675$ | $4,433$ | $10,321$ | $23,025$ | 34.37 | 33 | 80 | 12 | 8612.4 |
| 01 | interlaken_00_c | 83 | 50 | 34 | 50 | 150 | 32.83 | 32 | 44 | 22 | 103.8 |
| 02 | interlaken_00_d | 268 | 249 | 213 | 266 | 747 | 29.91 | 30 | 57 | 16 | 160.8 |
| 03 | interlaken_00_e | 300 | 260 | 213 | 290 | 780 | 32.22 | 31 | 62 | 18 | 180.0 |
| 04 | interlaken_00_f | 112 | 58 | 45 | 72 | 174 | 36.84 | 34 | 62 | 24 | 67.2 |
| 05 | interlaken_00_g | 256 | 200 | 92 | 239 | 600 | 34.11 | 33 | 65 | 21 | 128.0 |
| 06 | thun_00_a | 38 | 26 | 20 | 31 | 78 | 29.41 | 28 | 52 | 18 | 47.6 |
| 07 | zurich_city_00_a | 166 | 143 | 103 | 161 | 429 | 32.23 | 31 | 56 | 19 | 207.5 |
| 08 | zurich_city_00_b | 421 | 344 | 151 | 409 | $1,032$ | 32.84 | 32 | 61 | 12 | 252.6 |
| 09 | zurich_city_01_a | 196 | 137 | 61 | 189 | 411 | 34.71 | 33 | 66 | 20 | 98.0 |
| 10 | zurich_city_01_b | 187 | 120 | 90 | 152 | 360 | 33.86 | 33 | 67 | 18 | 93.5 |
| 11 | zurich_city_01_c | 291 | 221 | 115 | 285 | 663 | 35.10 | 34 | 59 | 18 | 145.5 |
| 12 | zurich_city_01_d | 271 | 182 | 94 | 266 | 546 | 32.77 | 32 | 58 | 17 | 135.5 |
| 13 | zurich_city_01_e | 570 | 385 | 194 | 562 | $1,155$ | 35.34 | 35 | 62 | 17 | 285.0 |
| 14 | zurich_city_01_f | 499 | 286 | 134 | 479 | 858 | 34.59 | 33 | 58 | 22 | 249.5 |
| 15 | zurich_city_02_a | 23 | 18 | 17 | 20 | 54 | 33.04 | 32 | 51 | 25 | 13.8 |
| 16 | zurich_city_02_b | 343 | 202 | 106 | 300 | 606 | 35.72 | 35 | 63 | 19 | 205.8 |
| 17 | zurich_city_02_c | 190 | 110 | 63 | 143 | 330 | 36.81 | 35 | 66 | 20 | 114.0 |
| 18 | zurich_city_02_d | 58 | 15 | 12 | 23 | 45 | 33.89 | 32 | 44 | 26 | 34.8 |
| 19 | zurich_city_02_e | 180 | 106 | 68 | 148 | 318 | 36.93 | 36 | 58 | 20 | 108.0 |
| 20 | zurich_city_03_a | 29 | 21 | 21 | 21 | 63 | 33.17 | 32 | 45 | 24 | 36.3 |
| 21 | zurich_city_04_a | 263 | 214 | 82 | 262 | 642 | 39.00 | 39 | 73 | 22 | 328.8 |
| 22 | zurich_city_04_b | 105 | 85 | 33 | 105 | 255 | 36.36 | 35 | 56 | 23 | 131.3 |
| 23 | zurich_city_04_c | 248 | 181 | 109 | 223 | 543 | 35.01 | 34 | 65 | 22 | 310.1 |
| 24 | zurich_city_04_d | 80 | 66 | 41 | 71 | 198 | 31.96 | 31 | 53 | 23 | 40.1 |
| 25 | zurich_city_04_e | 78 | 66 | 29 | 78 | 198 | 31.71 | 31 | 52 | 17 | 39.0 |
| 26 | zurich_city_04_f | 330 | 246 | 107 | 330 | 738 | 32.94 | 32 | 70 | 18 | 165.0 |
| 27 | zurich_city_05_a | 300 | 204 | 129 | 267 | 612 | 36.22 | 35 | 58 | 22 | 375.0 |
| 28 | zurich_city_05_b | 270 | 192 | 117 | 255 | 576 | 34.33 | 33 | 63 | 21 | 337.5 |
| 29 | zurich_city_06_a | 185 | 95 | 71 | 137 | 285 | 34.26 | 33 | 51 | 22 | 111.0 |
| 30 | zurich_city_07_a | 142 | 115 | 78 | 128 | 345 | 31.86 | 31 | 53 | 19 | 177.5 |
| 31 | zurich_city_08_a | 305 | 169 | 73 | 284 | 507 | 38.07 | 38 | 69 | 20 | 381.3 |
| 32 | zurich_city_09_a | 581 | 295 | 136 | 549 | 885 | 39.31 | 39 | 69 | 20 | 726.3 |
| 33 | zurich_city_09_b | 71 | 33 | 24 | 47 | 99 | 33.43 | 31 | 54 | 24 | 88.8 |
| 34 | zurich_city_09_c | 163 | 119 | 94 | 139 | 357 | 32.93 | 33 | 59 | 17 | 203.7 |
| 35 | zurich_city_09_d | 487 | 320 | 166 | 471 | 960 | 34.48 | 33 | 64 | 19 | 608.7 |
| 36 | zurich_city_09_e | 131 | 64 | 45 | 81 | 192 | 32.19 | 31 | 51 | 22 | 163.8 |
| 37 | zurich_city_10_a | 316 | 187 | 78 | 289 | 561 | 33.99 | 32 | 64 | 21 | 158.0 |
| 38 | zurich_city_10_b | 513 | 359 | 205 | 483 | $1,077$ | 33.70 | 32 | 80 | 19 | 256.5 |
| 39 | zurich_city_11_a | 144 | 95 | 50 | 138 | 285 | 39.25 | 39 | 60 | 24 | 180.0 |
| 40 | zurich_city_11_b | 359 | 263 | 167 | 345 | 789 | 40.56 | 41 | 72 | 22 | 179.6 |
| 41 | zurich_city_11_c | 607 | 434 | 228 | 600 | $1,302$ | 33.47 | 32 | 63 | 18 | 303.5 |
| 42 | zurich_city_16_a | 115 | 68 | 64 | 83 | 204 | 35.15 | 35 | 65 | 22 | 143.7 |
| 43 | zurich_city_17_a | 34 | 26 | 24 | 26 | 78 | 36.49 | 37 | 52 | 23 | 42.5 |
| 44 | zurich_city_18_a | 234 | 141 | 93 | 200 | 423 | 31.68 | 31 | 64 | 19 | 76.1 |
| 45 | zurich_city_19_a | 239 | 179 | 117 | 216 | 537 | 33.14 | 32 | 54 | 23 | 77.7 |
| 46 | zurich_city_20_a | 263 | 152 | 103 | 221 | 456 | 33.03 | 32 | 65 | 18 | 85.5 |
| 47 | zurich_city_21_a | 204 | 174 | 124 | 187 | 522 | 34.58 | 33 | 71 | 15 | 255.0 |

## A.3 Dataset Curation Details

We aim to transform raw multimodal sequences into temporally aligned grounding scenes enriched with language and attribute annotations. The curation process consists of: (1) selecting temporally coherent and visually informative scenes; (2) generating referring expressions using a large vision-language model; (3) prompting for four key attribute types; and (4) refining all captions through human verification.

### A.3.1 Data Selection Details

We begin by sampling keyframes from DSEC sequences at $5$ Hz, ensuring temporal diversity while minimizing redundancy. For each selected frame at timestamp $t_0$, we extract a centered event volume spanning $[t_0 - 100\text{ms}, t_0 + 100\text{ms}]$, discretized into $T$ temporal bins as described in the main paper. We retain objects from seven common urban categories: Car, Pedestrian, Bus, Truck, Bike,

Table 7: Summary of key statistic from the **test set** of the proposed 👫Talk2Event dataset.

| # | Sequence | Targets (total) | Targets (valid) | # of Scenes | # of Objects | # of Captions | # of Words Per Captions | | | | Time (minutes) |
|---|---|---|---|---|---|---|---|---|---|---|---|
| | | | | | | | Avg | Med | Max | Min | |
| - | **Summary [test]** | **3,331** | **2,555** | **1,134** | **3,137** | **7,665** | **33.82** | **33** | **87** | **15** | **3370.2** |
| 01 | interlaken_00_a | 102 | 56 | 48 | 56 | 168 | 32.99 | 32 | 56 | 20 | 127.5 |
| 02 | interlaken_00_b | 90 | 45 | 26 | 45 | 135 | 29.62 | 29 | 47 | 18 | 112.5 |
| 03 | interlaken_01_a | 188 | 157 | 97 | 174 | 471 | 31.80 | 31 | 53 | 19 | 235.1 |
| 04 | thun_01_a | 33 | 17 | 10 | 20 | 51 | 30.61 | 27 | 58 | 18 | 41.3 |
| 05 | thun_01_b | 127 | 109 | 60 | 121 | 327 | 34.46 | 33 | 59 | 21 | 158.7 |
| 06 | thun_02_a | 1,737 | 1,423 | 457 | 1737 | 4,269 | 33.69 | 33 | 87 | 16 | 2171.3 |
| 07 | zurich_city_12_a | 196 | 151 | 65 | 190 | 453 | 35.55 | 35 | 56 | 18 | 245.0 |
| 08 | zurich_city_13_a | 67 | 57 | 43 | 64 | 171 | 37.13 | 38 | 52 | 21 | 21.8 |
| 09 | zurich_city_13_b | 67 | 50 | 32 | 64 | 150 | 39.39 | 40 | 58 | 21 | 21.8 |
| 10 | zurich_city_14_a | 28 | 24 | 23 | 26 | 72 | 30.81 | 30 | 45 | 22 | 9.2 |
| 11 | zurich_city_14_b | 107 | 97 | 72 | 107 | 291 | 34.97 | 35 | 56 | 22 | 34.8 |
| 12 | zurich_city_14_c | 178 | 118 | 76 | 156 | 354 | 31.08 | 30 | 53 | 15 | 57.9 |
| 13 | zurich_city_15_a | 411 | 251 | 125 | 377 | 753 | 37.50 | 36 | 68 | 17 | 133.7 |

`Motorcycle`, and `Rider`. Each object is annotated with a 2D bounding box on the synchronized RGB frame, and nearby objects are also recorded to enable relational reasoning.

We filter out occluded, low-resolution, or ambiguous targets based on size, visibility, and contextual clarity. Valid targets must be clearly visible in both event and frame views, and be distinguishable within a reasonable neighborhood. The final corpus includes 5,567 curated scenes across 47 training sequences and 13 test sequences, totaling over 13K objects and 30K captions.

### A.3.2 Referring Expression Generation

To ensure rich and temporally grounded language annotations, we generate referring expressions using Qwen2-VL [87], a state-of-the-art vision-language model. Unlike existing grounding datasets that use a single image, we provide two RGB frames – sampled at $t_0 - \Delta t$ and $t_0 + \Delta t$ (where $\Delta t = 200\text{ms}$) – as context to describe the target object at time $t_0$. This dual-frame design allows the model to observe short-term temporal dynamics, such as motion direction, speed, and interactions with nearby objects, which aligns closely with the nature of event data.

Crucially, our prompting strategy is directly informed by the four grounding attributes defined in our framework: ①`Appearance`, ②`Status`, ③`Relation-to-Viewer`, and ④ `Relation-to-Others`.

We design a structured prompt that explicitly guides the model to describe each of these attributes before composing a final summary sentence. This formulation enables interpretable annotation, consistent multi-attribute supervision, and improved grounding coverage under various scene complexities.

We generate three distinct expressions per object, each based on varied prompts to encourage linguistic diversity. Captions are retained only after human verification (see below). This annotation pipeline yields 30,690 validated expressions with an average length of 34.1 words.

### A.3.3 Prompts

To generate high-quality, attribute-rich descriptions for grounded objects in dynamic scenes, we leverage a two-stage prompting strategy using Qwen2-VL-72B [87]. This process includes both the initial structured generation of captions and a rewriting stage to enhance linguistic diversity while preserving referential clarity.

**Attribute-Guided Generation Prompt.** The first caption for each object is generated using a highly structured prompt that reflects the four grounding attributes defined in our framework: `Appearance`, `Status`, `Relation-to-Viewer`, and `Relation-to-Others`. Importantly, the prompt is conditioned on two frames ($t_0 - \Delta t$ and $t_0 + \Delta t$), allowing the model to infer motion and contextual relationships from temporal changes. The detailed prompt is provided in Table 12.

Table 8: Summary of scene statistics from the **training set** of the proposed 👥Talk2Event dataset.

| # | Sequence | Total Number of Scenes (*w/* Number of Objects Per Scene) | | | | | | | | | | |
| | | Single | 2 | 3 | 4 | 5 | 6 | 7 | 8 | 9 | > 9 | All |
|---|---|---|---|---|---|---|---|---|---|---|---|---|
| - | **Summary** [training] | 1,681 | 1,853 | 1,674 | 1,049 | 685 | 387 | 223 | 95 | 22 | 6 | 7,675 |
| 01 | interlaken_00_c | 20 | 24 | 6 | - | - | - | - | - | - | - | 50 |
| 02 | interlaken_00_d | 176 | 37 | 36 | - | - | - | - | - | - | - | 249 |
| 03 | interlaken_00_e | 151 | 84 | 18 | 7 | - | - | - | - | - | - | 260 |
| 04 | interlaken_00_f | 25 | 24 | 5 | 4 | - | - | - | - | - | - | 58 |
| 05 | interlaken_00_g | 29 | 52 | 39 | 24 | 12 | 18 | 7 | 19 | - | - | 200 |
| 06 | thun_00_a | 11 | 11 | 4 | - | - | - | - | - | - | - | 26 |
| 07 | zurich_city_00_a | 65 | 38 | 32 | 6 | 2 | - | - | - | - | - | 143 |
| 08 | zurich_city_00_b | 36 | 70 | 70 | 78 | 68 | 22 | - | - | - | - | 344 |
| 09 | zurich_city_01_a | 11 | 17 | 44 | 31 | 23 | 11 | - | - | - | - | 137 |
| 10 | zurich_city_01_b | 52 | 32 | 26 | 5 | 4 | 1 | - | - | - | - | 120 |
| 11 | zurich_city_01_c | 38 | 60 | 28 | 24 | 42 | 29 | - | - | - | - | 221 |
| 12 | zurich_city_01_d | 18 | 48 | 44 | 22 | 11 | 37 | 2 | - | - | - | 182 |
| 13 | zurich_city_01_e | 43 | 76 | 107 | 55 | 30 | 41 | 29 | 2 | 2 | - | 385 |
| 14 | zurich_city_01_f | 33 | 49 | 28 | 16 | 26 | 40 | 41 | 41 | 6 | 6 | 286 |
| 15 | zurich_city_02_a | 14 | 4 | - | - | - | - | - | - | - | - | 18 |
| 16 | zurich_city_02_b | 16 | 48 | 53 | 61 | 4 | 13 | 4 | 3 | - | - | 202 |
| 17 | zurich_city_02_c | 20 | 39 | 24 | 7 | 7 | 8 | 5 | - | - | - | 110 |
| 18 | zurich_city_02_d | 3 | 9 | 3 | - | - | - | - | - | - | - | 15 |
| 19 | zurich_city_02_e | 26 | 29 | 22 | 11 | 18 | - | - | - | - | - | 106 |
| 20 | zurich_city_03_a | 21 | - | - | - | - | - | - | - | - | - | 21 |
| 21 | zurich_city_04_a | 17 | 25 | 53 | 29 | 30 | 5 | 50 | 5 | - | - | 214 |
| 22 | zurich_city_04_b | 2 | 8 | 34 | 30 | 11 | - | - | - | - | - | 85 |
| 23 | zurich_city_04_c | 44 | 52 | 58 | 3 | 24 | - | - | - | - | - | 181 |
| 24 | zurich_city_04_d | 24 | 12 | 19 | 11 | - | - | - | - | - | - | 66 |
| 25 | zurich_city_04_e | 4 | 15 | 23 | 21 | 3 | - | - | - | - | - | 66 |
| 26 | zurich_city_04_f | 5 | 45 | 78 | 93 | 20 | 5 | - | - | - | - | 246 |
| 27 | zurich_city_05_a | 46 | 75 | 46 | 22 | 8 | 4 | 3 | - | - | - | 204 |
| 28 | zurich_city_05_b | 46 | 55 | 43 | 24 | 16 | 8 | - | - | - | - | 192 |
| 29 | zurich_city_06_a | 28 | 41 | 12 | 7 | 7 | - | - | - | - | - | 95 |
| 30 | zurich_city_07_a | 48 | 32 | 12 | 15 | 8 | - | - | - | - | - | 115 |
| 31 | zurich_city_08_a | 5 | 15 | 28 | 42 | 48 | 17 | 11 | 3 | - | - | 169 |
| 32 | zurich_city_09_a | 8 | 23 | 54 | 55 | 72 | 36 | 40 | 7 | - | - | 295 |
| 33 | zurich_city_09_b | 6 | 21 | 5 | 1 | - | - | - | - | - | - | 33 |
| 34 | zurich_city_09_c | 58 | 40 | 21 | - | - | - | - | - | - | - | 119 |
| 35 | zurich_city_09_d | 30 | 77 | 68 | 80 | 50 | 15 | - | - | - | - | 320 |
| 36 | zurich_city_09_e | 22 | 19 | 17 | 6 | - | - | - | - | - | - | 64 |
| 37 | zurich_city_10_a | 12 | 27 | 27 | 33 | 18 | 31 | 20 | 5 | 14 | - | 187 |
| 38 | zurich_city_10_b | 66 | 109 | 99 | 21 | 25 | 18 | 11 | 10 | - | - | 359 |
| 39 | zurich_city_11_a | 7 | 22 | 36 | 27 | 2 | 1 | - | - | - | - | 95 |
| 40 | zurich_city_11_b | 54 | 116 | 48 | 25 | 14 | 6 | - | - | - | - | 263 |
| 41 | zurich_city_11_c | 46 | 98 | 163 | 61 | 48 | 18 | - | - | - | - | 434 |
| 42 | zurich_city_16_a | 54 | 3 | 9 | 2 | - | - | - | - | - | - | 68 |
| 43 | zurich_city_17_a | 22 | 4 | - | - | - | - | - | - | - | - | 26 |
| 44 | zurich_city_18_a | 44 | 27 | 26 | 28 | 13 | 3 | - | - | - | - | 141 |
| 45 | zurich_city_19_a | 63 | 34 | 37 | 32 | 13 | - | - | - | - | - | 179 |
| 46 | zurich_city_20_a | 29 | 58 | 46 | 19 | - | - | - | - | - | - | 152 |
| 47 | zurich_city_21_a | 83 | 49 | 23 | 11 | 8 | - | - | - | - | - | 174 |

**Diversity-Driven Rewriting Prompt.** To ensure linguistic variation while preserving grounding quality, we generate two additional captions per object using a rewriting prompt. These rewritten descriptions must remain semantically faithful and uniquely refer to the same object, but differ in phrasing, sentence structure, or vocabulary. This helps enrich the dataset for training models that generalize across varied linguistic inputs. The detailed prompt is provided in Table 13.

The two prompts serve complementary roles in our annotation pipeline:

- The **generation prompt** ensures that each expression comprehensively covers the grounding attributes, encouraging structured and attribute-aligned reasoning.

- The **rewriting prompt** promotes linguistic diversity and robustness, simulating realistic variation in how humans describe the same object under different language styles.

Together, they produce three validated, stylistically distinct captions per object, enhancing both the semantic richness and generalization capacity of the Talk2Event dataset.

Table 9: Summary of scene statistics from the **test set** of the proposed 👥Talk2Event dataset.

| # | Sequence | Total Number of Scenes (*w/* Number of Objects Per Scene) | | | | | | | | | | |
|---|----------|--------|-----|-----|-----|-----|-----|-----|-----|-----|-----|--------|
| | | Single | 2 | 3 | 4 | 5 | 6 | 7 | 8 | 9 | > 9 | All |
| - | **Summary** [test] | **338** | **519** | **446** | **365** | **334** | **218** | **237** | **64** | **23** | **11** | **2,555** |
| 01 | interlaken_00_a | 40 | 16 | - | - | - | - | - | - | - | - | 56 |
| 02 | interlaken_00_b | 13 | 18 | 6 | 8 | - | - | - | - | - | - | 45 |
| 03 | interlaken_01_a | 46 | 65 | 19 | 11 | 16 | - | - | - | - | - | 157 |
| 04 | thun_01_a | 3 | 6 | 8 | - | - | - | - | - | - | - | 17 |
| 05 | thun_01_b | 21 | 44 | 21 | 23 | - | - | - | - | - | - | 109 |
| 06 | thun_02_a | 49 | 188 | 216 | 232 | 230 | 189 | 221 | 64 | 23 | 11 | 1,423 |
| 07 | zurich_city_12_a | 12 | 22 | 45 | 28 | 36 | 8 | - | - | - | - | 151 |
| 08 | zurich_city_13_a | 27 | 17 | 13 | - | - | - | - | - | - | - | 57 |
| 09 | zurich_city_13_b | 12 | 17 | 17 | 2 | 2 | - | - | - | - | - | 50 |
| 10 | zurich_city_14_a | 21 | 2 | 1 | - | - | - | - | - | - | - | 24 |
| 11 | zurich_city_14_b | 49 | 25 | 15 | 8 | - | - | - | - | - | - | 97 |
| 12 | zurich_city_14_c | 28 | 41 | 29 | 13 | 7 | - | - | - | - | - | 118 |
| 13 | zurich_city_15_a | 17 | 58 | 56 | 40 | 43 | 21 | 16 | - | - | - | 251 |

### A.3.4   Human Verification

To ensure the accuracy, clarity, and grounding relevance of all referring expressions in Talk2Event, we incorporate a rigorous human verification stage following automatic caption generation.

**Verification Objectives.**   Each generated caption is manually reviewed by trained annotators to guarantee:

- **Referential Correctness**: The description must uniquely and unambiguously identify the intended object in the scene.
- **Attribute Coverage**: At least two of the four grounding attributes (`Appearance`, `Status`, `Relation-to-Viewer`, `Relation-to-Others`) must be present, and the content should not hallucinate unsupported attributes.
- **Linguistic Fluency**: The caption must be grammatically sound and easily readable, with no syntactic errors or awkward phrasing.

**Verification Interface.**   We develop a lightweight web-based annotation tool to streamline the verification process. For each object instance, the tool presents:

- The event-based scene and corresponding frames (at $t_0 - \Delta t$ and $t_0 + \Delta t$).
- The target object bounding box.
- All three candidate captions generated by the VLM.
- Controls for editing or discarding the caption.

Annotators can cross-reference bounding boxes with visual context to evaluate the semantic alignment of the description and make real-time corrections.

Each caption is reviewed independently. If a caption contains minor issues (*e.g.*, typos, vague words, or incorrect attribute usage), annotators edit the sentence directly. If the caption is fundamentally flawed – such as referencing a wrong object, hallucinating relationships, or being irredeemably ambiguous – it is discarded. Rewriting is only applied when meaningful correction is feasible.

**Effort and Quality.**   In total, this human verification process covers **30,690 expressions** over **13,458 objects** across both training and test sets. The annotation effort amounted to approximately:

- 8,612 minutes for the training set,
- 3,370 minutes for the test set,

performed by a team of five trained annotators over multiple rounds. To ensure consistency and reduce inter-annotator variability, we conducted calibration sessions before annotation and spot-check audits during review.

Table 10: Summary of scene statistics from the **training set** of the proposed ♟Talk2Event dataset.

| # | Sequence | Number of Target Objects | | | | | | | |
|---|---|---|---|---|---|---|---|---|---|
| | | Car | Truck | Pedestrian | Bike | Rider | Bus | Motorcycle | All |
| - | **Summary** [training] | 6,281 | 400 | 394 | 172 | 87 | 192 | 149 | 7,675 |
| 01 | interlaken_00_c | 44 | 1 | 5 | - | - | - | - | 50 |
| 02 | interlaken_00_d | 234 | 4 | 7 | - | 1 | 1 | 2 | 249 |
| 03 | interlaken_00_e | 254 | - | - | 1 | - | 5 | - | 260 |
| 04 | interlaken_00_f | 48 | 3 | 3 | 2 | 1 | - | 1 | 58 |
| 05 | interlaken_00_g | 163 | 2 | 7 | 15 | 13 | - | - | 200 |
| 06 | thun_00_a | 25 | 1 | - | - | - | - | - | 26 |
| 07 | zurich_city_00_a | 138 | - | 3 | - | - | - | 2 | 143 |
| 08 | zurich_city_00_b | 158 | - | 120 | 11 | 3 | 46 | 6 | 344 |
| 09 | zurich_city_01_a | 135 | - | 2 | - | - | - | - | 137 |
| 10 | zurich_city_01_b | 75 | 8 | 15 | - | - | 18 | 4 | 120 |
| 11 | zurich_city_01_c | 209 | 8 | 1 | - | - | 3 | - | 221 |
| 12 | zurich_city_01_d | 159 | 1 | - | - | - | 19 | 3 | 182 |
| 13 | zurich_city_01_e | 318 | 4 | 35 | 14 | 12 | - | 2 | 385 |
| 14 | zurich_city_01_f | 258 | 3 | 4 | - | - | 7 | 14 | 286 |
| 15 | zurich_city_02_a | 18 | - | - | - | - | - | - | 18 |
| 16 | zurich_city_02_b | 146 | 4 | 2 | - | - | 44 | 6 | 202 |
| 17 | zurich_city_02_c | 85 | 16 | 4 | - | - | 4 | 1 | 110 |
| 18 | zurich_city_02_d | 15 | - | - | - | - | - | - | 15 |
| 19 | zurich_city_02_e | 81 | 3 | 10 | 1 | - | 11 | - | 106 |
| 20 | zurich_city_03_a | 21 | - | - | - | - | - | - | 21 |
| 21 | zurich_city_04_a | 158 | 48 | 4 | 1 | - | - | 3 | 214 |
| 22 | zurich_city_04_b | 35 | 33 | 12 | - | - | - | 5 | 85 |
| 23 | zurich_city_04_c | 115 | 52 | 11 | 2 | - | - | 1 | 181 |
| 24 | zurich_city_04_d | 43 | 11 | 2 | 2 | 3 | - | 5 | 66 |
| 25 | zurich_city_04_e | 46 | 20 | - | - | - | - | - | 66 |
| 26 | zurich_city_04_f | 199 | 38 | 3 | 3 | - | 3 | - | 246 |
| 27 | zurich_city_05_a | 148 | 27 | 7 | 8 | - | - | 14 | 204 |
| 28 | zurich_city_05_b | 120 | 8 | 29 | 8 | 1 | 14 | 12 | 192 |
| 29 | zurich_city_06_a | 91 | 1 | 3 | - | - | - | - | 95 |
| 30 | zurich_city_07_a | 89 | 4 | 3 | 9 | 8 | - | 2 | 115 |
| 31 | zurich_city_08_a | 135 | 30 | - | - | - | - | 4 | 169 |
| 32 | zurich_city_09_a | 274 | 11 | 3 | - | - | 4 | 3 | 295 |
| 33 | zurich_city_09_b | 21 | - | 2 | - | - | 10 | - | 33 |
| 34 | zurich_city_09_c | 89 | - | 27 | - | - | - | 3 | 119 |
| 35 | zurich_city_09_d | 264 | - | 12 | 19 | 25 | - | - | 320 |
| 36 | zurich_city_09_e | 45 | - | 4 | 8 | 7 | - | - | 64 |
| 37 | zurich_city_10_a | 181 | - | - | 5 | 1 | - | - | 187 |
| 38 | zurich_city_10_b | 308 | - | 22 | 17 | 9 | 2 | 1 | 359 |
| 39 | zurich_city_11_a | 90 | - | - | 5 | - | - | - | 95 |
| 40 | zurich_city_11_b | 236 | 13 | - | 4 | - | 1 | 9 | 263 |
| 41 | zurich_city_11_c | 412 | - | 5 | 7 | - | - | 10 | 434 |
| 42 | zurich_city_16_a | 61 | 2 | 5 | - | - | - | - | 68 |
| 43 | zurich_city_17_a | 26 | - | - | - | - | - | - | 26 |
| 44 | zurich_city_18_a | 103 | - | 11 | 22 | 3 | - | 2 | 141 |
| 45 | zurich_city_19_a | 159 | 13 | 5 | - | - | - | 2 | 179 |
| 46 | zurich_city_20_a | 85 | 21 | 6 | 8 | - | - | 32 | 152 |
| 47 | zurich_city_21_a | 164 | 10 | - | - | - | - | - | 174 |

**Outcome.** This verification stage ensures that all captions in Talk2Event are high-quality, attribute-aligned, and grounded in the scene context, significantly enhancing the utility of the dataset for training and evaluating language-based grounding models in dynamic environments.

## A.4 Dataset Examples

To illustrate the diversity, richness, and real-world grounding challenges present in the Talk2Event dataset, we present a wide selection of qualitative examples across all seven object categories: Car, Pedestrian, Bus, Truck, Bike, Motorcycle, and Rider. Figures 7 - 15 showcase representative samples with natural language expressions, bounding boxes, and associated scene contexts.

Figure 7 and Figure 8 present grounded examples for the Car class, which is the most prevalent in our dataset. These include a mix of day and night scenes, parked and moving cars, and varied egocentric perspectives (*e.g.*, *"in front of the viewer"*, *"on the left side of the road"*). The descriptions capture

Table 11: Summary of scene statistics from the **test set** of the proposed 🧏Talk2Event dataset.

| # | Sequence | Number of Target Objects | | | | | | | |
| | | Car | Truck | Pedestrian | Bike | Rider | Bus | Motorcycle | All |
|---|---|---|---|---|---|---|---|---|---|
| - | **Summary** **[test]** | **1,699** | **90** | **328** | **157** | **212** | **56** | **13** | **2,555** |
| 01 | interlaken_00_a | 53 | 3 | - | - | - | - | - | 56 |
| 02 | interlaken_00_b | 42 | 3 | - | - | - | - | - | 45 |
| 03 | interlaken_01_a | 146 | 6 | 5 | - | - | - | - | 157 |
| 04 | thun_01_a | 17 | - | - | - | - | - | - | 17 |
| 05 | thun_01_b | 103 | - | 5 | 1 | - | - | - | 109 |
| 06 | thun_02_a | 736 | 19 | 278 | 146 | 206 | 29 | 9 | 1,423 |
| 07 | zurich_city_12_a | 135 | 10 | 4 | - | - | - | 2 | 151 |
| 08 | zurich_city_13_a | 44 | 9 | - | 3 | 1 | - | - | 57 |
| 09 | zurich_city_13_b | 47 | - | - | 2 | 1 | - | - | 50 |
| 10 | zurich_city_14_a | 14 | - | 6 | 2 | - | - | 2 | 24 |
| 11 | zurich_city_14_b | 82 | - | 6 | - | - | 9 | - | 97 |
| 12 | zurich_city_14_c | 84 | 4 | 12 | 1 | 1 | 16 | - | 118 |
| 13 | zurich_city_15_a | 196 | 36 | 12 | 2 | 3 | 2 | - | 251 |

nuanced relations such as "next to a cyclist" or "surrounded by trees", showing the ability to reason about both motion and spatial configuration.

Figure 9 focuses on the Truck category, demonstrating both stationary and moving instances. Descriptions often highlight size (*"large white truck"*), appearance (*"with a distinct white grille"*), and relational cues (*e.g.*, *"beside a black car"* or *"in front of a snow-covered building"*), which are crucial for distinguishing trucks from similarly shaped vehicles like buses.

Figure 10 showcases Bus examples, with expressions emphasizing route information (*e.g.*, *"displaying route number 21"*), positional attributes (*e.g.*, *"stationary at a bus stop"*), and scene context (*e.g.*, *"surrounded by buildings and pedestrians"*). These examples reflect the importance of dynamic attributes such as status and viewer-relative position.

Figure 11 and Figure 12 depict Pedestrian samples, both in daytime and nighttime conditions. Captions in these examples often include detailed clothing descriptions, behavioral cues (*e.g.*, *"carrying shopping bags"*, *"having a conversation"*), and complex spatial references such as *"closer to the viewer"* or *"standing near a crosswalk"*. These examples highlight our support for fine-grained grounding in crowded urban scenes.

Figure 13 shows annotated instances of the Rider class (people on bicycles), which often involve motion and interaction with surrounding traffic. Phrases like *"moving alongside a white car"* or *"approaching a traffic light"* demonstrate the role of temporal reasoning and relational disambiguation.

Figure 14 contains examples of the Bicycle class, most of which are stationary. The language captures fine visual and spatial details such as placement (*"on the sidewalk"*, *"near a street sign"*) and surrounding objects (*"next to a black motorcycle"*), emphasizing appearance and relation-to-others.

Figure 15 presents Motorcycle examples, highlighting static positions relative to the viewer and the environment (*e.g.*, *"near the sidewalk"*, *"surrounded by other vehicles"*). Despite limited motion, the descriptions capture distinct egocentric spatial cues, which are vital for differentiating motorcycles from bikes or scooters in complex scenes.

Across all examples, the referring expressions consistently cover multiple grounding attributes – appearance, status, egocentric relation, and relational context – showcasing our emphasis on attribute-aware and temporally grounded language. These examples demonstrate the practical grounding challenges addressed by Talk2Event and serve as a foundation for developing robust, multimodal vision-language systems in dynamic environments.

## A.5 License

The Talk2Event dataset is released under the Attribution-ShareAlike 4.0 International (CC BY-SA 4.0)[1] license.

---

[1] https://creativecommons.org/licenses/by-sa/4.0/legalcode.

Table 12: The text prompt used for referring expression generation in our work. The output from the model is a structured attribute response followed by a summary sentence.

---

**You are an assistant designed to generate natural language descriptions for objects in dynamic driving scenes.**

You are provided with two consecutive images, taken 400 milliseconds apart, and a bounding box highlighting the same object in both images.

Your task is to generate a detailed and unambiguous referring expression for this object, grounded in the visual and temporal context.

Please describe the following **four aspects**:

- **Appearance**:
  What does the object look like (*e.g.*, class, size, shape, color)?
- **Status**:
  What is the object doing? Is it moving or stationary? In what direction?
- **Relation to Viewer**:
  Where is the object located from the observer's point of view (*e.g.*, to the left, in front, far away)?
- **Relation to Other Objects**:
  Are there nearby objects, and how is this object positioned relative to them?

After describing these four attributes, compose a fluent summary sentence (less than 100 words) that uniquely identifies the object in the scene.

**Response Format:**
```
Appearance: [...]

Status: [...]

Relation to Viewer: [...]

Relation to Other Objects: [...]

Summary: [complete natural language sentence]
```

**Important:**
Ensure the response is unique, informative, and grounded in both the appearance and temporal context of the object.

---

# B Benchmark Construction Details

In this section, we detail the baseline models used in our benchmark. We organize them into three groups based on input modality: frame-only, event-only, and event-frame fusion. For each model, we describe its core design, intended use case, and relevance to the Talk2Event task. All models are adapted for visual grounding by attaching a DETR-style decoder and grounding head. Frame-only models are evaluated directly, while event and fusion models are adapted from detection tasks. Our goal is to provide a fair and comprehensive comparison across input modalities.

Table 13: The text prompt for diversity-aware rewriting of referring expressions.

**You are an assistant tasked with rewriting object descriptions while preserving meaning.**

Given a caption that describes a unique object in a dynamic driving scene, please rewrite it in two different ways. Your rewrites must:

- Refer to the **same object**, using only the visual and contextual details provided in the original.
- Be **linguistically diverse** – vary the phrasing, sentence structure, or attribute emphasis.
- Preserve the **four key attributes**: appearance, motion, relation to viewer, and relation to other objects.

**Example Input:**
"The red motorcycle ahead is turning right next to a white car."

**Example Rewrites:**
(1) Just in front, a red bike makes a right turn beside a white vehicle.
(2) The motorcycle painted in red is moving rightward, positioned next to a white car.

## B.1 Baseline Models

### B.1.1 Frame-Only Baselines

We benchmark several state-of-the-art frame-based grounding models, including both trained and zero-shot methods.

- MDETR [41]: A transformer-based grounding model trained on large-scale image-text datasets. MDETR jointly reasons over visual and textual inputs and predicts bounding boxes that correspond to referring phrases. It serves as a representative example of supervised multimodal grounding.

- BUTD-DETR [37]: Combines bottom-up region proposals with a DETR-style architecture to improve sample efficiency and open-vocabulary alignment. This model serves as the initialization backbone for our EventRefer framework.

- OWL [67]: A zero-shot object detection model that matches visual regions to text queries using contrastive vision-language pretraining. OWL supports open-vocabulary grounding without task-specific training.

- OWL-v2 [66]: An updated version of OWL with improved localization and multi-scale detection capabilities. It offers stronger performance in zero-shot grounding and supports few-shot adaptation.

- YOLO-World [16]: A real-time open-vocabulary detector that integrates language embeddings into the YOLO architecture. It demonstrates strong grounding performance under computational constraints.

- GroundingDINO [58]: A highly effective open-set detector that aligns language and vision using global-text conditioning and region proposals. It is one of the strongest generalist models in open-vocabulary grounding.

These baselines demonstrate the limits of frame-only grounding when applied to dynamic scenes. While effective for appearance-based descriptions, they struggle with temporally grounded attributes like motion or trajectory.

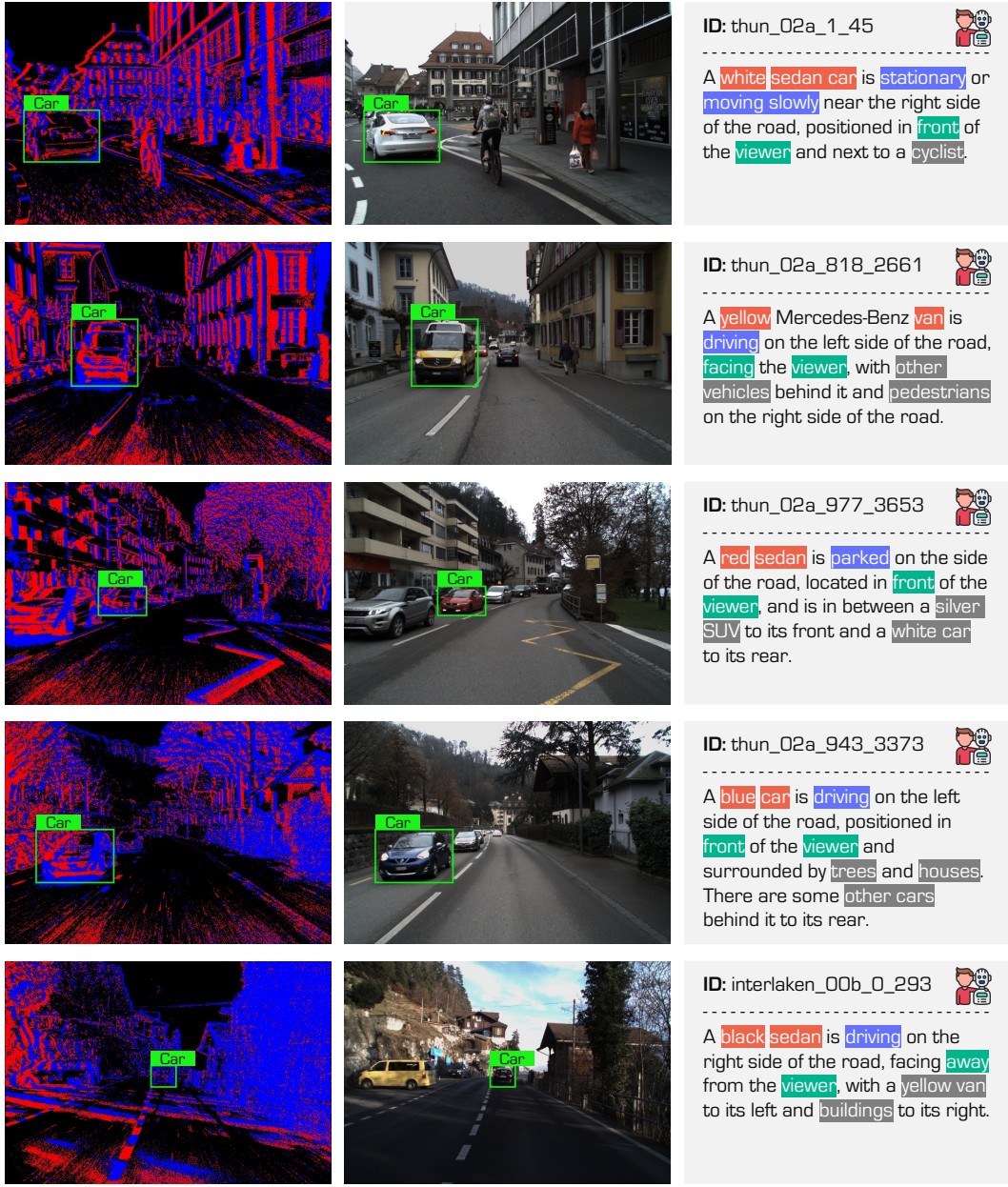

Figure 7: **Additional examples** of the `Car` class in the 🎭 Talk2Event dataset. The data from left to right are: the event stream (left column), the frame (middle column), and the referring expression (right column). Best viewed in colors and zoomed in for details.

### B.1.2 Event-Only Baselines

Due to the lack of existing event-based grounding methods, we adapt event-based object detectors by attaching a DETR-based grounding decoder. These models operate solely on event voxel grids.

- RVT [30]: A spatiotemporal transformer model for event-based detection. It captures long-term dependencies in voxelized event sequences and serves as a strong backbone for event-based perception.

- LEOD [92]: A lightweight transformer architecture optimized for efficient event-based detection under computational constraints.

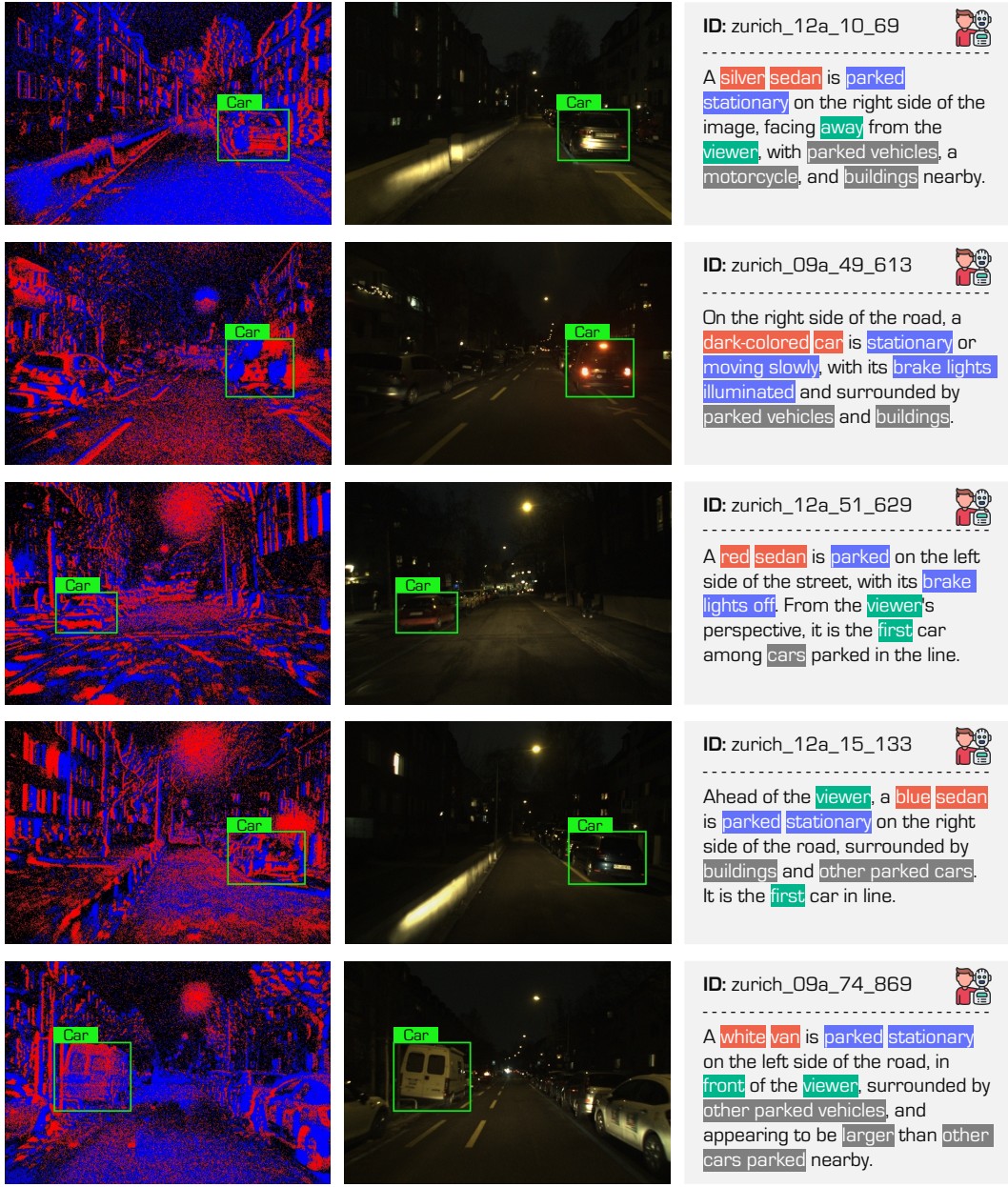

Figure 8: **Additional examples** of the `Car` class (in nighttime condition) in the 👥 Talk2Event dataset. The data from left to right are: the event stream (left column), the frame (middle column), and the referring expression (right column). Best viewed in colors and zoomed in for details.

- SAST [72]: Proposes a streaming attention framework tailored to asynchronous event data. It maintains causal context across frames, which is particularly useful for scenes with frequent motion.

- SSMS [118]: Uses spiking neural network dynamics for temporally precise detection. It offers biologically inspired mechanisms for capturing motion patterns in event streams.

- EvRT-DETR [83]: A DETR-style event-based transformer with tokenized event embeddings. It provides strong baseline performance and serves as a direct structural match to our grounding head.

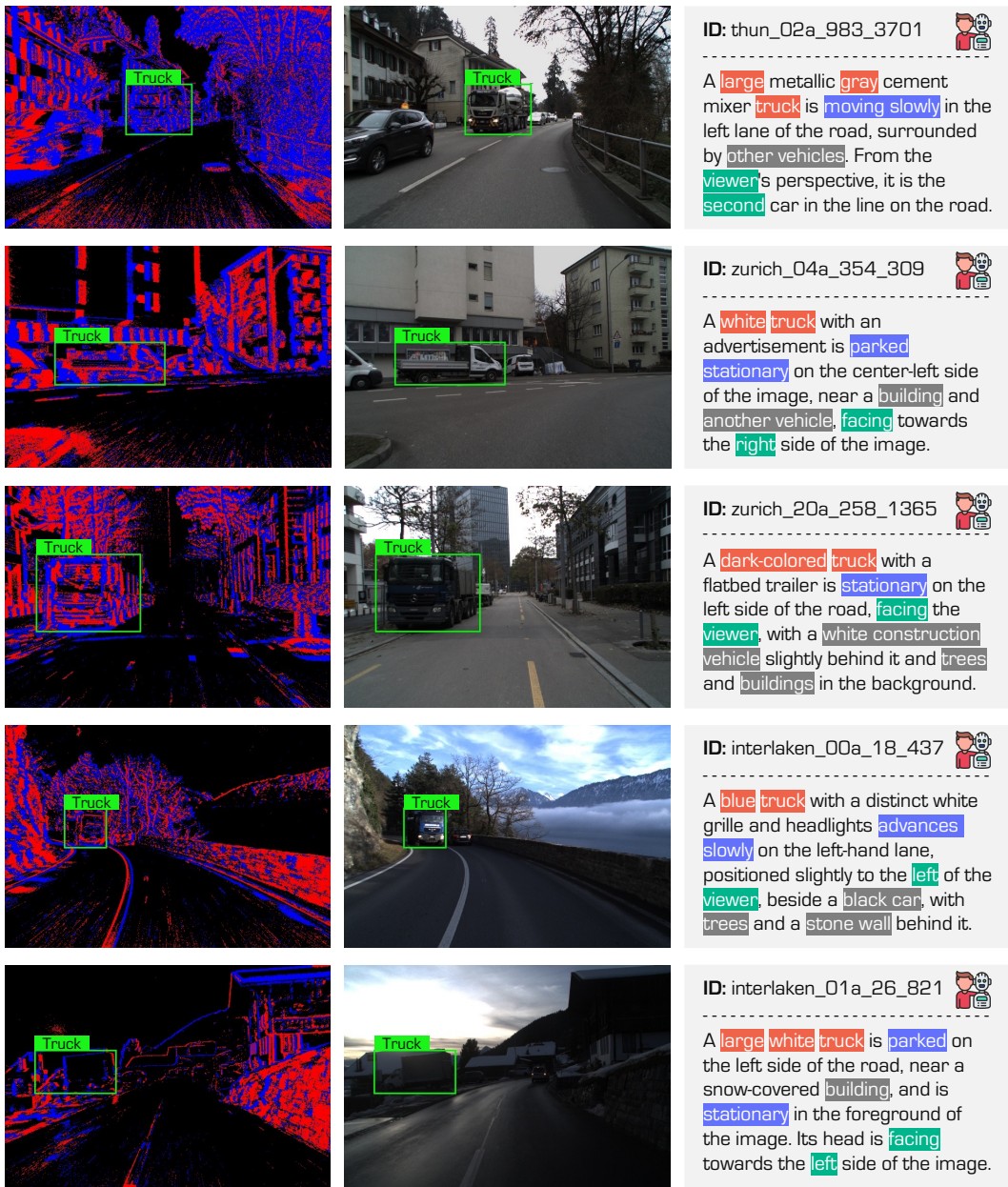

Figure 9: **Additional examples** of the Truck class in the 🧑 Talk2Event dataset. The data from left to right are: the event stream (left column), the frame (middle column), and the referring expression (right column). Best viewed in colors and zoomed in for details.

These baselines reflect the strong potential of event representations for motion-sensitive perception but are not trained for grounding, leading to limited language alignment and poor disambiguation in cluttered scenes.

### B.1.3   Event-Frame Fusion Baselines

To explore cross-modal grounding, we extend event-based detection models by incorporating a frame encoder. The event and frame features are fused before the grounding head.

- RENet [111]: A dual-stream architecture that combines RGB and event streams via residual feature fusion. We adapt it for grounding by aligning both modalities at the feature level.

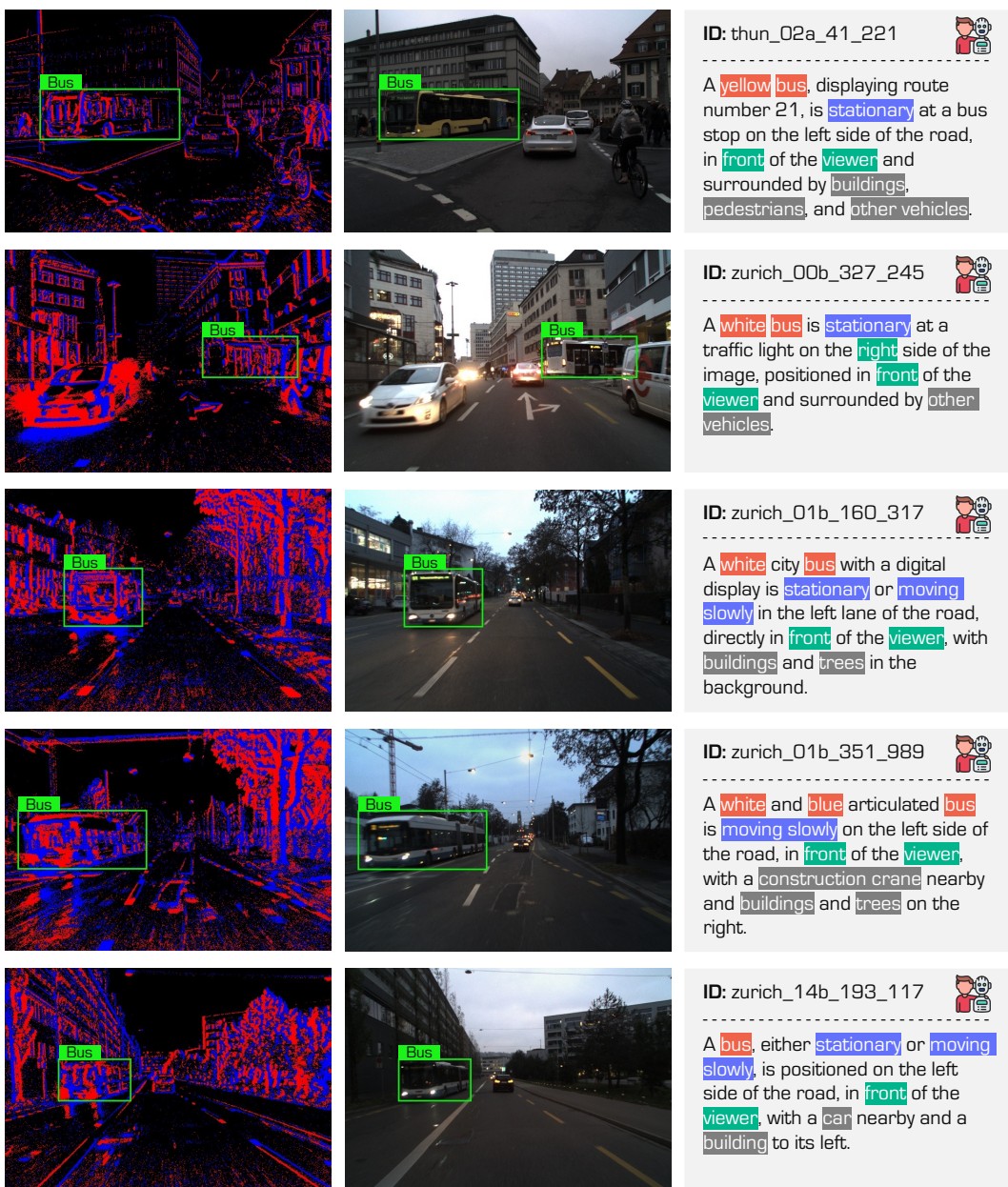

Figure 10: **Additional examples** of the Bus class in the 👥 Talk2Event dataset. The data from left to right are: the event stream (left column), the frame (middle column), and the referring expression (right column). Best viewed in colors and zoomed in for details.

- CAFR [8]: Fuses event and frame data using attention mechanisms and cross-modal transformers.
- DAGr [27]: Leverages dense attention between event- and frame-based queries for joint perception. It serves as a strong baseline for spatial-temporal fusion and alignment.
- FlexEvent [63]: Adapts the transformer encoder to event-specific sparsity while integrating image features through lightweight gating. Its modular fusion design supports flexible deployment across domains.

These methods highlight the benefits of fusing event and frame modalities. While originally designed for detection, we extend them to grounding and show that combining high-temporal resolution events with rich appearance cues improves both precision and robustness.

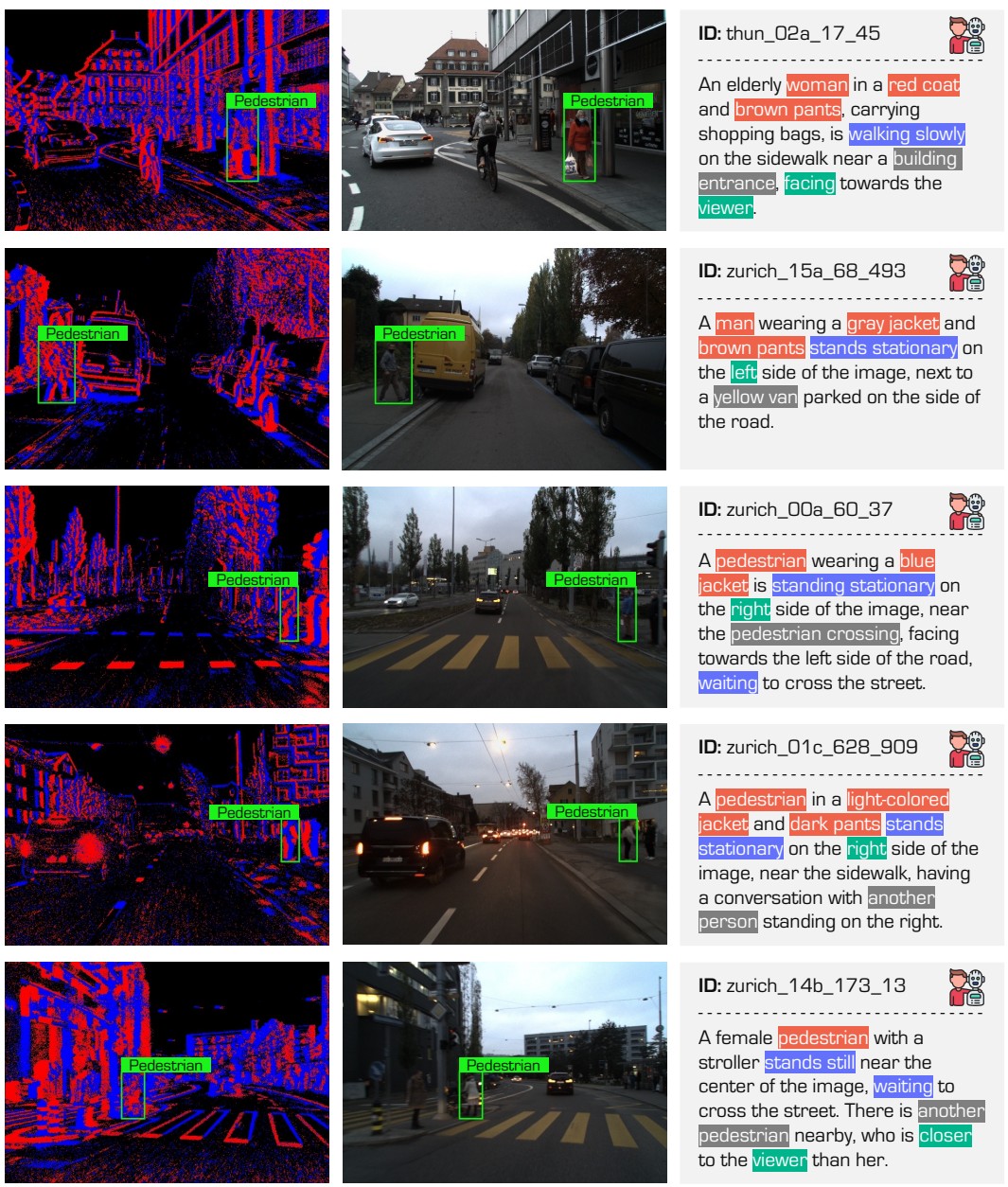

Figure 11: **Additional examples** of the `Pedestrian` class in the 👥 Talk2Event dataset. The data from left to right are: the event stream (left column), the frame (middle column), and the referring expression (right column). Best viewed in colors and zoomed in for details.

## B.2 Implementation Details

All models in our benchmark, including EventRefer and the baselines, are implemented using PyTorch [71] and trained on NVIDIA RTX A6000 GPUs.

**Tokenizer and Text Encoder.** We use the RoBERTa-base [61] tokenizer and encoder for all methods that process natural language inputs. The tokenizer operates with left padding and truncation up to a maximum of 64 tokens. The encoder outputs are 768-dimensional, followed by a linear projection to match the 256-dimensional visual token space. We fine-tune the text encoder for all non-zero-shot methods with a learning rate of $5 \times 10^{-6}$.

**Frame Backbone.** For frame-only and fusion models, we adopt ResNet-101 [34] as the frame encoder. It is initialized from ImageNet [18] pretrained weights. We extract multi-scale features from

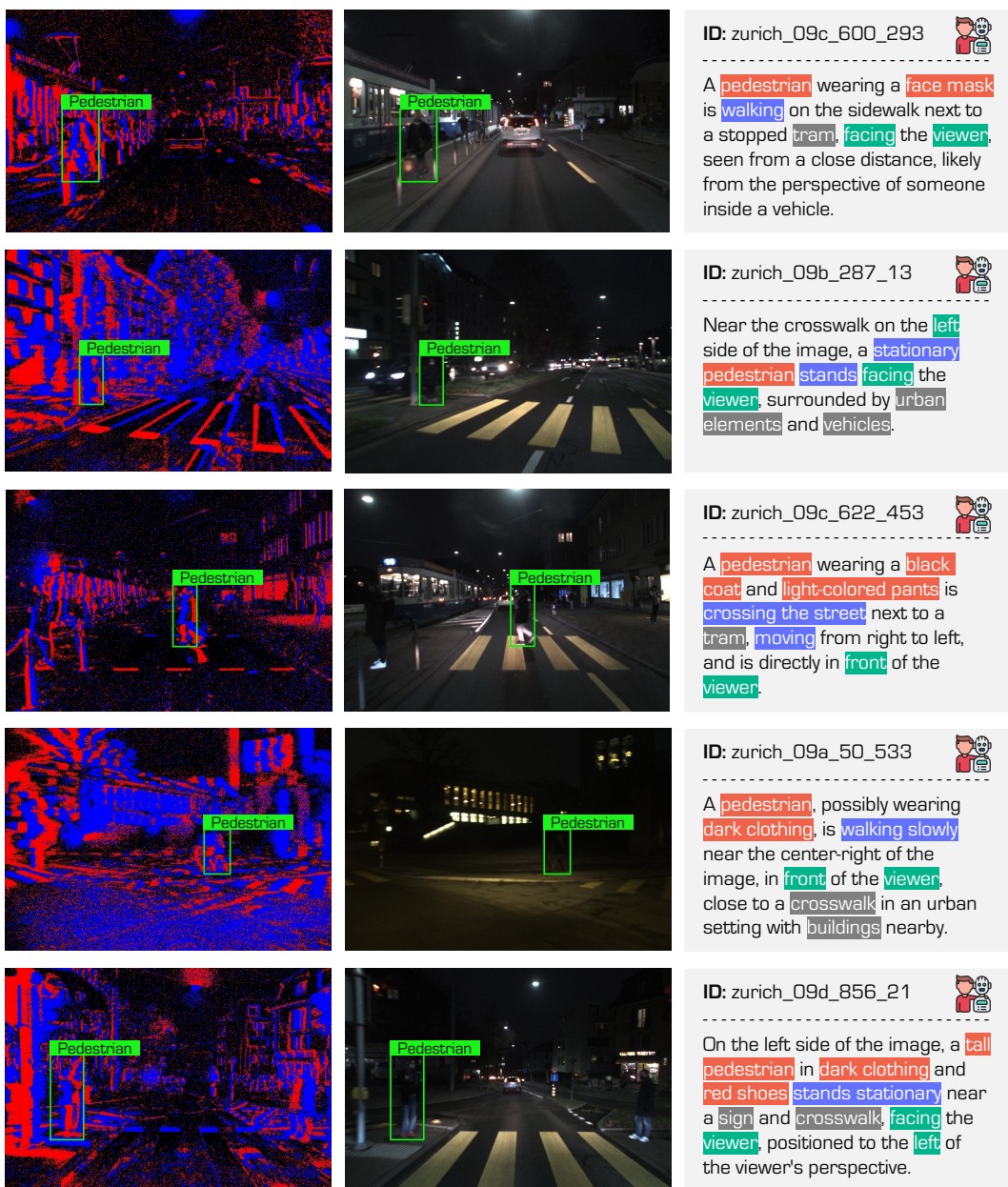

Figure 12: **Additional examples** of `Pedestrian` class (in nighttime condition) in the 🧑‍🤝‍🧑 Talk2Event dataset. The data from left to right are: the event stream (left column), the frame (middle column), and the referring expression (right column). Best viewed in colors and zoomed in for details.

layers `res3`, `res4`, and `res5`, each of which is passed through a $1 \times 1$ convolution and flattened into token embeddings of 256 dimensions. These features are concatenated before being passed to the Transformer encoder.

**Event Backbone.** For event-only and fusion models, we use the transformer-based encoder from FlexEvent [63], which operates on the voxelized event grid $\mathbf{E} \in \mathbb{R}^{2 \times T \times H \times W}$. We use $T=9$ temporal bins and downsample spatial resolution to $H=128$, $W=256$. The voxel grid is projected into 256-dimensional tokens through a 3D convolutional stem and grouped with sinusoidal temporal embeddings. Event tokens are fused through a stack of 6 transformer layers with causal attention.

**Fusion Design.** For all fusion baselines, we concatenate the event and frame tokens before feeding them into the DETR-style transformer encoder [9]. In EventRefer, we adopt the MoEE design

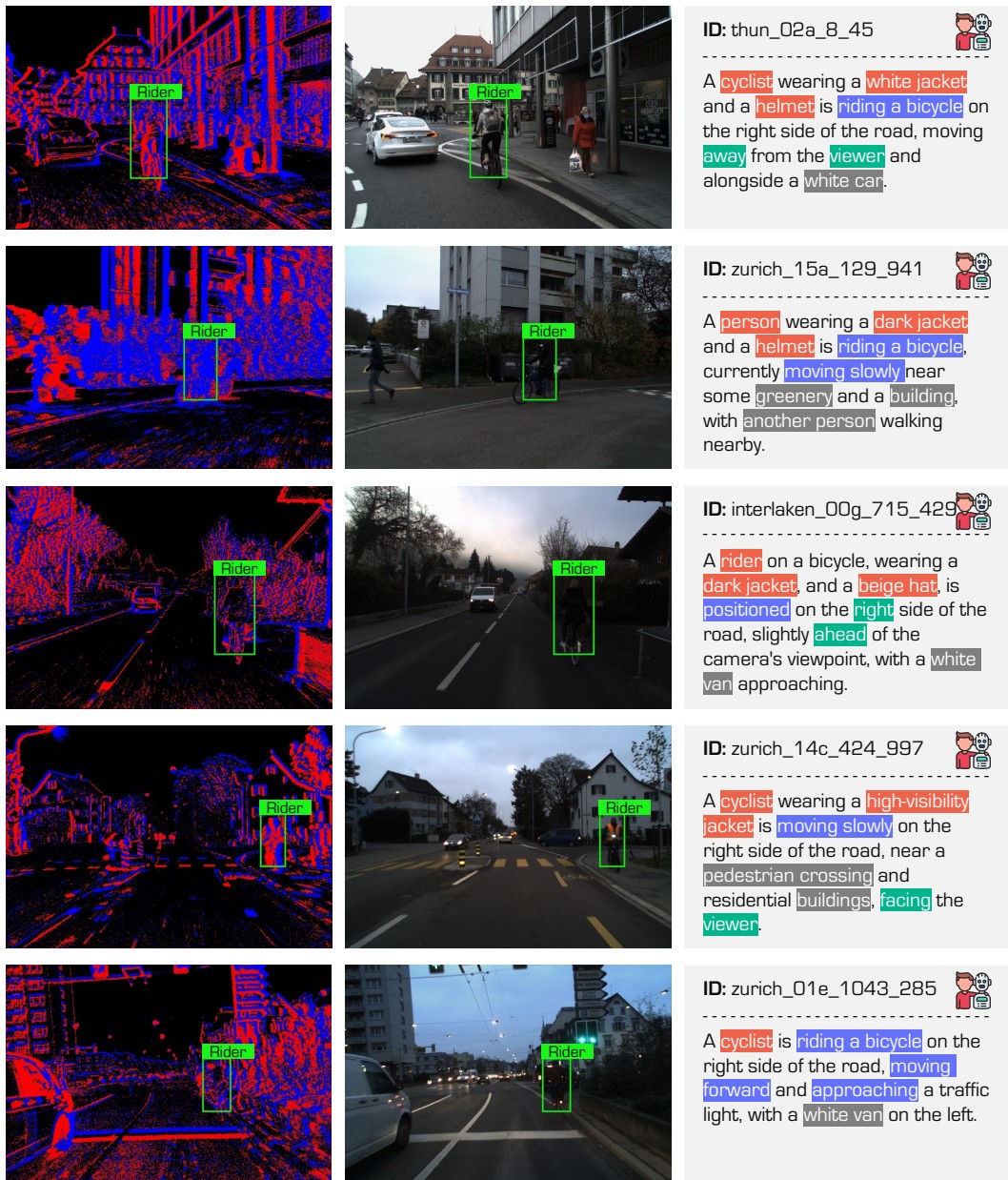

Figure 13: **Additional examples** of the `Rider` class in the 🎫 Talk2Event dataset. The data from left to right are: the event stream (left column), the frame (middle column), and the referring expression (right column). Best viewed in colors and zoomed in for details.

detailed in the main paper, while other fusion baselines use either additive fusion, channel-wise concatenation, or attention-based token mixing. For the simple baseline "RVT+ResNet+Attention", we concatenate frame and event tokens followed by a two-layer attention module.

**DETR Transformer and Decoder.** All models use a shared DETR transformer architecture for language–vision alignment and grounding. The encoder consists of 6 layers with 8 heads each. The decoder uses 100 object queries and 6 cross-attention layers. The transformer weights are initialized from the public BUTD-DETR [37] checkpoint and jointly trained during grounding.

**Training Configuration.** We use AdamW [62] as the optimizer with a weight decay of 0.01. Learning rates are as follows:

- $1 \times 10^{-6}$ for the frame backbone,

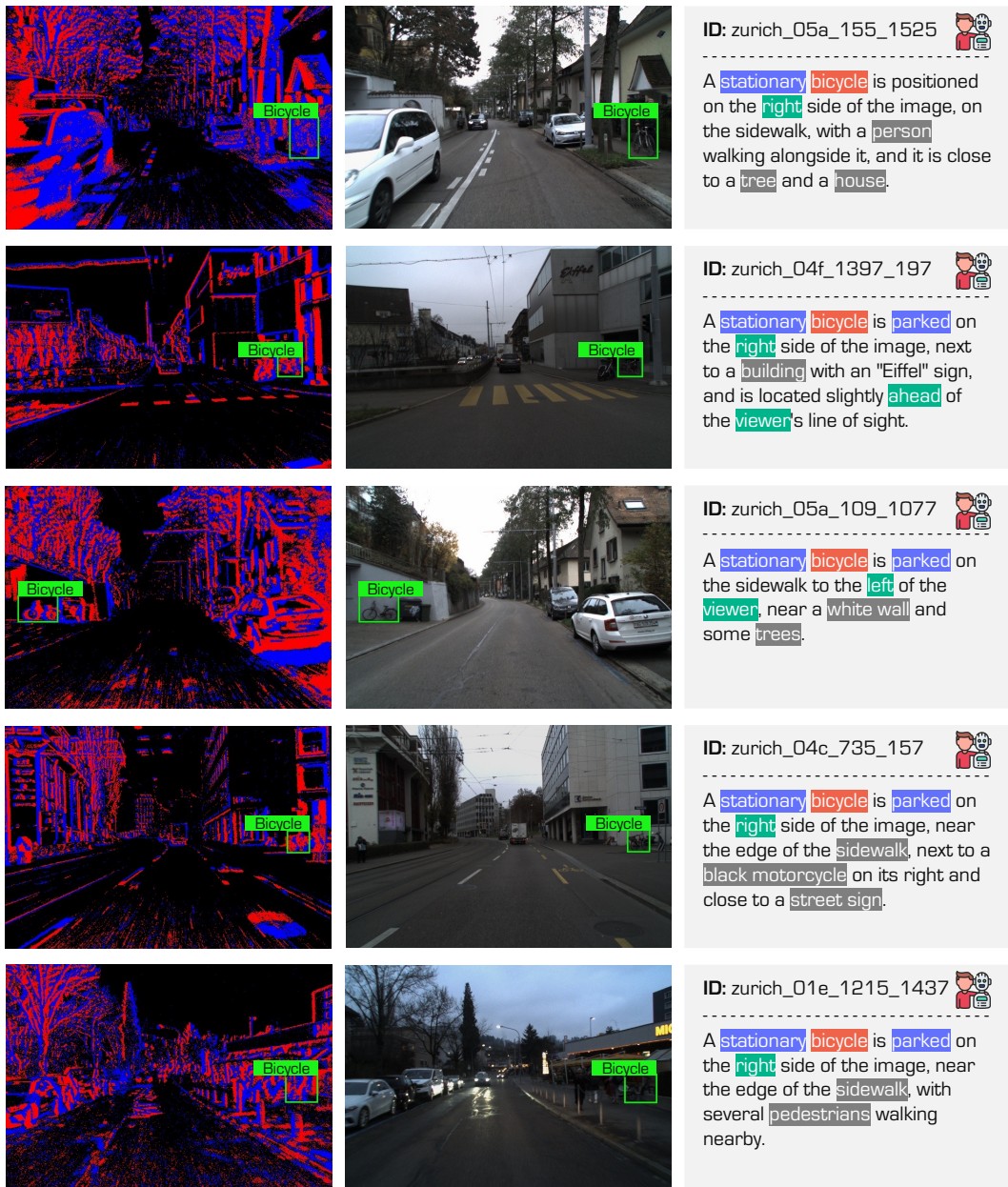

Figure 14: **Additional examples** of the `Bicycle` class in the 👥 Talk2Event dataset. The data from left to right are: the event stream (left column), the frame (middle column), and the referring expression (right column). Best viewed in colors and zoomed in for details.

- $5 \times 10^{-6}$ for the RoBERTa text encoder,
- $5 \times 10^{-5}$ for the event backbone, fusion module, and transformer layers.

All models are trained with a batch size of 16 and warm-up for the first 500 steps, followed by cosine decay. Training is conducted for 90K steps on the training split of Talk2Event, and models are selected using the best mAcc on a held-out validation split.

**Loss Function.** We adopt the same grounding loss $\mathcal{L}_{\text{ground}}$ for all methods, consisting of an $\ell_1$ regression loss and GIoU loss for bounding boxes, and a cross-entropy loss for token alignment.

EventRefer uses four attribute-specific token maps for dense supervision, while all baselines are trained with only class-name token supervision, following their original configurations.

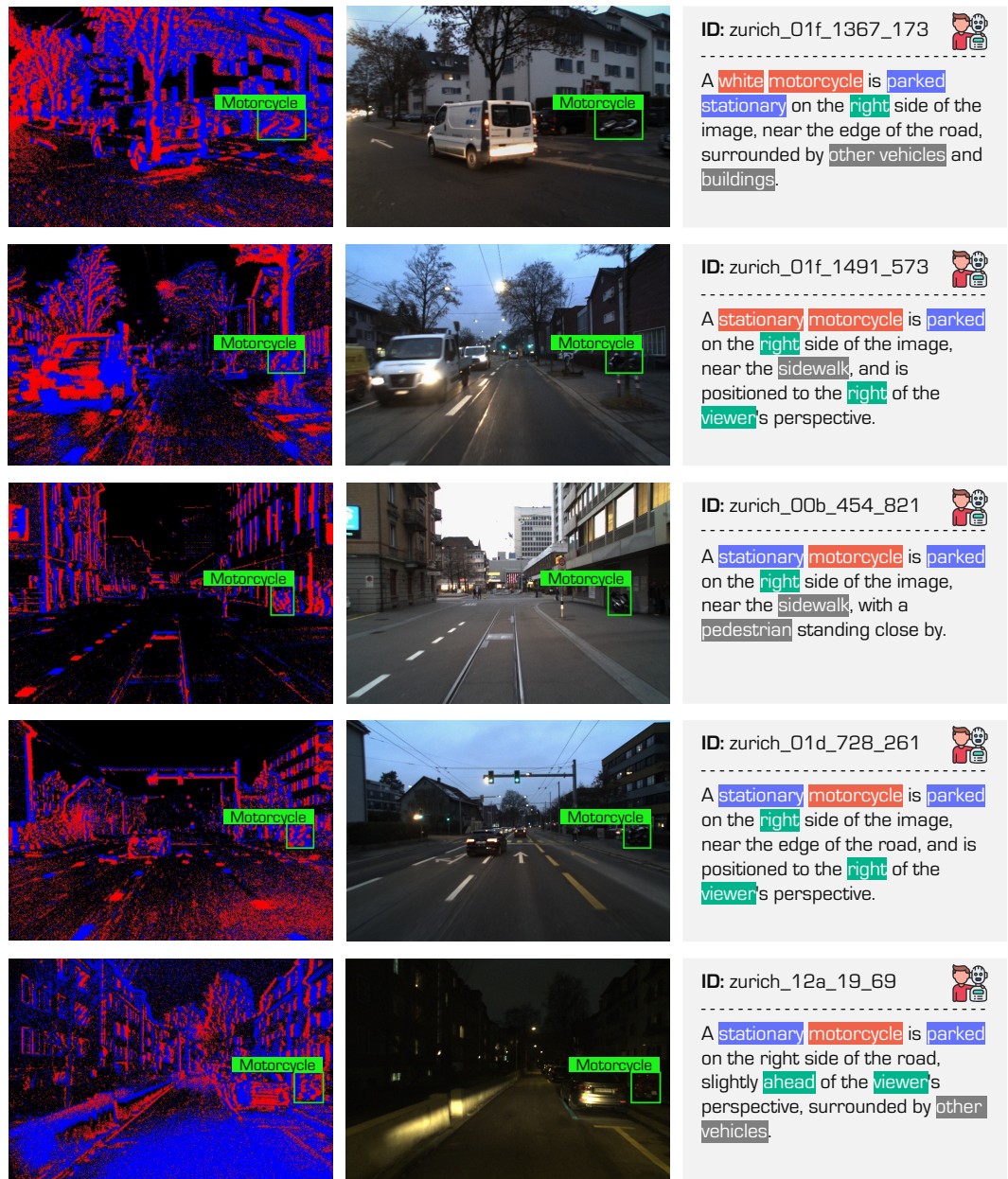

Figure 15: **Additional examples** of the `Motorcycle` class in the 🎮 Talk2Event dataset. The data from left to right are: the event stream (left column), the frame (middle column), and the referring expression (right column). Best viewed in colors and zoomed in for details.

## B.3 Evaluation Metrics

To comprehensively assess model performance in language-driven object localization from event camera data, we adopt two standard evaluation metrics used in prior grounding literature:

**Top-1 Accuracy (`Top-1 Acc`).** This metric computes the proportion of samples for which the predicted bounding box overlaps the ground-truth bounding box with an Intersection-over-Union (IoU) greater than or equal to a specified threshold. Formally, for a predicted box $\hat{\mathbf{b}}$ and a ground-truth box $\mathbf{b}$, the sample is considered correct if:

$$\text{IoU}(\hat{\mathbf{b}}, \mathbf{b}) > \theta \,, \tag{4}$$

where $\theta$ is the IoU threshold. We follow recent trends in high-precision grounding and set $\theta = 0.95$, a stringent threshold that emphasizes precise spatial alignment and penalizes loosely overlapping predictions. This high threshold is particularly relevant for dynamic driving scenes where bounding boxes may shift rapidly due to motion or occlusion.

**Mean IoU (`mIoU`).** This metric averages the IoU between each predicted box $\hat{\mathbf{b}}$ and the corresponding ground-truth box $\mathbf{b}$ across all samples. It serves as a complementary, continuous-valued metric that provides insight into the overall localization quality beyond binary thresholds. Formally,

$$\text{mIoU} = \frac{1}{N} \sum_{i=1}^{N} \text{IoU}(\hat{\mathbf{b}}_i, \mathbf{b}_i) , \tag{5}$$

where $N$ is the total number of samples.

Together, `Top-1 Acc.` at `IoU@0.95` and `mIoU` allow us to evaluate both precision-critical scenarios and general localization fidelity.

### B.4 Evaluation Protocol

To ensure fair and reproducible comparisons across methods, we standardize the evaluation pipeline as follows.

**Test-Time Configuration.** During inference, each model receives the input event voxel grid $\mathbf{E}$ (and optionally the synchronized frame $\mathbf{F}$) along with the referring expression $\mathcal{S}$. The model outputs a set of candidate bounding boxes $\hat{\mathbf{b}}_n$ and token distributions $\hat{\mathbf{m}}_n$ for each query $n$.

**Scoring and Selection.** Each candidate query is scored against the target attribute token distributions. In EventRefer, the final box prediction is selected as:

$$\hat{\mathbf{b}} = \arg\max_{\hat{\mathbf{b}}_n} < \text{softmax}(\hat{\mathbf{m}}_n), \text{softmax}(\mathbf{m}_i) > , \tag{6}$$

where the score reflects alignment with the most informative attribute for the given scene. For baseline methods, which are only trained with class-name tokens, the scoring uses the softmax similarity against a binary indicator map for the class token.

**Scene-Level Aggregation.** Metrics are computed per sample and then averaged over all test scenes. We further break down performance by object class and modality setting (event-only, frame-only, fusion) to analyze robustness under different conditions.

**Reproducibility.** All models are evaluated using the same test split of Talk2Event. The test split contains 1,134 unique scenes and 3,137 objects, covering all 7 traffic-related categories with varying density and occlusion levels. Evaluation code and scripts will be released alongside the dataset to support standardized benchmarking.

### B.5 Quantifying Event Response Strength

To analyze how different levels of event activity influence the reliance on specific attribute experts, we introduce a quantitative metric called **event response strength**, which captures the density and spread of event signals across the input volume.

**Definition.** Given an input event voxel grid $\mathbf{E} \in \mathbb{R}^{2 \times T \times H \times W}$, where the first channel dimension corresponds to the event polarity ($\pm 1$), we define the *event response strength* of a scene as the total number of non-zero spatial-temporal positions:

$$\text{Strength}(\mathbf{E}) = \sum_{p=1}^{2} \sum_{\tau=1}^{T} \sum_{x=1}^{H} \sum_{y=1}^{W} \mathbf{1} \left[ \mathbf{E}(p, \tau, x, y) > 0 \right] . \tag{7}$$

This count measures how many voxels are activated in the event tensor, thus reflecting the spatial extent and temporal frequency of events observed during the 200ms window centered at $t_0$.

**Normalization and Binning.** To analyze trends across the full dataset, we normalize the raw response strengths using min-max normalization over all test samples:

$$\text{NormalizedStrength}(s) = \frac{s - s_{\min}}{s_{\max} - s_{\min}} , \tag{8}$$

where $s$ is the raw strength value of a sample, and $s_{\min}$ and $s_{\max}$ are the minimum and maximum values observed across the test set.

The normalized values are then discretized into **seven bins** of equal range:

$$\mathcal{B}_i = \left[ \frac{i-1}{7}, \frac{i}{7} \right), \quad \text{for } i = 1, \dots, 7 \, . \tag{9}$$

Each bin groups samples of similar event activity levels, from low (Bin #1) to high (Bin #7) response strength.

**Usage in Analysis.** We analyze the top-1 and top-2 activated attribute experts across each bin to study how the model adapts its reliance on appearance, motion, and relational reasoning under varying event signal intensity. This allows us to reveal behavior such as:

- Appearance and viewer-centric cues dominate under weak event response (low motion).

- Status and relational cues emerge under stronger responses (high dynamics).

This stratified analysis sheds light on the input-adaptive design of our MoEE module and validates the attribute-aware grounding strategy under real-world dynamic conditions.

# C    Additional Experimental Results

In this section, we provide further analyses and evaluations to complement the main results presented in the paper. We begin by presenting additional quantitative comparisons across semantic classes and varying scene complexities to validate the generality and robustness of our method. We then offer detailed qualitative results that highlight the grounding behavior of EventRefer in diverse scenarios, followed by a discussion of common failure cases. These insights help contextualize the strengths and limitations of our framework and benchmark.

## C.1    Additional Quantitative Results

### C.1.1    Class-Wise mIoU Analysis

Table 14 presents a breakdown of class-wise mean Intersection-over-Union (mIoU) across various grounding methods on the validation set of Talk2Event. We group the results by input modality – *frame-only*, *event-only*, and *event-frame fusion* – to facilitate comparison.

In the **Frame-Only** setting, our method outperforms all baselines, achieving 85.76% mIoU overall. Compared to BUTD-DETR [37] (84.30%), we observe consistent improvements across all classes, particularly on small and dynamic categories like Pedestrian (+2.6%) and Rider (+8.1%). This suggests that the attribute-aware formulation in EventRefer enhances spatial precision and interpretability even in appearance-driven modalities.

In the **Event-Only** group, EventRefer again leads with 76.46% mIoU, outperforming EvRT-DETR [83] (75.66%) and other strong detectors. Notably, it achieves the highest class-wise scores for Rider (85.03%), Truck (76.35%), and Motorcycle (78.17%). These results indicate the benefits of explicitly modeling motion- and relation-centric attributes when operating on high-frequency, appearance-sparse event streams.

In the **Event-Frame Fusion** setting, EventRefer achieves the highest overall mIoU of 87.32%, surpassing all fusion baselines including DAGr [27] (86.90%) and FlexEvent [63] (86.83%). Our method exhibits strong gains for challenging classes such as Rider (+1.94% over DAGr) and Motorcycle (+6.2% over DAGr), highlighting the effectiveness of our Mixture of Event-Attribute Experts (MoEE) in fusing multi-attribute features under varying scene dynamics.

Overall, this analysis confirms that EventRefer achieves superior localization performance by incorporating structured attribute reasoning, consistently improving grounding quality across modalities and semantic categories.

Table 14: **Comparisons among state-of-the-art methods** on the *val* set of the 👥Talk2Event dataset. The results are the **mIoU scores** with respect to different **semantic classes**. The methods are grouped based on input modalities. All scores are given in percentage (%).

| Method | mIoU | Ped | Rider | Car | Bus | Truck | Bike | Motor |
|---|---|---|---|---|---|---|---|---|
| **• Modality: Frame Only** | | | | | | | | |
| MDETR [41] | 80.09 | 70.20 | 75.90 | 83.09 | 79.50 | 80.52 | 74.52 | 73.50 |
| BUTD-DETR [37] | 84.30 | **75.37** | 81.39 | 86.39 | 89.80 | 83.34 | 82.47 | 88.81 |
| OWL [67] | 69.89 | 55.30 | 62.00 | 75.20 | 89.54 | 67.76 | 46.28 | 88.49 |
| OWL-v2 [66] | 72.81 | 53.12 | 65.57 | 75.84 | 86.89 | **84.38** | 77.91 | 90.17 |
| YOLO-World [16] | 59.76 | 45.45 | 53.49 | 61.37 | 89.40 | 68.85 | 61.87 | 85.13 |
| GroundingDINO [58] | 68.67 | 58.46 | 50.83 | 73.81 | 82.45 | 72.38 | 48.01 | **94.01** |
| 👥 EventRefer | **85.76** | 73.97 | **89.53** | **87.44** | **92.70** | 84.30 | **85.84** | 81.19 |
| **• Modality: Event Only** | | | | | | | | |
| RVT† [30] | 74.52 | 64.80 | 72.10 | 78.05 | 65.88 | 62.79 | 70.62 | 63.44 |
| LEOD† [92] | 74.37 | 65.15 | 74.68 | 77.24 | 64.40 | 65.67 | 70.63 | 74.60 |
| SAST† [72] | 74.94 | 65.52 | 74.90 | 77.95 | 64.82 | 65.39 | 71.59 | 70.94 |
| SSMS† [118] | 75.14 | 65.83 | 73.06 | 78.55 | 65.75 | 63.54 | 71.25 | 65.29 |
| EvRT-DETR† [83] | 75.66 | **65.86** | 75.07 | **79.09** | 64.38 | 63.10 | 72.20 | 61.31 |
| 👥 EventRefer | **76.46** | 62.13 | **85.03** | 78.36 | **74.44** | **76.35** | **74.86** | **78.17** |
| **• Modality: Event-Frame Fusion** | | | | | | | | |
| RVT‡ [30] | 86.64 | **79.53** | 85.23 | 88.62 | 89.37 | 80.53 | 84.98 | 81.56 |
| RENet‡ [111] | 87.02 | 78.94 | 86.89 | 88.97 | 91.72 | 81.62 | 84.30 | 87.99 |
| CAFR‡ [8] | 86.13 | 76.26 | 84.96 | 88.22 | 91.92 | 82.23 | 85.47 | **92.28** |
| DAGr‡ [27] | 86.90 | 78.24 | 85.39 | 88.90 | 87.20 | **87.28** | 85.65 | 80.92 |
| FlexEvent‡ [63] | 86.83 | 76.03 | 85.43 | 89.07 | 92.29 | 84.50 | **86.12** | 82.07 |
| 👥 EventRefer | **87.32** | 77.42 | **87.83** | **89.32** | **92.70** | 83.69 | 85.71 | 88.45 |

### C.1.2 Top-1 Accuracy *vs.* Scene Complexity

Table 15 examines model performance with respect to *scene complexity*, measured by the number of objects per frame. As the object count increases, the visual grounding task becomes significantly more challenging due to increased ambiguity and visual clutter. This analysis allows us to assess the robustness of each model in complex multi-object scenarios.

In the **Frame-Only** setting, EventRefer consistently outperforms prior models, achieving a peak Top-1 accuracy of 80.97% in single-object scenes and maintaining strong performance as complexity increases. Notably, our method delivers the highest score in dense scenes with > 9 objects (45.45%), significantly outperforming GroundingDINO [58] (22.58%) and BUTD-DETR [37] (15.15%). This demonstrates the advantage of attribute-aware representations in disambiguating targets amidst visual and semantic overlap.

In the **Event-Only** setting, although overall performance is lower due to the lack of appearance cues, EventRefer maintains the strongest performance among all event-based baselines, particularly in dense scenes. For instance, its accuracy remains stable across 4–6-object scenes (23–24%), whereas all other methods fall below 15%. This highlights our model's ability to leverage motion and relational cues to resolve ambiguity in appearance-sparse environments.

In the **Event-Frame Fusion** group, EventRefer again leads across almost all complexity levels. It achieves 85.07% accuracy in single-object scenes and maintains above 50% even in scenes with 5 or more objects – surpassing CAFR [8], DAGr [27], and FlexEvent [63]. Interestingly, although a slight dip is observed in the > 9 category (to 31.03%), the overall trend confirms that our attribute-aware fusion design (MoEE) enables more robust disambiguation across increasing scene clutter.

Together, these results demonstrate that EventRefer scales effectively across varying scene complexities, reinforcing the importance of compositional attribute modeling in open-world event-based grounding.

Table 15: **Comparisons among state-of-the-art methods** on the *val* set of the 🧑‍💻Talk2Event dataset. The results are the **top-1 Acc scores** with respect to the **number of objects per scene**. The methods are grouped based on input modalities. All scores are given in percentage (%).

| Method (# Scenes) | Single (338) | 2 (519) | 3 (446) | 4 (365) | 5 (334) | 6 (218) | 7 (237) | 8 (64) | 9 (23) | > 9 (11) |
|---|---|---|---|---|---|---|---|---|---|---|
| • **Modality: Frame Only** | | | | | | | | | | |
| MDETR [41] | 70.12 | 51.77 | 39.61 | 31.69 | 30.54 | 22.48 | 20.68 | 19.27 | 15.94 | 9.09 |
| BUTD-DETR [37] | 77.51 | 61.40 | 53.14 | 40.73 | 36.93 | 30.89 | 28.97 | 28.65 | 17.39 | 15.15 |
| OWL [67] | 69.43 | 52.34 | 40.73 | 29.86 | 30.54 | 26.61 | 24.05 | 17.19 | 14.49 | 27.27 |
| OWL-v2 [66] | 65.38 | 57.29 | 46.86 | 35.89 | 33.63 | 23.85 | 27.29 | 23.44 | 20.29 | 18.18 |
| YOLO-World [16] | 55.13 | 38.72 | 32.78 | 28.70 | 28.57 | 21.63 | 27.55 | 27.81 | 19.35 | 22.58 |
| GroundingDINO [58] | 81.32 | 57.17 | 44.08 | 32.81 | 31.71 | 26.49 | 26.11 | 25.13 | 22.58 | 22.58 |
| 🧑‍💻Talk2Event | 80.97 | 64.48 | 60.46 | 46.67 | 43.21 | 44.34 | 37.97 | 42.19 | 26.09 | 45.45 |
| • **Modality: Event Only** | | | | | | | | | | |
| RVT[†] [30] | 52.47 | 39.50 | 26.08 | 19.73 | 11.48 | 7.65 | 13.22 | 16.15 | 13.04 | 9.09 |
| LEOD[†] [92] | 45.66 | 38.86 | 25.64 | 18.17 | 14.27 | 8.87 | 9.28 | 8.85 | 10.14 | 9.09 |
| SAST[†] [72] | 50.00 | 39.88 | 27.95 | 20.37 | 14.37 | 10.09 | 11.67 | 8.85 | 11.59 | 12.12 |
| SSMS[†] [118] | 54.54 | 42.58 | 29.82 | 20.46 | 13.17 | 8.26 | 13.64 | 15.62 | 11.59 | 9.09 |
| EvRT-DETR[†] [83] | 55.33 | 43.61 | 31.02 | 21.64 | 14.97 | 9.63 | 14.49 | 14.58 | 13.04 | 12.12 |
| 🧑‍💻Talk2Event | 62.62 | 36.67 | 28.92 | 23.56 | 23.75 | 24.16 | 21.38 | 17.19 | 17.39 | 18.18 |
| • **Modality: Event-Frame Fusion** | | | | | | | | | | |
| RVT[‡] [30] | 83.43 | 68.34 | 58.37 | 49.22 | 46.11 | 40.98 | 38.68 | 42.19 | 33.33 | 36.36 |
| RENet[‡] [111] | 82.15 | 68.66 | 58.89 | 46.76 | 44.81 | 42.35 | 39.66 | 44.27 | 33.33 | 39.39 |
| CAFR[‡] [8] | 85.40 | 68.34 | 58.22 | 47.49 | 45.21 | 47.71 | 43.18 | 47.40 | 31.88 | 39.39 |
| DAGr[‡] [27] | 84.38 | 70.46 | 59.77 | 50.79 | 46.01 | 45.35 | 37.79 | 42.62 | 38.24 | 41.38 |
| FlexEvent[‡] [63] | 84.97 | 73.68 | 62.52 | 50.80 | 47.62 | 42.95 | 37.16 | 47.59 | 32.26 | 35.48 |
| 🧑‍💻Talk2Event | 85.07 | 75.05 | 65.36 | 52.91 | 51.43 | 48.72 | 40.44 | 43.17 | 36.76 | 31.03 |

### C.1.3 mIoU *vs.* Scene Complexity

Table 16 evaluates how grounding precision varies with scene complexity, measured by the number of objects per frame. Here, we report the average IoU between predicted and ground-truth boxes for correctly grounded objects, offering a more fine-grained view of spatial accuracy compared to Top-1 accuracy.

In the **Frame-Only** group, EventRefer consistently outperforms all baselines across almost every bin. It achieves a peak of $95.20\%$ in single-object scenes and remains strong even in highly complex settings ($> 9$ objects), with $89.10\%$ – notably higher than the next-best baseline BUTD-DETR ($81.74\%$). This indicates that our attribute-aware formulation not only improves grounding recall but also enhances localization precision in crowded scenarios.

In the **Event-Only** group, EventRefer achieves $91.22\%$ in simple scenes and maintains higher mIoU than all other event-based methods across most object-count bins. For instance, in mid-complexity scenes with 5–6 objects, our method scores $71.33\%$ and $75.43\%$, respectively, whereas the strongest alternative (EvRT-DETR) scores $67.59\%$ and $70.29\%$. This shows that our model benefits from structured reasoning over motion and relational cues even without appearance information.

In the **Event-Frame Fusion** setting, where models have access to both modalities, EventRefer maintains top-tier performance across the board. It reaches $95.96\%$ in single-object scenes and achieves $86.47\%$ even in the $> 9$ category – on par with or exceeding other strong baselines like CAFR ($90.45\%$) and FlexEvent ($89.31\%$). Although some competing methods have slight gains in specific bins (*e.g.*, FlexEvent on single-object), our method exhibits consistently high performance with minimal drop-off as scene complexity increases.

Together with the Top-1 accuracy results in Table 15, these findings confirm that EventRefer delivers strong, stable localization even in cluttered scenes, validating the benefits of our attribute-aware grounding design across both recognition and localization axes.

Table 16: **Comparisons among state-of-the-art methods** on the *val* set of the 🦾Talk2Event dataset. The results are the **mIoU scores** with respect to the **number of objects per scene**. The methods are grouped based on input modalities. All scores are given in percentage (%).

| Method
(# Scenes) | Single
(338) | 2
(519) | 3
(446) | 4
(365) | 5
(334) | 6
(218) | 7
(237) | 8
(64) | 9
(23) | > 9
(11) |
|---|---|---|---|---|---|---|---|---|---|---|
| **• Modality: Frame Only** | | | | | | | | | | |
| MDETR [41] | 93.51 | 84.77 | 78.90 | 73.66 | 76.70 | 77.56 | 73.53 | 72.23 | 67.16 | 76.51 |
| BUTD-DETR [37] | 94.75 | 87.40 | 85.06 | 80.48 | 80.03 | 82.09 | 78.33 | 78.30 | 69.06 | 81.74 |
| OWL [67] | 89.90 | 76.17 | 67.53 | 63.30 | 63.99 | 62.63 | 60.57 | 61.21 | 60.83 | 66.82 |
| OWL-v2 [66] | 84.21 | 78.41 | 71.13 | 68.09 | 69.12 | 68.38 | 66.72 | 67.35 | 65.20 | 62.69 |
| YOLO-World [16] | 74.82 | 62.41 | 57.20 | 55.95 | 54.88 | 53.65 | 56.58 | 61.01 | 53.97 | 46.59 |
| GroundingDINO [58] | 90.77 | 74.05 | 67.76 | 60.77 | 60.11 | 58.94 | 62.96 | 61.22 | 65.61 | 65.70 |
| 🦾Talk2Event | 95.20 | 88.96 | 85.09 | 82.15 | 81.20 | 84.28 | 81.90 | 83.28 | 70.38 | 89.10 |
| **• Modality: Event Only** | | | | | | | | | | |
| RVT$^\dagger$ [30] | 88.46 | 84.19 | 72.89 | 62.23 | 64.33 | 58.57 | 57.70 | 75.69 | 55.71 | 58.35 |
| LEOD$^\dagger$ [92] | 88.73 | 80.80 | 74.54 | 68.98 | 68.07 | 68.50 | 64.26 | 70.41 | 65.88 | 67.16 |
| SAST$^\dagger$ [72] | 88.93 | 80.75 | 75.67 | 69.90 | 68.41 | 69.07 | 65.46 | 71.10 | 66.77 | 67.17 |
| SSMS$^\dagger$ [118] | 88.92 | 82.26 | 75.34 | 70.38 | 66.95 | 68.68 | 65.50 | 73.97 | 67.32 | 72.43 |
| EvRT-DETR$^\dagger$ [83] | 88.66 | 81.92 | 76.30 | 71.43 | 67.59 | 70.29 | 66.66 | 73.07 | 67.05 | 73.45 |
| 🦾Talk2Event | 91.22 | 81.15 | 73.62 | 70.61 | 71.33 | 75.43 | 70.05 | 70.96 | 67.81 | 75.07 |
| **• Modality: Event-Frame Fusion** | | | | | | | | | | |
| RVT$^\ddagger$ [30] | 95.87 | 89.37 | 86.22 | 83.06 | 82.71 | 83.95 | 83.92 | 81.69 | 77.52 | 89.47 |
| RENet$^\ddagger$ [111] | 95.51 | 90.12 | 86.16 | 83.61 | 82.68 | 84.49 | 84.85 | 84.08 | 79.39 | 89.33 |
| CAFR$^\ddagger$ [8] | 95.91 | 89.40 | 86.51 | 82.14 | 81.46 | 84.74 | 81.00 | 80.54 | 72.29 | 90.45 |
| DAGr$^\ddagger$ [27] | 95.58 | 90.18 | 86.83 | 82.98 | 82.53 | 85.03 | 83.43 | 81.37 | 78.36 | 84.94 |
| FlexEvent$^\ddagger$ [63] | 96.19 | 90.50 | 87.55 | 81.75 | 82.70 | 84.08 | 81.84 | 84.42 | 75.88 | 89.31 |
| 🦾Talk2Event | 95.96 | 90.71 | 87.05 | 83.44 | 83.04 | 85.33 | 83.51 | 84.14 | 75.90 | 86.47 |

## C.2 Additional Qualitative Results

Figure 16 and Figure 17 showcase qualitative examples from the validation set of Talk2Event, highlighting the predictions of our proposed EventRefer across diverse scenes and object types.

Our model successfully grounds complex, attribute-rich descriptions involving appearance, motion, and relational cues. For example, in Row #1 of Figure 16, the pedestrian in a blue jacket is correctly localized among nearby distractors, showing that EventRefer can distinguish subtle visual traits and relative positions. In Row #2, despite the target car being close to the image boundary and partially occluded, the model leverages spatial relations (*e.g.*, *"right side of the road"*, *"surrounded by other vehicles"*) to ground the object accurately.

These results confirm that EventRefer produces robust, semantically aligned predictions in varied environmental and linguistic conditions, leveraging the strengths of both event and frame modalities.

## C.3 Failure Cases and Analyses

Despite its overall strong performance, EventRefer exhibits some failure cases, primarily arising in scenes with high visual clutter or overlapping object instances. In some examples, the model might misalign the bounding box due to under-attending to relational cues – *e.g.*, predicting a nearby pedestrian when the expression specifies *"next to the cyclist"*.

Failure modes are more common when:

- Objects have similar appearance (*e.g.*, multiple dark-clothed pedestrians).
- Viewpoint or occlusion reduces visibility of key attributes.
- Expressions emphasize abstract relations or implicit motion (*"approaching an intersection"* without clear directionality).

Additionally, performance slightly degrades in extreme low-light scenes, where even event data may exhibit sparse activations. Nevertheless, qualitative observations suggest that failure often results from

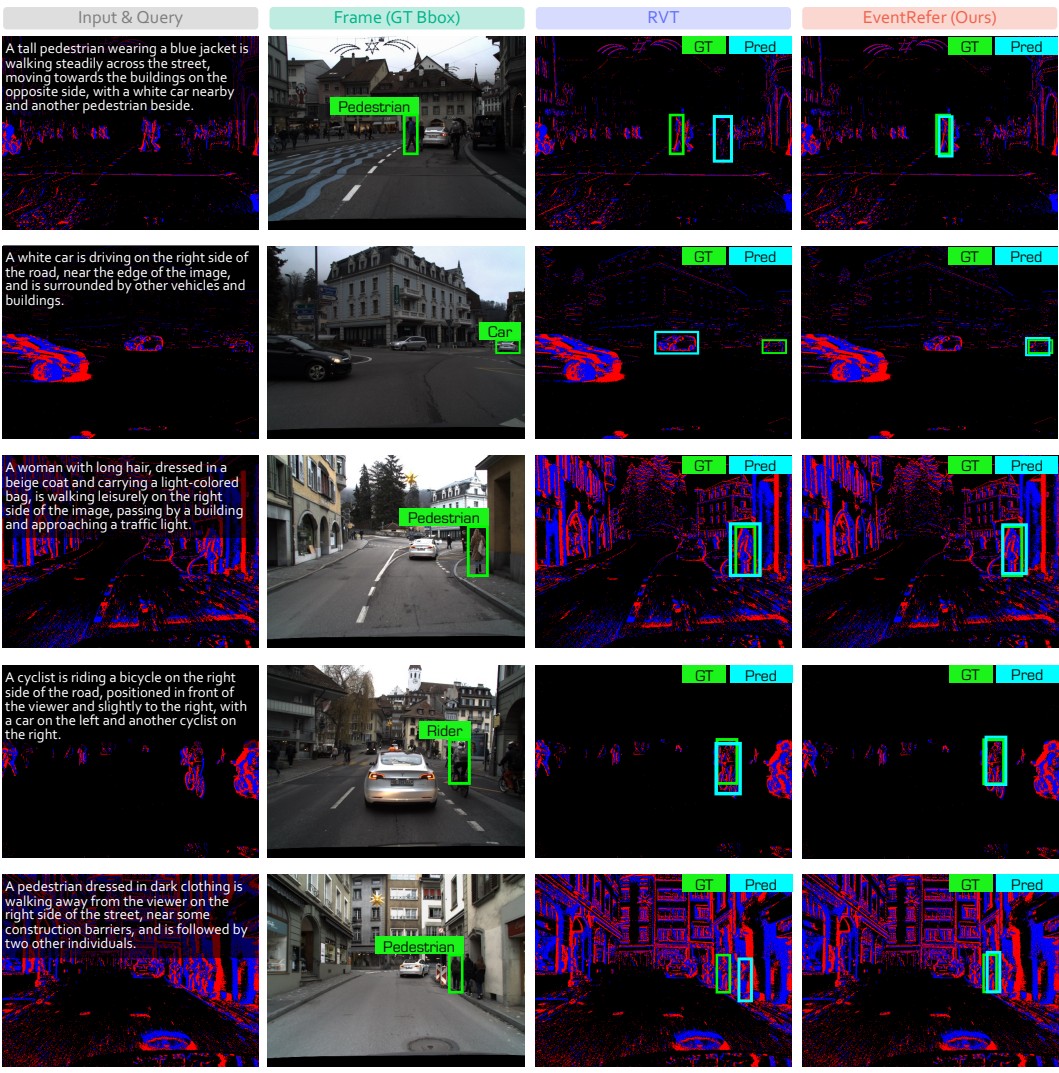

Figure 16: **Additional qualitative results (1/2)** of method tested on the 🧑‍🤝‍🧑 Talk2Event dataset. The ground truth and predicted boxes are denoted in green and blue colors, respectively.

partial grounding, where the prediction is semantically plausible but falls short of full localization accuracy. These cases highlight potential areas for improvement, such as stronger modeling of inter-object relations, leveraging 3D context, or using temporally longer event sequences. We leave these extensions as future work to build on the strong baseline established by EventRefer.

# D    Broader Impact & Limitations

In this section, we elaborate on the broader impact, societal implications, and potential limitations of the proposed Talk2Event dataset and framework.

## D.1    Broader Impact

Talk2Event introduces a new benchmark and methodology for grounded understanding of dynamic scenes using event cameras. By bridging asynchronous vision with natural language, our dataset has the potential to facilitate safer and more intelligent decision-making in robotics and autonomous systems, particularly under challenging conditions such as fast motion, low light, or occlusion. We believe that integrating language grounding into event-based perception opens new directions for explainable and interactive AI in domains such as assistive robotics, smart city infrastructure,

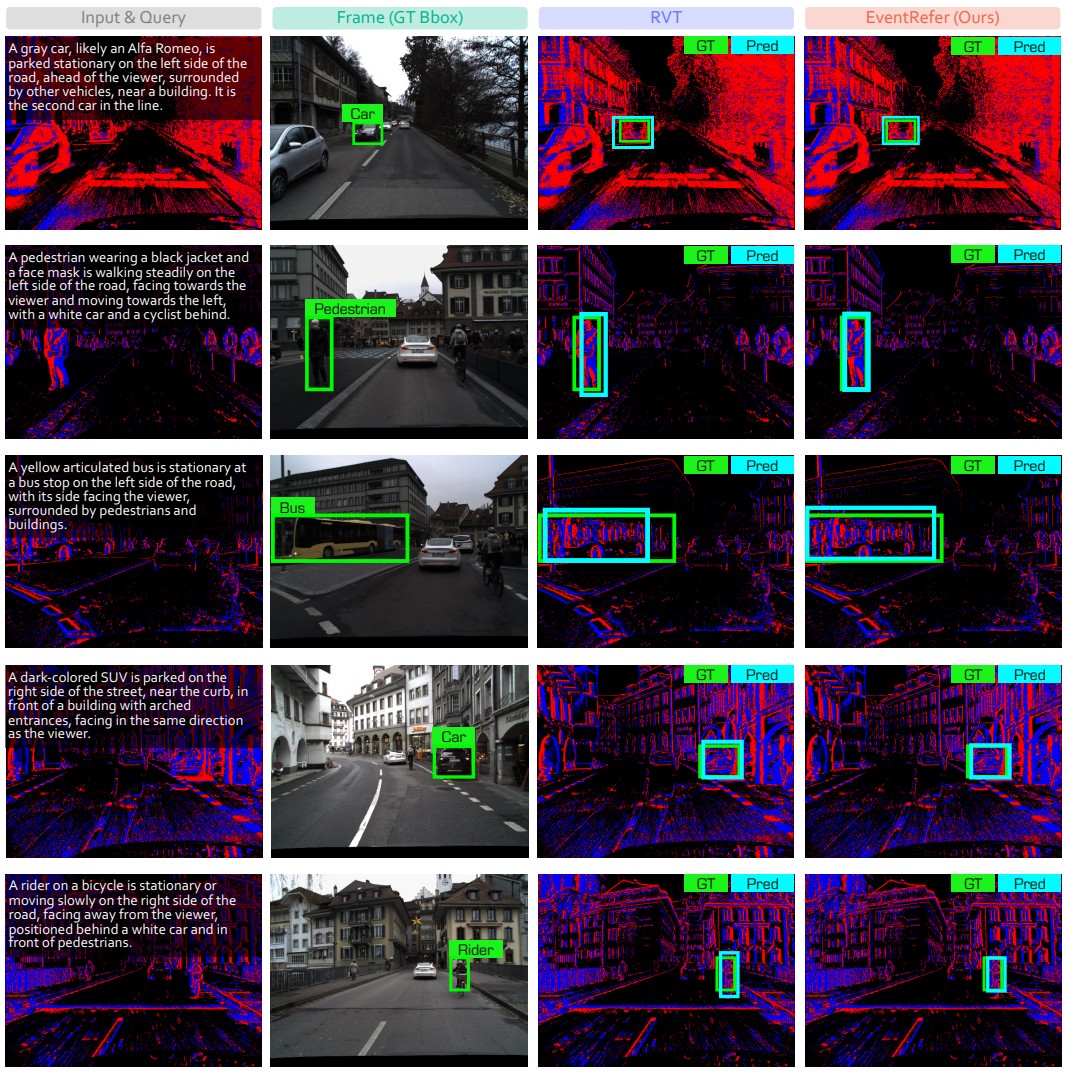

Figure 17: **Additional qualitative results (2/2)** of method tested on the Talk2Event dataset. The ground truth and predicted boxes are denoted in green and blue colors, respectively.

and autonomous navigation. Additionally, our multi-attribute formulation encourages interpretable reasoning and lays the foundation for more transparent visual-language understanding systems.

## D.2  Societal Influence

Our work contributes to the advancement of vision-language research by offering an open benchmark that emphasizes efficiency, temporal sensitivity, and multimodal reasoning. Event cameras, due to their low power consumption and robustness, offer compelling advantages for sustainable AI deployment. By enabling grounded scene understanding in this domain, we hope to stimulate future research that benefits both academic and industrial communities. That said, as with any dataset involving real-world driving scenes, there remains a responsibility to ensure that these tools are used ethically, especially in safety-critical applications. We emphasize that our dataset contains no biometric or personally identifiable data and has been carefully curated to focus on object-centric annotations.

### D.3 Potential Limitations

Despite its novelty and strengths, Talk2Event has limitations. First, while the dataset captures a wide range of urban scenarios, it is currently limited to driving scenes from a specific region and camera setup, which might introduce bias and limit generalization to other geographic or environmental settings. Second, the language annotations, though human-verified, are generated with the help of vision-language models and might reflect model-specific biases. Third, our current focus is on visual grounding from projected event volumes, and does not yet extend to 3D or spatiotemporal grounding in world coordinates. Lastly, the framework assumes synchronized RGB frames, which might not always be available in pure event-driven systems. Future work could explore grounding directly from events alone, or leverage additional sensing modalities such as depth or LiDAR.

## E  Public Resource Used

In this section, we acknowledge the use of the public resources, during the course of this work:

### E.1  Public Datasets Used

- DSEC[2] . . . . . . . . . . . . . . . . . . . . . . . . . . . . . . . . . . . . . . . . . . . . . . . . . . . . . . . . . CC BY-SA 4.0 License

### E.2  Public Implementation Used

- BUTD-DETR[3] . . . . . . . . . . . . . . . . . . . . . . . . . . . . . . . . . . . . . . . . . . . . . . . . . CC BY-SA 4.0 License
- RVT[4] . . . . . . . . . . . . . . . . . . . . . . . . . . . . . . . . . . . . . . . . . . . . . . . . . . . . . . . . . . . . . . MIT License
- SAST[5] . . . . . . . . . . . . . . . . . . . . . . . . . . . . . . . . . . . . . . . . . . . . . . . . . . . . . . . . . . . . . MIT License
- SSMS[6] . . . . . . . . . . . . . . . . . . . . . . . . . . . . . . . . . . . . . . . . . . . . . . . . . . . . . . . . . . . . . . Unknown
- DAGr[7] . . . . . . . . . . . . . . . . . . . . . . . . . . . . . . . . . . . . . . . . . . . GNU General Public 3.0 License
- FlexEvent[8] . . . . . . . . . . . . . . . . . . . . . . . . . . . . . . . . . . . . . . . . . . . . . . . . . . . . . . . MIT License
- OWL[9] . . . . . . . . . . . . . . . . . . . . . . . . . . . . . . . . . . . . . . . . . . . . . . . . . . . . . . . Apache 2.0 License
- YOLO-World[10] . . . . . . . . . . . . . . . . . . . . . . . . . . . . . . . . . . . GNU General Public 3.0 License
- GroundingDINO[11] . . . . . . . . . . . . . . . . . . . . . . . . . . . . . . . . . . . . . . . . . . . . . . . Apache 2.0 License

---

[2]https://dsec.ifi.uzh.ch.
[3]https://github.com/nickgkan/butd_detr.
[4]https://github.com/uzh-rpg/RVT.
[5]https://github.com/Peterande/SAST.
[6]https://github.com/uzh-rpg/ssms_event_cameras.
[7]https://github.com/uzh-rpg/dagr.
[8]https://github.com/DylanOrange/flexevent.
[9]https://github.com/google-research/scenic/tree/main/scenic/projects/owl_vit.
[10]https://github.com/AILab-CVC/YOLO-World.
[11]https://github.com/IDEA-Research/GroundingDINO.

