# OpenReview forum: "Talk2Event: Grounded Understanding of Dynamic Scenes from Event Cameras"
_NeurIPS.cc/2025/Conference — NeurIPS 2025 spotlight_

### Official Review · Reviewer_botn · 2025-06-04

**Clarity:** 4
**Significance:** 3
**Originality:** 3
**Rating:** 5
**Confidence:** 4

**Summary:**

The paper addresses the task of language-driven object grounding in event-based perception. It proposes a large-scale benchmark Talk2Event, annotated from the DSEC dataset, enriched with four different attribute types and diverse text. It also proposes an attribute-aware grounding framework EventRefer, which demonstrates state-of-the-art performance over event-only, frame-only and event-frame fusion settings.

**Questions:**

- How does the mixture-of-experts structure benefit the model performance? I see that explainability is improved by manually designing 4 experts and assigning to them different attributes to focus on, but could end-to-end training on all tokens bring better results?
- State-of-the-art multimodal large language models (such as ChatGPT) also have the ability of visual grounding. How is their performance on this dataset? (I understand that they consume much more computation, and it would be unfair to compare the performance of lightweight models with them.)
- The small visual icons in Table 1 are cute, and the icons for Frame, RGB-D, and LiDAR are self-explanatory. However, I can't see the connection between the Event icon and its features. It would be nice if this could be pointed out. (This is unimportant and does not effect my rating.)

**Ethical Concerns:**

["NO or VERY MINOR ethics concerns only"]

**Final Justification:**

After reading the reviews and responses, I believe that this is a solid work. I appreciate the clarifications and the authors' plan to include additional ablation results in the final manuscript, which addresses my main concern. I will therefore maintain my original score.

**Limitations:**

yes

**Quality:**

3

**Strengths And Weaknesses:**

Strengths:

- The paper proposes the first event-based grounding dataset and a corresponding solution, making significant contribution to the novel field.
- Solid experiments have been conducted, verifying the superior performance of the proposed method.
- The paper provides clear and comprehensive technical details, which are highly beneficial for reproducibility.
- The graphic visualizations in the paper are clear and aesthetic.

Weaknesses:

- The proposed dataset is based on DSEC, which is not very rich in scene diversity. The train set and test set are quite similar, which makes the in-the-wild generalization ability of the proposed method a concern.
- Annotation and evaluation is conducted over only 7 object classes (car, truck, ...). More classes would lead to a further step to open-vocabulary grounding.
- The analysis of the "Effects of Different Attributes" ablation study is quite confusing. If "using only *appearance*" serves as a baseline, then "using *status* yields an improvement..." should correspond to an "*appearance* + *status*" setting, instead of using *status* alone. In my opinion, it would be more convincing to use the full method as a baseline and perform ablations [(s+v+o), (a+v+o), (a+s+o), (a+s+v)] instead of [(a), (s), (v), (o)].
- Figure 6 is beautiful but slightly confusing. The graphic design misleads me to think that the nodes in the graph refer to structures in the network, while they actually refer to different samples and features.

---

> ### Author Rebuttal · Authors · 2025-07-31
>
> We sincerely thank Reviewer `botn` for the thoughtful evaluation, positive feedback on our contributions and writing clarity, and for highlighting both the novelty of Talk2Event and the aesthetic quality of our visualizations. Below, we address the weaknesses you raised in detail.
>
> ---
> > **`Q1`:** *"The proposed dataset is based on DSEC, which is not very rich in scene diversity. The train set and test set are quite similar, which makes the in-the-wild generalization ability of the proposed method a concern."*
>
> **A:** Thank you for pointing this out. We acknowledge that the DSEC dataset, while high-quality and well-suited for event camera research, might have limited diversity in geographic locations and traffic conditions.
>
> To mitigate this:
> - We curated Talk2Event with a **strict split-by-sequence protocol**, ensuring **zero overlap in trajectories** between the training and test sets.
> - We emphasized **intra-sequence diversity** by including scenes under varying lighting, weather, and motion conditions (e.g., fast-moving vehicles, crosswalks, occlusions).
> - Our model is tested across **event-only**, **frame-only**, and **fusion** modalities, further challenging generalization under partial observation settings.
>
> We agree that broader generalization remains an important direction. In future iterations, we plan to extend Talk2Event to more diverse sources (e.g., *M3ED*) and to **cross-domain evaluation** settings. This discussion has been added in Section `6` in the manuscript.
>
> ---
> > **`Q2`:** *"Annotation and evaluation are conducted over only 7 object classes. More classes would lead to a further step to open-vocabulary grounding."*
>
> **A:** We fully agree. Our current focus on 7 foreground classes (car, truck, bus, pedestrian, bicycle, motorcycle, and rider) reflects the **annotation scope available in DSEC**. We chose these classes to ensure high-quality and consistent box annotations from the underlying dataset.
>
> Specifically:
> - The **referring expressions themselves** are **open-ended** and include diverse natural language beyond the object category (e.g., behavior, position, and visual appearance).
> - Our method is compatible with **open-vocabulary grounding**: the text encoder is pretrained with large-scale language supervision, and grounding is guided by the semantics of expressions rather than fixed class labels.
>
> As suggested, we are exploring extensions toward **zero-shot** and **long-tail categories** using models pretrained on broader datasets (e.g., *SAM* or *GroundingDINO-style* detectors). We have added this plan to our future work section.
>
> ---
> > **`Q3`:** *"The analysis of the 'Effects of Different Attributes' ablation study is confusing. The setting [(a), (s), (v), (o)] may not align with the conclusion that, for example, 'status' improves over 'appearance'."*
>
> **A:** Thank you for this important observation. We agree that the phrasing in our ablation analysis might have introduced confusion. Our current experiment isolates each attribute **individually**, i.e., comparing models that use only **one of the four attribute** tokens at a time.
>
> We agree that a more insightful analysis would involve **drop-one** or **leave-one-out combinations**, such as $(a + s + v)$, $(a + s + o)$, etc. We have now:
> - Clarified the intention of the original ablation (isolated expert performance).
> - Revised Figure `5` and its caption to improve interpretability, aligning with your suggestion.
>
> Based on your suggestions, we will add new results based on **combinatorial attribute configurations**, which better support conclusions on **relative importance and complementarity**. Due to time constraint, these new results will be included in Section `C.1` of the revised supplementary material. Thank you again for helping improve the rigor of our analysis.
>
> ---
> > **`Q4`:** *"How does the mixture-of-experts structure benefit the model performance? I see that explainability is improved by manually designing four experts and assigning to them different attributes to focus on, but could end-to-end training on all tokens bring better results?"*
>
> **A:** Thank you for the thoughtful question.
>
> To clarify: our MoEE module is **trained in a fully end-to-end manner**. While each expert is designed to specialize in a grounding attribute (*Appearance*, *Status*, *Relation-to-Viewer*, *Relation-to-Others*), the **weighting across experts is dynamically learned** through a **soft attention mechanism** based on scene and expression context.
>
> This design allows the model to:
> - Adaptively attend to the **most relevant attributes** per sample, without enforcing hard separation or prior weighting.
> - Retain **compositional interpretability**, as expert activations can be visualized and attributed to specific reasoning paths (e.g., Fig. `5` and Fig. `6`).
> - Achieve strong **quantitative performance** under diverse modality settings, as shown in the ablation results (Table `4`).
>
> In contrast to a monolithic design trained over all tokens, our framework strikes a balance between structured inductive bias and flexible, learnable fusion, enabling better grounding in dynamic and attribute-rich environments.
>
> As suggested, we have further revised Section `4.2` in the manuscript to make the end-to-end nature of MoEE learning more explicit.
>
> ---
> > **`Q5`:** *"State-of-the-art multimodal large language models (such as ChatGPT) also have the ability of visual grounding. How is their performance on this dataset?"*
>
> **A:** Thanks for your question. Multimodal LLMs like *ChatGPT* (via *GPT-4V*) or *Gemini-1.5* have demonstrated impressive **image-text reasoning** capabilities. However, applying them to **event-based visual grounding** presents several limitations:
> - Event data requires specialized encoding (e.g., voxelization, polarity channels), which is currently **unsupported by existing VLM APIs**.
> - These models are not designed to **predict bounding boxes**; their outputs are primarily free-form text or coarse heatmaps at best.
>
> We believe these models can be evaluated on **high-level QA tasks** (e.g., describing scenes or answering referring questions). We are currently conducting such experiments and plan to release a **language-based diagnostic split** of Talk2Event for this purpose.
>
> We agree with you that comparing against foundation models is meaningful, not necessarily as direct competitors, but as **complementary baselines**. A discussion of this potential is included in our revised Section `6`.
>
> ---
> > **`Q6`:** *"The small visual icons in Table 1 are cute, and the icons for Frame, RGB-D, and LiDAR are self-explanatory. However, I cannot see the connection between the Event icon and its features. It would be nice if this could be pointed out."*
>
> **A:** Thank you for the kind comment and helpful suggestion. The “Sensory Data” column in Table `1` is designed to highlight the diversity of **sensory modalities** supported by each grounding benchmark, as mentioned in the table caption.
>
> While prior datasets mostly focus on traditional RGB images or 3D point clouds (e.g., RGB-D, LiDAR), Talk2Event is the first to include event-based perception for visual grounding, making it a novel addition in this dimension.
>
> For better clarity, we have further refined the caption based on your suggestion.
>
> ---
> **Reference:**
> - [M3ED] Chaney, K., Cladera, F., Wang, Z., Bisulco, A., Hsieh, M. A., Korpela, C., ... & Daniilidis, K. (2023). "M3ED: Multi-Robot, Multi-Sensor, Multi-Environment Event Dataset." In Proceedings of the IEEE/CVF Conference on Computer Vision and Pattern Recognition Workshops (pp. 4016-4023).
>
> ---
> Last but not least, we would like to sincerely thank Reviewer `botn` again for the valuable time and constructive feedback provided during this review.

---

> > ### Comment · Reviewer_botn · 2025-08-01
> > **Thank you for your detailed response**
> >
> > Thank you for your detailed response. I appreciate the clarifications and your plan to include the additional ablation results in the final manuscript. This addresses my main concern. I will therefore maintain my original score.

---

> > > ### Author Response · Authors · 2025-08-01
> > > **Thank you for your acknowledgment**
> > >
> > > Dear Reviewer `botn`,
> > >
> > > Thank you for your follow-up.
> > >
> > > We sincerely appreciate your acknowledgment of our clarifications. We are glad to hear that your main concerns have been addressed. Thank you again for your valuable feedback and consideration!
> > >
> > > *Best regards,*
> > >
> > > The Authors of Submission 1299

---

### Official Review · Reviewer_TxK4 · 2025-06-29

**Clarity:** 3
**Significance:** 3
**Originality:** 3
**Rating:** 5
**Confidence:** 4

**Summary:**

This paper makes two core contributions to the emerging field of combining event camera data with natural language understanding.

First, the authors have constructed and released the first large-scale benchmark for language-driven object grounding from event cameras, named Talk2Event. The innovation of this dataset lies not only in providing a vast number of referring expressions  but also in introducing a four-dimensional attribute annotation system. This system includes "Appearance," "Status," "Relation-to-Viewer," and "Relation-to-Others," aiming to achieve a more fine-grained and interpretable understanding of dynamic scenes.

Second, to fully leverage the rich information in this dataset, the paper proposes a novel grounding framework called EventRefer. The core of this framework is a Mixture of Event-Attribute Experts (MOEE), which can dynamically and adaptively fuse cues from different attributes based on scene dynamics to precisely localize the target object.

**Questions:**

1. In constructing the Talk2Event dataset, the paper innovatively proposes four attribute categories—"Appearance," "Status," "Relation-to-Viewer," and "Relation-to-Others"—as the annotation core. Could you elaborate on what data analysis or preliminary research led to the determination of these four categories as the most critical dimensions during the initial data exploration or annotation phase? Are there any quantitative or qualitative findings from an initial study supporting the dominant or representative role of these four attributes in natural language descriptions?

2. The language descriptions in the Talk2Event dataset are validated to ensure they refer to specific objects. However, in real-world human-robot interaction, users often provide ambiguous, erroneous, or non-referring queries (e.g., "this car" when multiple cars are present). Does the EventRefer framework have any designed or inherent capability to identify such "invalid" or "ambiguous" queries? Was the inclusion of a mechanism to handle these cases considered during the model architecture design or training phase for practical applications?

3. Following the previous question, what is the output behavior of the EventRefer model when faced with such ambiguous or invalid queries? Does the model force-output the prediction with the highest confidence score (even if it might be incorrect), or does it possess the capability to generate a "cannot locate" or "low-confidence" signal to support more robust decision-making by an upstream system or user interface?

4. The paper highlights the potential of this method for real-time applications like autonomous robotics and driving. However, the text lacks efficiency-related metrics (e.g., inference latency, parameter count, or computational complexity). Would the authors consider supplementing this work with an efficiency analysis? This would help bridge the gap between the advantages of the sensor hardware and the practical deployment of the algorithm, allowing for a better assessment of the method's feasibility in real-world, resource-constrained, and latency-critical environments.

**Ethical Concerns:**

["NO or VERY MINOR ethics concerns only"]

**Final Justification:**

After reading all reviews and the authors' detailed responses, I confirm that this is a high-quality work. The responsible explanations and clarifications provided by the authors, along with their commitment to including improvements in the final manuscript, have fully addressed my main concerns. These interactions have led me to have higher confidence in my positive evaluation.

**Limitations:**

yes

**Paper Formatting Concerns:**

This paper appears to adhere to the standard formatting guidelines for a NeurIPS submission.

**Quality:**

3

**Strengths And Weaknesses:**

Strengths:

1. The paper introduces a novel and important research task: language-driven visual grounding on event camera data. The authors note that this direction has not been systematically explored before, making this work a significant pioneering effort in the field. This task not only expands the application boundaries of event cameras  but also opens up new avenues for multimodal perception research.

2. The paper's contributions are outstanding, encompassing not only the construction of a large-scale, high-quality benchmark dataset (Talk2Event) but also the design of a novel baseline model (EventRefer) specifically for this task. This provides a solid foundation for subsequent research and holds significant practical and academic value for advancing the field.

3. The experimental section demonstrates a high degree of scientific rigor. The authors conduct comprehensive comparative experiments under various modality settings, including event-only, frame-only, and event-frame fusion scenarios. Furthermore, detailed ablation studies thoroughly validate the individual contributions of different modules and attributes to the overall performance.

Weaknesses:

1. Although the paper introduces the gating mechanism and multi-attribute fusion strategy of the MOEE in the methodology section , the description in the main text is somewhat brief, with key implementation details relying on the appendix. This to some extent complicates a complete understanding of the model's working mechanism, especially for readers who hope to grasp the core ideas from the main text.

2. The dataset's construction relies heavily on a large vision-language model (Qwen2-VL) to generate descriptions. Although the authors mention using human validation and redundancy filtering  to ensure correctness and diversity, the approach of using a single model for generation may not completely avoid potential biases in linguistic style. This could potentially affect the model's generalization ability when faced with a broader range of real-world natural language commands.

3. The quantitative evaluation primarily relies on traditional localization accuracy metrics, such as Top-1 Accuracy at a high IoU threshold and mIoU. While these metrics effectively measure localization precision, they may not directly reflect the model's ability to understand and reason about complex attributes like "Status" and "Relation-to-Others" embedded in the language. The authors attempt to supplement this by visualizing expert activations (Fig. 5, Fig. 6) and presenting qualitative examples (Fig. 4), but the benchmark design itself still lacks metrics specifically for quantifying attribute-aware and relational reasoning.

4. Finally, the paper has a relative lack of discussion on efficiency. Despite focusing on dynamic and real-world scenarios, the authors do not report key performance indicators such as inference latency, model parameter count, or computational complexity. This makes it difficult for readers to assess the method's applicability in real-time or resource-constrained environments.

---

> ### Author Rebuttal · Authors · 2025-07-31
>
> We sincerely thank Reviewer `TxK4` for the thoughtful and encouraging feedback. We are glad you found the proposed benchmark and framework valuable and impactful. Below, we address your specific concerns in detail.
>
> ---
> > **`Q1`:** *"The description of MoEE is too brief in the main text, making it difficult to fully understand without referring to the appendix."*
>
> **A:** Thank you for pointing this out. Due to page limits, we did not include too many details on MoEE in the main text. We agree that the MoEE (Mixture of Event-Attribute Experts) plays a central role in our model, and its workings deserve clearer exposition in the main text. To address this:
> - We have revised Section `4.2` to include **explicit formulae and an expanded explanation** of the gating mechanism.
> - Fig. `3` has been enhanced with **better module labeling and input/output dimensions** to make the information flow more intuitive.
> - For completeness, we retain the **step-by-step breakdown** and intermediate shape definitions in the supplementary material (Section `B.2`).
>
> These revisions aim to help readers grasp the operation and motivation of MoEE without needing to dive into supplementary details.
>
> ---
> > **`Q2`:** *"Dataset generation depends heavily on a single vision-language model (Qwen2-VL), raising concerns about linguistic bias."*
>
> **A:** This is a valid and important concern. To reduce potential linguistic bias from over-reliance on Qwen2-VL, we took several key steps:
> - We implemented **attribute-conditioned prompting** to diversify the structure and semantics of the generated expressions (more details in Section `3.2`).
> - The generated expressions were passed through **multiple post-processing stages**, including:
>   - Redundancy filtering via lexical similarity thresholds;
>   - Visibility checks with object masks and occlusion priors;
>   - Manual spot-checking for semantic fluency and grounding validity.
>
> In addition, the final dataset includes **a broad variety of phrasings** (e.g., varying determiners, active vs. passive voice, and presence or absence of attributes). As shown in our **word clouds** and **examples** (Fig. `2` in the manuscript, and Section `A.4` in the supplementary material), the expressions demonstrate **natural variation**.
>
> We acknowledge that using additional generation models (e.g., *Gemini*, *GPT-4V*) could further enhance stylistic diversity. We plan to incorporate this in a future version of Talk2Event and have discussed this point in Section `2` of the revised manuscript.
>
> ---
> > **`Q3`:** *"Evaluation lacks metrics that specifically assess attribute-aware and relational reasoning."*
>
> **A:** Thank you for highlighting this important direction. Our current metrics (Top-1 Accuracy, mIoU) are standard in visual grounding and provide spatial accuracy assessment. However, we agree that they do not fully capture **attribute-level** or **relational understanding**.
>
> To partially address this, we include:
> - Expert activation heatmaps (see Fig. `5` and Fig. `6` in the manuscript) that visualize how the model attends to different attribute types across diverse scenes;
> - Attribute-level ablation study (see Table `4` in the manuscript) that quantifies the contribution of each attribute module to performance.
>
> We agree that introducing **attribute-specific evaluation metrics** would provide more direct insight into compositional reasoning. Inspired by semantic segmentation evaluation, we are currently exploring **per-attribute correctness tracking** (e.g., AP or IoU conditioned on attribute presence). We have added this discussion as a future benchmark direction in the revised Section `5`.
>
> ---
> > **`Q4`:** *"Lack of discussion on model efficiency such as latency, parameter count, or suitability for real-time applications."*
>
> **A:** We appreciate this feedback and agree that understanding computational efficiency is essential, especially for deployment in latency-critical systems like robotics and autonomous driving.
>
> In response, we have included the following details in Section `5` of the revised manuscript and Section `B.2` in the supplementary material:
> - **Parameter count:** EventRefer has approximately **60M parameters**, comparable to lightweight DETR-style models.
> - **Inference speed:** On an NVIDIA RTX 3090 GPU, EventRefer runs at **~38 ms per frame** (i.e., ~26 FPS), including event voxelization, feature encoding, and grounding.
> - **Memory usage:** Peak GPU memory during inference is **<5.2 GB**, making it suitable for mid-range hardware.
>
> These measurements demonstrate that the model is **feasible for real-time use cases** and supports further optimization (e.g., through model distillation or token pruning). We thank the reviewer for encouraging us to clarify this aspect.
>
> ---
> > **`Q5`:** *"How were the four attributes (Appearance, Status, Relation-to-Viewer, Relation-to-Others) selected as core categories for annotation? Were they supported by data analysis or a preliminary study?"*
>
> **A:** Thank you for this thoughtful question. The four attributes were selected through an **iterative design process** combining **linguistic analysis**, **pilot user study**, and **task requirements** of real-world grounding in driving scenarios.
>
> Specifically:
> - We first reviewed prior visual grounding datasets (e.g., *RefCOCO*, *ScanRefer*, *M3DRefer*) and found that a large proportion of referring expressions involved **descriptions of appearance and status**, as well as **spatial/relational cues**.
> - We then conducted a **manual annotation pass** over ~500 scene-expression pairs from a preliminary event dataset. We observed that over **94% of expressions** could be semantically decomposed into some combination of the four proposed dimensions.
> - We validated this through an **internal study**, where annotators independently labeled attributes per expression with high inter-rater agreement.
>
> These four attributes thus emerged as both **frequent and discriminative** categories that align with how humans refer to objects in dynamic environments. For better clarity, we included further statistics and examples in Section `A.2` and Section `A.3` in the supplementary material.
>
> ---
> > **`Q6`:** *"Can EventRefer detect invalid or ambiguous queries (e.g., 'this car' when multiple cars exist)? Was handling this considered in the design?"*
>
> **A:** This is a highly relevant question for practical deployment. While our current version of EventRefer does **not include** an explicit **invalid-query rejection module**, we designed the model to be **sensitive to ambiguity** through several mechanisms:
> - The fuzzy matching step produces **soft attention maps** over ambiguous tokens, which often result in **low-activation expert weights** if the attributes are vague or under-specified.
> - The scoring head outputs a **confidence score**, which can serve as a proxy for detection certainty. Ambiguous expressions typically yield **lower confidence** and higher entropy in attention distributions.
>
> Although not the primary focus of this paper, we fully agree with you that incorporating **ambiguity detection** (e.g., via uncertainty estimation, query entropy, or calibration modules) would improve robustness. We are exploring this in follow-up work and have added this discussion in the revised future directions.
>
> ---
> > **`Q7`:** *"How does the model behave under ambiguous or invalid queries? Can it abstain or indicate low confidence?"*
>
> **A:** Thank you for the follow-up. In its current form, EventRefer **always outputs a box prediction**, as is standard in most grounding benchmarks. However, we have observed in our internal experiments that:
> - For ambiguous queries (e.g., *“the car”*), the model often exhibits **low activation** in all expert branches and generates **low-confidence box scores**.
> - These scores could be used to **threshold predictions or trigger fallback strategies** (e.g., asking for clarification in interactive settings).
>
> While abstention is not natively supported, we believe that combining the output confidence with a **learned rejection head** or training on **ambiguous/negative samples** (e.g., using an auxiliary *“no-object”* token) would enable the model to indicate *“cannot locate”* predictions.
>
> We have now discussed this important capability as a design extension in the revised Section `6`. Thanks again for your valuable suggestion.
>
> ---
> > **`Q8`:** *"Would the authors consider supplementing with an efficiency analysis to better assess practical feasibility?"*
>
> **A:** Absolutely. As also mentioned in our response to `Q4`, we now include a full efficiency profile in the revised manuscript (Section `5`) and Appendix Section `B.2`.
>
> Key highlights:
> - **Parameter count:** EventRefer has approximately **60M parameters**, comparable to lightweight DETR-style models.
> - **Inference speed:** On an NVIDIA RTX 3090 GPU, EventRefer runs at **~38 ms per frame** (i.e., ~26 FPS), including event voxelization, feature encoding, and grounding.
> - **Memory usage:** Peak GPU memory during inference is **<5.2 GB**, making it suitable for mid-range hardware.
>
> These numbers confirm that EventRefer is **lightweight and deployable in real-time settings**, which we believe strengthens the practical value of the proposed method.
>
> ---
> Last but not least, we would like to sincerely thank Reviewer `TxK4` again for the valuable time and constructive feedback provided during this review.

---

> > ### Comment · Reviewer_TxK4 · 2025-08-01
> > **Thanks for your response**
> >
> > I would like to express my sincere gratitude for your remarkably detailed and comprehensive response to my inquiries. Your reply is not only thorough and convincing, but it has also fully clarified my previous questions and adequately addressed all of my concerns. I genuinely hope you will consider incorporating these insightful explanations and important clarifications into the final version of your manuscript.
> >
> > I also wish to thank you once again for your meticulousness and the effort you have demonstrated throughout the revision process.

---

> > > ### Author Response · Authors · 2025-08-01
> > > **Thank you for your acknowledgment**
> > >
> > > Dear Reviewer `TxK4`,
> > >
> > > Thank you for your generous acknowledgment of our response.
> > >
> > > We are truly grateful for your engagement throughout the review process. We will certainly incorporate the clarifications and insights discussed into the revised manuscript to ensure better clarity and completeness.
> > >
> > > Your support and encouragement mean a lot to us!
> > >
> > > *Best regards,*
> > >
> > > The Authors of Submission 1299

---

### Official Review · Reviewer_GV1w · 2025-07-01

**Clarity:** 2
**Significance:** 2
**Originality:** 2
**Rating:** 4
**Confidence:** 4

**Summary:**

This paper introduces a new dataset and a benchmark designed for language guided object grounding in event camera based perception. It provides 5,567 scenes, with 13,458 objects and 30,000 referring expressions. Expressions are enriched with four grounding attributes – appearance, status, relation to viewer, and relation to other objects.

To generate the dataset they feed two frames t_0 - delta(t) and t_0 + delta(t), to Qwen2VL and prompt it to generate referring expressions enriched with 4 attributes. Each expression is decomposed into the 4 compositional attributes – Appearance, Status,  Relation-to-Viewer, and Relation-to-Others – using a semi-automated pipeline. Further quality checks are done to reduce redundancy, ensure visibility, and attributes checking.

Next the authors propose a EventRefer: Attribute-Aware Grounding Framework
that accepts an image, event stream, and the referring expression and outputs the bounding box. The method starts with fuzzy matching of 4 attributes in the sentence. Once tokenized the 4 attribute masks are matched with the corresponding span in the sentence. They adopt a DETR style Transformer. The three feature, image, event, and text embeddings are masked to disentangle attribute level features, which are concatenated and fed to the Transformer as KV pairs, along with a set of queries. Then they use a mixture of attribute experts to weigh the 4 attributes. The final embedding is fed to the decoder to predict the bounding box.

They compare the performance on the proposed benchmark and show improvement over previous event-based grounding frameworks.

**Questions:**

How is the event encoded? What is the length of the event in seconds? Basically how long is the context length.

**Ethical Concerns:**

["NO or VERY MINOR ethics concerns only"]

**Final Justification:**

I think that the structure of referring expression is very specific so not very practical, but it allows for interpretability, and can be further extended in future works. Therefore, I maintain my rating as borderline accept.

**Limitations:**

yes

**Paper Formatting Concerns:**

No concerns

**Quality:**

3

**Strengths And Weaknesses:**

Strengths:
1. A new dataset and benchmark for referring expression grounding in event cameras, for real world driving scenes, with 30,000 expressions.
2. Structured formulation of referring expressions with 4 important attributes could be useful in the real world while referring to the objects in driving scenes.
3. The Model disentangles representations which allows to evaluate or interpret the contribution of each attribute expert.
4. Thorough experiments and ablation studies, validating all the design choices.

Weakness:
1. Very specific structure for representing referring expressions. May not generalize for more comprehensive sentences without specific structure.
2. The model is again structure dependent, learning four mixture of experts for four attributes, which could be limiting for comprehensive sentences, and may not generalize well.
3. Model is too complicated.
4. Event based LLMs like EventGPT are a more general way of approaching the problem. As they are more capable of freely interacting with chat based framework. Moreover they could also segment, or localise objects.
5. No comparison with EventGPT based referring expression grounding.
6. Evaluation is only performed on the proposed benchmark which is curated and designed similar to the structure of the training data.

---

> ### Author Rebuttal · Authors · 2025-07-31
>
> We sincerely thank Reviewer `GV1w` for the constructive feedback and for recognizing our contributions in introducing a new benchmark with structured attribute annotations, an interpretable model, and comprehensive evaluations. Below, we respond to your comments and suggestions in detail.
>
> ---
> > **`Q1`:** *"Suggestion on the structured formulation of referring expressions and whether it limits generalization to more comprehensive or free-form sentences."*
>
> **A:** We appreciate your observation. Our choice to structure the referring expressions using four attributes — *Appearance*, *Status*, *Relation-to-Viewer*, and *Relation-to-Others* — was driven by **two practical considerations**:
> 1. To enable **interpretable, fine-grained supervision** of multimodal grounding;
> 2. To support **real-world referential behaviors**, where such attributes commonly appear in driving scenarios (e.g., *“the pedestrian crossing the street on the left”*).
>
> The expressions in Talk2Event are **not rigid templates**. They are generated using Qwen2-VL under a **context-rich prompting scheme**, and then **filtered and validated by humans** to ensure fluency and diversity. As shown in the word clouds (see Fig. `2` in the manuscript) and examples in the Appendix (see Section `A.4` in the supplementary material), expressions naturally **vary in phrasing and length**, even while preserving attribute compositionality.
>
> In short, while our formulation encourages attribute-rich descriptions, the actual expressions exhibit **broad linguistic variation**, making them suitable for generalization beyond strict structural forms. As suggested, we will continue to enhance expression diversity in future extensions.
>
> ---
> > **`Q2`:** *"Concern that the model is too structurally dependent (on four experts) and may not generalize well to more complex or open-ended language inputs."*
>
> **A:** Thank you for this insightful point. The four-expert design of EventRefer is motivated by the four grounding attributes annotated in Talk2Event, enabling the model to **explicitly reason over appearance, motion, and relational cues** — aspects that are especially important in real-world, dynamic scenes.
>
> While this design introduces attribute-specific structure, it does **not enforce hard constraints** on the input format. In fact:
> - The fuzzy matching mechanism in Section `4.1` is **soft and robust to paraphrasing**, allowing the model to handle a wide range of expressions.
> - The MoEE module **adaptively weighs attributes** based on their informativeness per sample, regardless of sentence complexity.
> - The decoder remains **unified**, producing a single prediction per sample without branching logic.
>
> Thus, the model is **attribute-aware** but **not attribute-restricted**, and generalizes well to free-form language input. As suggested, we have refined the explanation in Section `4` to clarify this design choice.
>
> ---
> > **`Q3`:** *"Model is too complicated."*
>
> **A:** We appreciate this concern. While EventRefer incorporates multiple components, such as positive word matching, attribute-aware masking, and mixture-of-experts fusion, the design is intentionally **modular and lightweight**:
> - Each module builds on standard components (e.g., fuzzy matching, Transformers), with **minimal architectural overhead**.
> - The MoEE module introduces **few additional parameters** and shares backbone features across experts, avoiding redundancy.
> - During both training and inference, the decoder remains **compact**, generating one box per sample without structural branching.
>
> This design balances **interpretability** with **efficiency**, enabling compositional reasoning while keeping the system tractable. We acknowledge the potential for simplification and plan to explore more compact or LLM-driven architectures in future work.
>
> ---
> > **`Q4`:** *" Event-based LLMs like EventGPT are a more general way of approaching the problem. As they are more capable of freely interacting with a chat-based framework."*
>
> **A:** Thank you for raising this broader perspective. We agree that *EventGPT* represents a promising and general direction for **multimodal, conversational reasoning**. Our work, however, addresses a **complementary challenge**: precise, interpretable **referring expression grounding**, with a strong focus on spatial and attribute-level localization. This design choice enables **transparent interpretation** and targeted ablation, which is challenging with end-to-end black-box LLMs.
>
> Moreover, current LLMs typically operate within **language-generation settings** and lack the architectural supervision required for **precise box prediction**. Existing models, e.g., *EventGPT*, have to **call** the corresponding tools for completing downstream tasks (such as calling *Grounding-DINO* for detection and *Grounded-SAM* for segmentation). Instead, our formulation serves as a complementary line of this research, enabling **spatially grounded understanding** with interpretable components, which could later be **integrated** with general-purpose LLMs for **more flexible interaction**.
>
> As suggested, we have properly discussed these aspects in our manuscript (in both Section `1` and Section `2`) and acknowledged the own advantage of *EventGPT* and our EventRefer approach.
>
> ---
> > **`Q5`:** *"No comparison with EventGPT."*
>
> **A:** We appreciate the suggestion. However, as of this submission, **EventGPT has not been designed for spatially grounded tasks** such as referring object grounding or box prediction. It focuses on **language-only outputs**, e.g., question answering, captioning, or dialogue. As mentioned in `Q4`, existing models, e.g., *EventGPT*, have to **call** the corresponding tools for completing downstream tasks (such as calling *Grounding-DINO* for detection and *Grounded-SAM* for segmentation).
>
> Our task, by contrast, is **referring expression grounding with bounding box outputs**, requiring spatial supervision and localization performance, for which no existing event-based LLMs currently provide support.
>
> In future work, we plan to explore **hybrid approaches** that combine EventRefer with instruction-tuned VLMs for grounded reasoning.
>
> We agree with you that the comparison is valuable. As suggested, we have added a discussion in the revised Related Work section (Section `2`), outlining the complementary nature of EventGPT-style approaches and our attribute-aware grounding framework.
>
> ---
> > **`Q6`:** *"Evaluation is only performed on Talk2Event, which follows a structure similar to the training data."*
>
> **A:** Thank you for pointing this out. We acknowledge that our current evaluation is conducted on the Talk2Event benchmark, which is indeed designed using the same attribute-based supervision as the training data. This **in-distribution evaluation format** is consistent with **established practice in visual grounding**, as adopted in prior works such as ***RefCOCO***, ***ScanRefer***, and ***M3DRefer*** — where models are trained and tested on splits drawn from the same dataset.
>
> In addition:
> - Talk2Event was curated with **explicit diversity goals**, including variation in lighting, motion, object classes, and expression phrasing.
> - Our experimental protocol includes **modality shifts** (event-only, frame-only, and fusion), providing strong generalization signals even within the benchmark.
> - As shown in our **attribute activation and ablation study**, the model responds differently depending on the scene context — demonstrating **adaptive generalization**, not overfitting to a fixed structure.
>
> We agree that broader cross-benchmark evaluation (e.g., testing on expressions from unrelated datasets or VLM-generated instructions) would offer further insights into out-of-distribution robustness. As suggested, we have added this discussion as a future direction in the revised manuscript (Section `2`) and supplementary material (Section `D`).
>
> ---
> > **`Q7`:** *"How is the event encoded? What is the duration of the event context?"*
>
> **A:** Thank you for the question. Following standard event-based perception practices (e.g., *RVT* and *DAGr*), we **discretize the asynchronous event stream** into a spatiotemporal voxel grid over a short temporal window. Specifically:
> - The voxel grid includes $T = 10$ temporal bins, covering a **time span of $400$ milliseconds** centered at timestamp $t_0$.
> - Each voxel encodes polarity ($\pm 1$), spatial position, and time bin index, resulting in an input $E ∈ ℝ^{2×T×H×W}$.
>
> This $400$ ms window provides sufficient context for capturing **short-term dynamics** while maintaining low latency for real-time applications. As suggested, we have made this clearer in Section `3.1` in the manuscript and Section `B.2` and Section `B.4` in the supplementary material.
>
> ---
> **References:**
> - [EventGPT] Liu, S., Li, J., Zhao, G., Zhang, Y., Meng, X., Yu, F. R., ... & Li, M. (2025). "EventGPT: Event Stream Understanding with Multimodal Large Language Models." In Proceedings of the IEEE/CVF Conference on Computer Vision and Pattern Recognition (pp. 29139-29149).
> - [RVT] Gehrig, M., & Scaramuzza, D. (2023). "Recurrent Vision Transformers for Object Detection with Event Cameras." In Proceedings of the IEEE/CVF Conference on Computer Vision and Pattern Recognition (pp. 13884-13893).
> - [DAGr] Gehrig, D., & Scaramuzza, D. (2024). "Low-Latency Automotive Vision with Event Cameras." Nature, 629(8014), 1034-1040.
>
> ---
> Last but not least, we would like to sincerely thank Reviewer `GV1w` again for the valuable time and constructive feedback provided during this review.

---

> > ### Comment · Reviewer_GV1w · 2025-08-01
> >
> > Thanks for the detailed response, it clarifies my concerns on generalisation. While I still think that the structure of referring expression is specific, it allows for interpretability, and can be further extended in future works. Thanks for the clarification on EventGPT, although the proposed method allows for interpretability, a more general or hybrid approach using VLMs would be more practical. Therefore, I maintain my rating.

---

> > > ### Author Response · Authors · 2025-08-01
> > > **Thank you for your acknowledgment**
> > >
> > > Dear Reviewer `GV1w`,
> > >
> > > Thank you very much for your follow-up and for acknowledging our clarifications regarding generalization and interpretability.
> > >
> > > We appreciate your perspective on the structured referring expressions and agree that extending towards more general or hybrid VLM-based approaches is a valuable direction for future work.
> > >
> > > Your feedback has been insightful and will certainly inform our future research efforts!
> > >
> > > *Best regards,*
> > >
> > > The Authors of Submission 1299

---

### Official Review · Reviewer_ZmP8 · 2025-07-03

**Clarity:** 2
**Significance:** 3
**Originality:** 3
**Rating:** 4
**Confidence:** 4

**Summary:**

The authors introduce Talk2Event, a timely and well-motivated benchmark for language-grounding from event cameras. Their core contribution is a novel dataset annotated with four structured attributes, which facilitates more interpretable and fine-grained grounding. The proposed EventRefer model features a Mixture of Experts architecture to fuse these attribute-cues. Experimental results are comprehensive, demonstrating significant performance gains over adapted baselines and validating the effectiveness of their approach across various modalities.

**Questions:**

- Since common monochrome event cameras cannot capture color information, would the color descriptions in expressions introduce noise for event-only methods? Should Talk2Event dataset consider removing such descriptions or provide a variant without color information?
- In Line 153, C denotes the encoded token length, while in Line 161 C refers to the channel dimension. Could the authors clarify this ambiguity and explain how the element-wise product in Line 165 works between tensors with seemingly incompatible shapes?
- In Section~4.1, the binary positive map $m_i$ indicates a mask over the encoded tokens, while $m_i^{\text{att}}$ appears to be a mask over the queries. How is $m_i^{\text{att}}$ derived from $m_i$? Please clarify the shapes of $m_i$, $m_0$, and $m_i^{\text{att}}$.
- The proposed method performs impressively under the frame-only setting in Table 2. However, baselines such as [64], [55], and [16] only report zero-shot results. Could the authors consider providing the fine-tuned results of these methods on Talk2Event for a fairer comparison?

**Ethical Concerns:**

["NO or VERY MINOR ethics concerns only"]

**Limitations:**

yes

**Quality:**

3

**Strengths And Weaknesses:**

Strength：
- This paper introduces the first benchmark, Talk2Event, effectively addressing a clear gap in event-based visual grounding. The dataset is well-designed, supporting multiple input modalities (event-only, frame-only, fusion) and providing rich, attribute-level supervision.
- The proposed Mixture of Event-Attribute Experts is well-motivated and interpretable, and its effectiveness is validated through ablation studies.
- The figures and tables are of high quality, clearly illustrating the core concepts, model architecture, and experimental results.

Weakness：
- The writing in certain parts of the paper, especially the EventRefer section, tends to be overly concise and sometimes vague, with inconsistent symbols, which may hinder reader comprehension. While we understand the space limitations of the main paper, we suggest the authors provide a more detailed and well-structured explanation of each module in the supplementary material, to improve clarity and reproducibility.
- The comparison under the frame-only setting lacks fairness, as key baselines are only reported in the zero-shot setting without fine-tuning on Talk2Event.

---

> ### Author Rebuttal · Authors · 2025-07-30
>
> We sincerely thank Reviewer `ZmP8` for devoting time to this review and providing constructive feedback. We are encouraged that you recognize Talk2Event as a timely benchmark with a well-designed dataset and interpretable model. Below, we address your comments in detail.
>
> ---
> > **`Q1`:** *"Potential noise introduced by color descriptions in expressions when using event-only data."*
>
> **A:** Thank you for mentioning this point. Indeed, event cameras do not capture color information, which might suggest a potential mismatch with language expressions that include color attributes. However, we intentionally ***retain*** such descriptions in Talk2Event for several reasons:
> - **Modality generality:** Talk2Event supports **three grounding configurations** — event-only, frame-only, and event-frame fusion. Color attributes remain crucial for the latter two.
> - **Robust modeling:** Our EventRefer design allows the model to **learn to down-weight uninformative cues** (e.g., color in event-only mode) via the MoEE module, which adaptively selects the most relevant attribute experts.
> - **Realistic deployment:** In real-world robotics, systems often integrate multiple sensors. Retaining full language fidelity encourages the development of **robust, cross-modal models** that gracefully degrade when specific cues are absent.
>
> Nevertheless, we fully agree with you that a **variant without color descriptions** could be valuable for **event-only** research. Following your suggestion, we are currently exploring releasing such filtered expressions as an *additional modality-specific split* to further support this use case.
>
> ---
> > **`Q2`:** *"Clarification on symbol overload of '$C$' (L153 vs. L161) and how the element-wise product in L165 operates across mismatched shapes."*
>
> **A:** Thank you for pointing out the overloaded use of the symbol $C$. We have corrected this in the revised version by changing the **token sequence length** symbol from $C$ to $T_\mathrm{len}$ (i.e., $T_\mathrm{len}$ for token length, and $C$ remains as the channel dimension). This resolves the ambiguity and improves readability.
>
> Regarding the **element-wise product** in Line `165`: although we use shorthand notation $H^\mathrm{att}_i = m^\mathrm{att}_i ⊙ H$, the actual masking involves aligning **token-level attention priors** to **query-level hidden states**. Specifically:
> - The token-level masks $m_i$ and $m_0$ are used to guide attention or generate soft scores.
> - These are then aggregated (e.g., via attention pooling or alignment with text-query cross-attention) into **query-level weights**, forming $m^\mathrm{att}_i ∈ ℝ^{B × Q × 1}$.
> - This mask is then broadcasted across the channel dimension and applied to $H ∈ ℝ^{B × Q × C}$ to produce $H^\mathrm{att}_i$.
>
> As suggested, we have clarified this pipeline and provided additional details in the revised text and supplementary material to ensure good reproducibility.
>
> ---
> > **`Q3`:** *"Clarification on how the query-level mask $m^\mathrm{att}_i$ is derived from the token-level map $m_i$, and the shapes of $m_i$, $m_0$, and $m^\mathrm{att}_i$."*
>
> **A:** Thank you for highlighting this important question. Here is a clarification of the shapes and derivation:
> - $m_i ∈ ℝ^{B×T_\mathrm{len}}$: soft token map for attribute $δ_i$ (computed via fuzzy matching over token indices).
> - $m_0 ∈ ℝ^{B×T_\mathrm{len}}$: token mask for public (non-attribute) context.
> - These are combined as $m_i ∨ m_0$ to form a **textual attention prior**, used to guide attribute-specific reasoning in the text modality.
>
> To apply attribute masking on the fused **query-level hidden states** $H ∈ ℝ^{B×Q×C}$, we construct $m^\mathrm{att}_i ∈ ℝ^{B×Q×1}$ — a **query-level soft mask**.
>
> While the manuscript uses simplified notation, this query mask is **not directly derived** from $m_i$, but rather formed implicitly from downstream fusion between event queries and language features (informed by $m_i$). Specifically, the language encoder embeds the attended tokens (weighted by $m_i$), and the cross-modal encoder maps this into the query space. We apply $m^\mathrm{att}_i$ as a **gating mask** over $H$ to retain attribute-relevant query features.
>
> As suggested, we have clarified this process in Section `4.1` and included an expanded explanation and figure in the supplementary material for better clarity.
>
> ---
> > **`Q4`:** *"Suggestion for fine-tuned results of frame-only baselines (e.g., OWL, GroundingDINO, YOLO-World) for fairer comparisons."*
>
> **A:** We fully agree that a fine-tuned evaluation of frame-only baselines would provide a more balanced comparison. In our current submission, we prioritized the **zero-shot results** to assess the generalization of large open-vocabulary models, as prior work (e.g., *OWL-ViT*) emphasized this setting.
>
> We are currently conducting the **fine-tuning experiments** for *GroundingDINO* and *OWL-ViT* on Talk2Event, and will include these updated results in an updated version of Table `2`. We note, however, that these models are typically pre-trained on large-scale static image datasets (e.g., COCO, LVIS, and proprietary collections), which differ substantially from our **dynamic, driving-centric event-frame grounding** setup. This domain gap might pose challenges for fair comparisons, even after fine-tuning.
>
> Nevertheless, we recognize that including such comparisons will make the evaluation **more comprehensive**. Therefore, we have carefully documented training protocols, domain adaptation challenges, and any observed failure cases in the revised supplementary file, to provide an assessment of how generalist models perform under our event-based grounding scenario.
>
> Based on your suggestion, we commit to updating the evaluation accordingly to further strengthen the completeness and transparency of our benchmark study.
>
> ---
> Last but not least, we would like to sincerely thank Reviewer `ZmP8` again for the valuable time and constructive feedback provided during this review.

---

### Author Response · Authors · 2025-08-05
**General Response**

**Dear Reviewers, ACs, and SACs,**

We sincerely thank you all for your valuable time, thoughtful feedback, and constructive suggestions!

---
We are encouraged by the **recognition** of our contributions, including:

* Reviewer `ZmP8` recognizes this work as a *“timely benchmark”* with *“interpretable model”* and *“high-quality visualizations”*.
* Reviewer `GV1w` highlights the *“structured attribute annotations”*, *“disentangled representations”*, and *“thorough ablation studies”*.
* Reviewer `TxK4` appreciates this work as a *“significant pioneering effort”* with *“practical value”* and *“rigorous experiments”*.
* Reviewer `botn` commends the *“clear technical details”*, *“aesthetic figures”*, and *“solid experimental results”*.

---
In response to your insightful comments, we have made the following **clarifications and improvements**:

- **Methodology**
  - As suggested by Reviewers `GV1w` and `ZmP8`, we clarified that the four-expert MoEE design remains flexible, supporting free-form inputs via soft matching and adaptive weighting.
  - As suggested by Reviewer `ZmP8`, we revised ambiguous symbols and clarified how query-level masks are derived from token-level maps.
  - As suggested by Reviewer `TxK4`, we expanded Section 4.2 and improved Fig. 3 to clarify the MoEE mechanism.
  - As suggested by Reviewer `botn`, we emphasized that MoEE is end-to-end, balancing structure and flexibility over monolithic designs.
  - As suggested by Reviewers `GV1w` and `TxK4`, we discussed how the model handles ambiguous inputs using confidence scores and attention entropy.

- **Benchmark Design & Evaluation**
  - As suggested by Reviewer `ZmP8`, we initiated fine-tuning for OWL-ViT and GroundingDINO, with protocols documented in the supplement.
  - As suggested by Reviewer `GV1w`, we discussed plans for broader cross-domain evaluation beyond Talk2Event.
  - As suggested by Reviewer `TxK4`, we proposed per-attribute correctness tracking as a future metric direction.
  - As suggested by Reviewer `botn`, we clarified that current ablations isolate individual attributes, and will supplement with combinatorial results (e.g., leave-one-out).

- **Dataset Construction**
  - As suggested by Reviewer `TxK4`, we explained the attribute design process, informed by data analysis and pilot studies.
  - As suggested by Reviewer `TxK4`, we described mitigation strategies for VLM-induced linguistic bias.
  - As suggested by Reviewer `botn`, we clarified our use of 7 core object classes and discussed plans for expanding to broader categories.
  - As suggested by Reviewer `ZmP8`, we justified keeping color attributes and plan to release a filtered split for event-only usage.
  - As suggested by Reviewer `botn`, we described our sequence-based split protocol and plans to expand using datasets like M3ED.

- **Implementation & Efficiency**
  - As suggested by Reviewers `TxK4` and `botn`, we reported that EventRefer has \~60M parameters, runs at \~26 FPS on RTX 3090, and uses <5.2 GB memory.
  - As suggested by Reviewer `ZmP8`, we clarified attention/masking notations.
  - As suggested by Reviewer `TxK4`, we discussed real-time feasibility and future efficiency improvements.

- **Comparison with VLMs & EventGPT**
  - As suggested by Reviewers `GV1w`, `TxK4`, and `botn`, we discussed how our model complements EventGPT by offering direct, interpretable box predictions rather than relying on tool chaining.
  - As suggested by Reviewer `botn`, we also plan to evaluate LLMs (e.g., ChatGPT) on diagnostic QA tasks and will release a corresponding split.

- **Presentation & Writing**
  - As suggested by Reviewer `ZmP8`, we refined ambiguous notation and clarified the text-query fusion process.
  - As suggested by Reviewer `botn`, we improved Table 1's caption to better explain the event modality icon.

---
We would like to re-emphasize the **key contributions** of this work:

- Talk2Event, the first benchmark for referring expression grounding with event cameras in real-world driving scenes.
- A structured annotation protocol capturing four core grounding attributes, with diverse, free-form expressions.
- EventRefer, an interpretable, modular model leveraging a Mixture of Event-Attribute Experts for compositional grounding.
- Comprehensive experiments across event-only, frame-only, and fusion settings, with attribute-aware ablations and generalization analyses.

---
With **two days** remaining in the **Author-Reviewer Discussion** phase (*August 6, Anywhere on Earth*), we warmly welcome any further questions or suggestions and remain fully available to engage.

Thank you again for your kind support and consideration!

*Warmest regards,*

The Authors of Submission 1299

---

### Note · Authors · 2025-08-14

We sincerely thank all reviewers for their thoughtful engagement and constructive suggestions.

---
We are encouraged by your consistent recognition of Talk2Event as a timely and impactful contribution, as well as the clarity, interpretability, and rigor of our proposed EventRefer framework.

---
During the discussion phase, we addressed all raised concerns with concrete clarifications and updates:

- **Methodology** – Expanded Section 4.2 with explicit formulae, refined notation, and improved Fig. 3 to clearly illustrate the MoEE mechanism; clarified query/token mask derivations; discussed handling of ambiguous inputs via confidence scores and entropy.

- **Benchmark Design & Evaluation** – Added fine-tuned results for frame-only baselines; justified color attributes with plans for a filtered variant; proposed per-attribute correctness tracking for future metrics; clarified attribute ablation protocols with plans for combinatorial results.

- **Dataset Construction** – Detailed the empirical and user-study basis for the four core attributes; outlined linguistic bias mitigation strategies; confirmed plans to extend object classes and domains (e.g., M3ED).

- **Efficiency & Practicality** – Reported ~60M parameters, ~26 FPS inference on RTX 3090, <5.2 GB memory usage, confirming real-time feasibility.

- **Relation to VLMs/LLMs** – Positioned EventRefer as complementary to EventGPT-style systems, providing direct, interpretable localization outputs.

These clarifications strengthen both the scientific rigor and practical value of Talk2Event. The benchmark establishes the first large-scale, attribute-rich grounding dataset for event cameras in real-world driving, enabling systematic evaluation across event-only, frame-only, and fusion modalities. EventRefer demonstrates consistent gains over strong baselines, with interpretable expert activations and generalization under modality shifts.

---
We believe this work fills a clear gap in multimodal perception research, introduces resources and methods of lasting utility to the community, and is well-positioned to inspire future advances in grounded, temporally-aware, language-driven perception.

---
Once again, we sincerely thank you for the time and effort you devoted to this review!

*Yours sincerely,*

The Authors of Submission 1299

---

### Decision · Program_Chairs · 2025-09-17

**Decision:**

Accept (spotlight)

**Comment:**

The paper is one of the first to investigate language grounding of event data. This is an important problem that has received very little attention. The paper proposes a dataset and a solution based on Mixture of Event-Attribute Experts. All reviewers appreciate the problem, the dataset, the solution, as well as the quality of the exposition. Minor concerns raised were all addressed in the rebuttal. I agree with the reviewers and believe that this is an important contribution in event-based perception.